# Citizen rain gauges improve hourly radar rainfall bias correction using a two-step Kalman filter

Punpim Puttaraksa Mapiam [1*], Monton Methaprayun [1], Thom Adrianus Bogaard [2], Gerrit Schoups [2], Marie-Claire Ten Veldhuis [2]

[1]Department of Water Resources Engineering, Kasetsart University, PO Box 1032, Bangkok, 10900, Thailand
[2]Department of Water Management, Delft University of Technology, PO Box 5048, 2600 GA Delft, The Netherlands

*Correspondence to*: Punpim P. Mapiam (punpim.m@ku.th)

**Abstract.** Low density of conventional rain gauge networks is often a limiting factor for radar rainfall bias correction. Citizen rain gauges offer a promising opportunity to collect rainfall data at higher spatial density. In this paper hourly radar rainfall bias adjustment was applied using two different rain gauge networks consisting of tipping buckets (measured by Thai Meteorological Department, TMD) and daily citizen rain gauges. The radar rainfall bias correction factor was sequentially updated based on TMD and citizen rain gauge data using a two-step Kalman filter to incorporate the two-gauge datasets of contrasting quality. Radar reflectivity data of Sattahip radar station, gauge rainfall data from the TMD and citizen rain gauges located in Tubma basin, Thailand were used in the analysis. Daily data from the citizen rain gauge network were downscaled to hourly resolution based on temporal distribution patterns obtained from radar rainfall time series and the TMD gauge network. Results show that an improvement of radar rainfall estimates was achieved by including the downscaled citizen observations compared to bias correction based on the conventional rain gauge network only. These outcomes emphasize the value of citizen rainfall observations for radar bias correction, in particular in regions where conventional rain gauge networks are sparse.

**Keywords**: Kalman filter, citizen rain gauge, radar rainfall, bias correction, downscaling, Thailand.

## 1 Introduction

Hydrometeorological hazards, like flash floods and landslides cause severe damage to economies, properties, and human lives worldwide. In this context, flood forecasting and warning systems are a valuable non-structural measure to mitigate damage. However, such systems require input of rainfall data at a high spatial and temporal resolution. In most regions of the world, automatic rain gauge networks are insufficient for this purpose. Weather radar, which can better capture the variation of rainfall fields at fine spatial and temporal resolutions could be used as an alternative rainfall product for improving the accuracy of flash flood estimates and warnings. (Collinge and Kirby, 1987; Sun et al., 2000; Uijlenhoet 2001; Bedient et al., 2003; Creutin and Borga, 2003; Mapiam et al., 2009a, 2014; Mapiam and Chautsuk, 2018; Corral et al., 2019). However, a weather radar provides an indirect measurement of backscattered electromagnetic waves called radar reflectivity data (Z)

and quantitative estimation of radar rainfall data (R) is acknowledged as a complex process. Various sources of errors affect radar rainfall estimates, mainly errors in reflectivity measurements and reflectivity-rainfall conversion (Jordan et al., 2000). Correction of these two sources of error is a crucial procedure to increase the accuracy of radar rainfall estimates. Ground-truthing by rain gauge data is required to calibrate the Z-R relationship ($Z=AR^b$). The calibrated parameter *A* in the *Z-R* relationship will include any errors in the radar system caused by the electrical calibration of the radar (Seed et al., 2002).

The calibrated Z-R equation is used to convert the measured instantaneous reflectivity data to rainfall intensity and thereafter accumulating them into the required temporal resolution. However, the parameters A and b vary significantly, even within a single storm event depending on the rainfall characteristics which can exhibit a highly dynamic raindrop size distribution (DSD) (Ulbrich, 1983: Smith et al., 2009). Additionally, past studies found that the Z-R parameters are sensitive to the temporal resolution of rain gauge rainfall data that is used for the Z-R calibration (Hitchfeld and Bordan, 1954; Smith et al., 1975; Wilson and Brandes, 1979; Klazura, 1981; Steiner et al., 1995; Mapiam and Sriwongsitanon, 2008; Mapiam et al., 2009b). Consequently, an important source of error remains associated with the Z-R conversion process (Jordan et al., 2000; Berne and Krajewski, 2013). Many researchers attempted to correct this kind of error by classification of the measured reflectivity data into different storm types and thereafter constructing the Z-R equation corresponding to the classified storm characteristics. (Joss and Waldvogel, 1970; Rogers, 1971; Battan, 1973; Klazura, 1981; Austin, 1987; Rosenfeld et al., 1992, 1993; Tokay and Short, 1996; Amitai, 2000; Arai et al., 2005; Fang et al., 2018). For the effect of using rain gauge data with different temporal resolutions on Z–R relationships, Mapiam et al. (2009b) developed a universal scaling transformation function for converting the reference parameters A (obtained from using daily gauge rainfall data in the calibration) to the parameter A for sub-daily resolutions. This improved accuracy of the estimated sub-daily radar rainfall, especially in locations with limited short-duration rain gauge measurements.

After Z-R conversion, bias is expected to remain between the assessed radar rainfall and the true rainfall amount at the rain gauge locations if a fixed Z-R relationship is used to estimate radar rainfall over the entire radar domain (Chumchean et al., 2006; Wang et al., 2015). An effective bias correction technique is key for enhancing the quality of radar rainfall estimates (Steiner et al., 1999) and to remove the residual errors between radar rainfall obtained from the Z-R relationship and rain gauge data. Mean field bias (MFB) adjustment is the conventional method to obtain a static bias factor which assumes that the Z-R relationship is homogeneous in space but varies in time (Smith et al., 2007; Vieux and Bedient, 2004; Wilson, 1970). In this method, a multiplicative correction factor is applied uniformly across the radar coverage. Since the MFB approach does not consider noise and uncertainty of the rain gauge observations, nor spatial variability in observation bias, this can lead to large errors in radar rainfall estimates, particularly in areas where the density of rain gauge networks is limited. Kalman filter (KF) is an efficient algorithm that has been applied to correct the spatially uniform mean field bias, especially in real-time by accounting for the temporal variation of the mean bias as well as uncertainties in the ground rainfall measurements (Ahnert, 1986; Smith and Krajewski, 1991; Anagnostou et al., 1998; Seo et al., 1999; Dinku et al., 2002; Chumchean et al., 2006).

Previous studies used the KF for predicting and correcting the mean field bias to mitigate the observation error

variances affecting the mean field bias estimate. Chumchean et al. (2006) found that the density of the rain gauge network also plays an important role in the radar rainfall bias adjustment. They found that lowering the density of rain gauge observations in the KF process reduced accuracy of radar rainfall estimates. Additionally, the KF approach outperforms the use of MFB if rain gauge density is less than 1 per 90 km$^2$, and both KF and MFB produce identical performance when the rain gauge density is greater than 1 per 70 km$^2$.

In basins where a dense rainfall network is not available, Citizen Science (CS) offers a promising opportunity for enhancing the density of rainfall observations (Davids, et al., 2019). With the popularization of smartphones and the availability of (relatively) simple and cheap equipment, abundant mobile applications and projects have been initiated in Water Resources Management to measure hydrometeorological variables like rainfall, water level height or water quality, as well as to ground-truth remotely sensed information on e.g. land use (Srivastra et al., 2018; Davids et al., 2019; See, 2019; Seibert et al., 2019).

In the current study, we focus on rainfall measured by local citizens using a network of cheap rain gauges and a specially designed mobile application. Since citizen rainfall observations are typically provided at daily scale, a temporal downscaling technique is needed for sub-daily applications. There has been a variety of temporal rainfall downscaling methods developed since the 1970s. The simplest approach is to distribute daily rainfall data to sub-daily resolutions by assuming uniform distributions. Stochastically generating sub-period data or spatially transferring finer resolution rainfall from a nearby rain

gauge station to the study area based on spatial correlations are alternative approaches (Koutsoyiannis, 2003; Debele et al., 2007). However, these methods are usually not designed for real-time data disaggregation over large areas. Instead, a common approach for such scenarios is to downscale daily rainfall based on a simple fraction technique by considering the distribution patterns of high-resolution gridded rainfall products from radar or satellite sensors (Paulat et al., 2008; Wüest et al., 2010; Vormoor and Skaugen, 2013; Sideris et al., 2014; Barton et al., 2019). This study aimed to modify the KF logic by integrating

hourly rain gauge data with daily citizen rain gauge data that are downscaled to hourly time scale using a simple fraction method. The radar rainfall bias correction factor was sequentially updated using a two-step Kalman filter accounting for the contrasting quality of the hourly rain gauge data and downscaled citizen rain gauge data. The question we set out to answer is to what extent the downscaled citizen rainfall observations improve the accuracy of hourly radar rainfall estimates. Several scenarios of hourly rainfall distribution patterns were applied for downscaling to investigate the most suitable technique for

hourly radar rainfall assessment. Tubma basin located in Rayong province, eastern Thailand, was used as a case study area to test the approach.

## 2. Study Area and Data

### 2.1 Study Area

The study area is the Tubma basin located in Rayong province, eastern Thailand, situated between latitude 12.6789

to 12.8775, and longitude 101.0881 to 101.2975 (Fig. 1). It covers a catchment area of 197 km$^2$ with basin elevation ranging

from 4 to 416 m MSL. The main river, Klong Tubma, is 42 km in length and originates in Chom Hae, Kate, and Kra Bok mountains and flows downstream to the northwest before meeting the Gulf of Thailand at Pak Nam district. The Tubma watershed is susceptible to flooding, in particular Rayong. In Figure 1, we show the climatological variation across the study area and its surroundings, based on 30-year (1987-2017) annual mean rainfall from the network of 311 daily rain gauges owned by the TMD and situated within 200 km range from the Tubma basin. Spatial rainfall patterns were generated by inverse distance squared (IDS) between the gauge locations. The map shows that while there is a small gradient in mean annual rainfall (1,100 to 1,700 mm mean annual rainfall) across the area of Rayong and Chonburi provinces (within 90 km from the study area), changes are more pronounced when the distance exceeds beyond the 90 km boundary, especially to the east of the study area. This is because these areas are affected differently by the southwest monsoon. Consequently, evaluating the effectiveness of bias correction techniques was carried out within a 90 km radius from the study area with similar climatology.

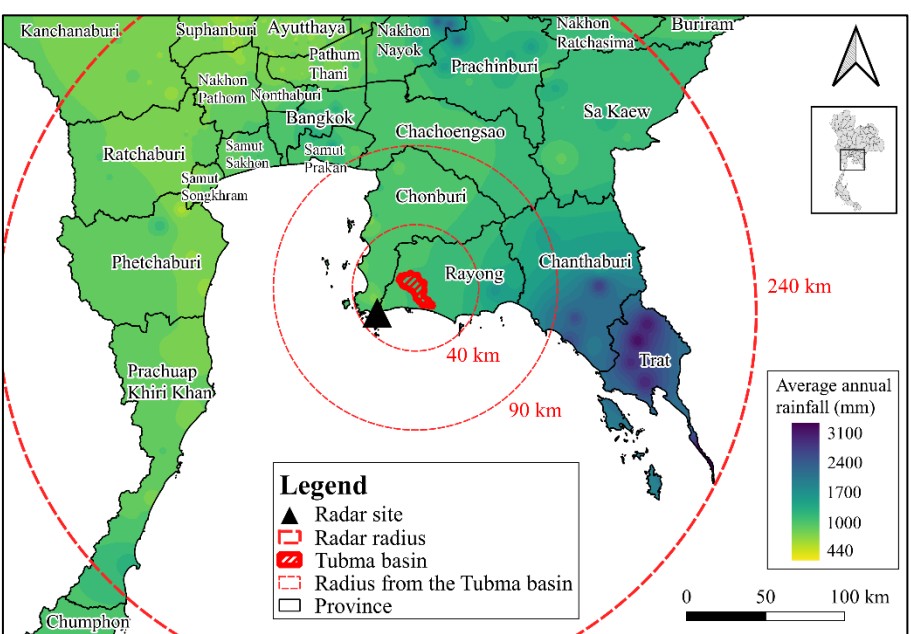

Figure 1: Climatological spatial rainfall distribution in and around the Tubma basin calculated from 30-year average annual rainfall data of 311 daily rain gauge network by using IDS method.

## 2.2 Radar Data

### 2.2.1 Reflectivity Data Collection

The Tubma basin is located within the coverage of the Sattahip radar station. The Sattahip radar, which belongs to the Department of Royal Rainmaking and Agricultural Aviation (DRRAA), is a S-band Doppler radar that transmits radiation with a frequency of 2.9 GHz and a half power beam width of 1.0°. The radar reflectivity product is provided in a Cartesian grid covering 240x240 km extent with 0.6x0.6 km spatial and 6-min temporal resolution. The Sattahip radar provides the reflectivity data derived from the 2.5-km constant altitude plan position indicator (CAPPI). The CAPPI reflectivity data originate from

below the climatological freezing level, so effects of measurement errors caused by bright band were considered negligible. The effects of ground clutter were removed from the reflectivity data by finding the clutter locations and discarding the radar measurements in these areas. Additionally, the noise power caused by various sources such as emission from space (cosmic noise), radiation from electrical sources near the radar antenna, and internally generated noise were eliminated by setting reflectivity values below 15 dBZ to zero (Doviak and Zrnic, 1984). While Fulton et al. (1998) suggested that measured

reflectivity that are greater than 53 dBZ are limited to 53 dBZ in radar rainfall estimation to mitigate false interpretation caused by hail. The hail cap can be seen as an adaptable threshold representing the maximum expected instantaneous rain rate which is unfortunately quite difficult to determine for a particular storm. Note that in tropical environments also slightly higher values have been reported as hail threshold. After data quality control, we separated the data into three datasets. The first dataset during May–October 2013 and May–September 2014 was used for the climatological Z-R calibration. The second dataset in

October 2014 were used for the Z-R verification, and the dataset for August–October 2019 was used in the bias correction processes.

### 2.2.2 The Z-R calibration and radar rainfall aggregation

The Z–R conversion error is a crucial source of error in radar rainfall estimates. The Z-R relationship was used to convert the measured reflectivity data (Z, $mm^6/m^3$) into rainfall rates ($R$, mm/h). The Z–R calibration and verification are

essential procedures to ascertain the parameters $A$ and $b$ in the relationship. Firstly, the instantaneous 6-minute radar reflectivity was converted to rainfall intensity using the climatological relationship $Z=200R^{1.6}$ proposed by Marshall and Palmer (1948). Secondly, the estimated 6-min initial instantaneous radar rainfall data were aggregated to hourly rainfall resolution using the accumulation algorithm proposed by Fabry et al. (1994). Thirdly, gauge rainfall was aggregated to hourly resolution. Finally, the optimal value of the parameter $A$ was established by minimizing the mean absolute error (MAE) between the gauge and

radar rainfall estimates, while the exponent $b$ was at 1.5 in our study. This is because radar rainfall estimates are relatively insensitive to $b$ with typical values between 1.2 and 1.8 (Battan 1973; Ulbrich 1983). The value of 1.5 was generally suitable to represent the exponent $b$ in the Z–R relation (Doelling et al., 1998; Steiner and Smith, 2000; Hagen and Yuter; 2003; Germann et al., 2006; Chantraket et al., 2016). The MAE is illustrated in Eq. (1).

$$MAE = \frac{1}{TN_G}\sum_{t=1}^{T}\sum_{i=1}^{N_G}|G_{i,t} - R_{i,t}| \qquad (1)$$

where $G_{i,t}$ is the gauge rainfall (mm/h) at gauge $i$ for hour $t$, $R_{i,t}$ is the hourly radar rainfall accumulation (mm) at the pixel corresponding to the $i^{th}$ rain gauge for hour $t$, $N_G$ is the total number of rain gauges, and $T$ is the total period used in the calculation. Results found that the optimal climatological Z-R relationship for the Sattahip radar used in this study is $Z=251R^{1.5}$. This relation is appropriate for both the calibration and validation datasets with the MAE of 1.36 mm and 1.47 mm, respectively.

## 2.3 Rain Gauge Data

### 2.3.1 Rainfall Data Collection

Data from the network of 297 continuous tipping-bucket gauge stations located within the Sattahip radar radius were collected (Fig. 2). These rain gauge data with a temporal resolution of 15 minutes are owned and operated by the Thai Meteorological Department (TMD). All continuous rain gauges used in this study have a resolution of 0.5 mm. The data quality screening was first carried out using double mass curves method of two adjacent rain gauges. To avoid no-rainfall events and systematical underreporting by the tipping-bucket rain gauge, hourly data above the tipping-bucket resolution of 0.5 mm were selected in the next step. Rain gauges with more than 80% of the recorded rainfall amounts below the 0.5 mm threshold at daily scale were excluded from the analysis. It turns out that many of these faulty gauges recorded zero rainfall throughout most of the study period. We found that rainfall data obtained from 134 rain gauges corresponding to the collected reflectivity datasets were used for the Z-R calibration and validation processes. For the bias adjustment computation, the selection of rain gauge networks with rainfall behavior similar to the study area is necessary. We selected 14 rain gauges of TMD in the region surrounding Tubma basin (Rayong and Chonburi provinces) based on spatial decorrelation analysis for this the process.

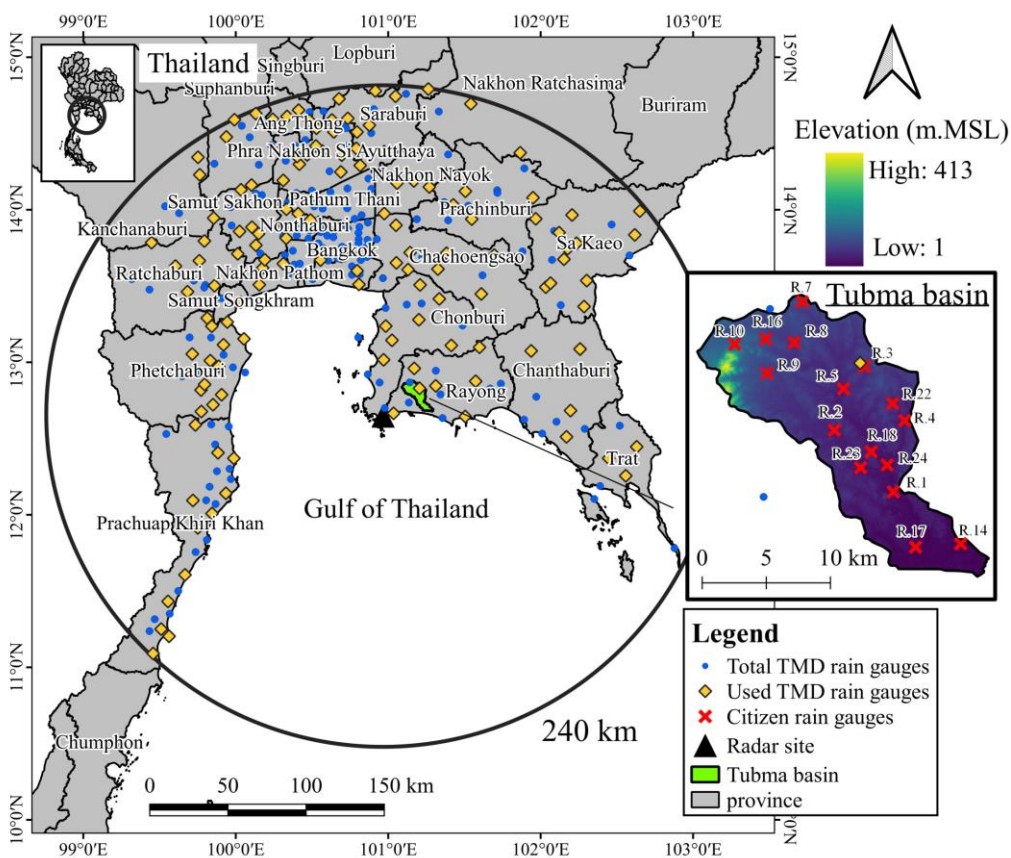

Figure 2: Location of study domain, showing Thai Meteorological Department (TMD) automatic rain gauges, citizen rain gauges, Sattahip radar, and Tubma basin.

2.3.2 Citizen Rain Observation

Out of the total TMD rain gauge network, only one rain gauge is located in the Tubma basin. To increase the density of the rain gauge network in the basin, low-cost citizen rain gauges were implemented in this study to capture the spatial heterogeneity of rainfall in the basin. Sixteen citizen rain gauges were installed (Fig. 2) with local residents taking daily measurements. The additional 16 citizen rain gauges with one station located at the same place as the existing TMD gauge

increased the density of rain gauges in the Tubma basin to 1 gauge/~12 km$^2$. The citizen observations were made by installing a cone-shape transparent plastic rain gauge in an open space area around a school, monastery, bridge or other building. This rain gauge is standardly used in South Africa (Fig. A1) with a diameter of 5 inches and a maximum capacity of 100 mm of rainfall. Mobile application developed by Mobile Water Management (MWM) (Mobile Water Management, 2020), the Netherlands, was used to record rainfall data for each rain gauge on a daily basis. The application has a an easily accessible

and user-friendly interface where participants simply fill in the observed rainfall amount, take a photo of the rain gauge and

upload this to the application. The photo and the rainfall data, together with the measuring location and time, are automatically stored in the database. Photos are used for visual validation of the recorded rainfall depth to eliminate errors.

In this study, participants were recruited amongst government officers, teachers, and local residents living close to the stations and were trained to take measurements at around 7 a.m. daily according to the TMD standards. Quality of the collected data was assured by the high photo resolution for double-checking the observations and strict requirements on measurement times to be consistent with the same standard of TMD for daily rainfall recording. Validation of the cone-shaped citizen gauges was conducted based on a citizen gauge co-located with an automatic TMD gauge located in the Tubma basin, during August – October 2019. The citizen gauge installed at the same location R.3 (Fig. A2) as a TMD gauge showed good similarity with an RMSE and Bias of 5.5 mm and 1.04, respectively.

Quality control consisted of screening all citizen rain gauge data for errors and inconsistencies using double mass curves. If citizen rain gauges reported >100 mm/day rainfall (maximum capacity of the citizen rain gauge), this data was excluded from the analysis. If days with no-rainfall data were found from all citizen rain gauges, the bias correction of that day was discarded from the dataset. By considering the data selection criteria, rainfall data recorded during August–October 2019 with rainy days, more than 80% of the whole period for the bias adjustment process was then used for further analysis.

## 3. Methods

The methodology for radar rainfall bias correction using tipping bucket and citizen gauges consists of the following steps. Firstly, daily citizen rain gauge data were downscaled to hourly time scale to be used as input for bias correction. The downscaling methods used in this paper are discussed in section 3.1. Next, an hourly radar bias correction model was developed combining rain gauge as well as downscaled citizen rain gauge data using a KF approach, as presented in section 3.2.

## 3.1 Downscaling daily to hourly rainfall

To downscale the daily citizen rain gauge data to hourly time-scale, information on the hourly storm distribution pattern is needed. Methodologies to obtain the temporal rainfall distribution patterns are outlined in Table 1.

Table 1: The four methods used in this study to downscale daily citizen rainfall amounts to hourly rainfall data.

| Distribution Code | Methodologies Description | Code Description |
|---|---|---|
| $R_P$ | Hourly rainfall patterns derived from radar rainfall time series of the radar pixel corresponding to citizen rain gauge location were used for downscaling. | The distribution patterns of radar rainfall at each radar pixels. |

| Distribution Code | Methodologies Description | Code Description |
|---|---|---|
| $R_{MP}$ | Hourly radar rainfall distributions of all radar pixels corresponding to citizen rain gauge locations were averaged to represent the mean temporal distribution pattern of radar rainfall. The $R_{MP}$ downscaling pattern was applied to all citizen rain gauges. | The mean distribution pattern of radar rainfall. |
| $G_{MP}$ | Hourly gauge rainfall patterns of all 14 gauges in the region surrounding Tubma basin were averaged to construct the mean hourly distribution pattern of regional rain gauge rainfall. The $G_{MP}$ was applied to all citizen rain gauges. | The mean distribution pattern of rain gauge rainfall. |
| $G_{Tubma}$ | The hourly rainfall pattern of the single rain gauge situated in the Tubma basin was used for correction of all citizen rain gauges in the basin. | The distribution pattern of the rain gauge in the Tubma basin. |

## 3.2 Hourly radar bias model

This section describes how radar bias is modelled (section 3.2.1), how observations are assimilated in the model to correct the bias (section 3.2.2), and how model parameters are estimated (section 3.2.3). Our approach extends a previously used Kalman filter radar bias model by including two different types of rainfall observations (data from traditional and from citizen rain gauges) and by using a likelihood-based method for parameter estimation.

*3.2.1 Kalman filter with two observations: model assumptions*

Mean field bias (MFB) adjustment is a common technique for bias correction in radar rainfall relative to ground stations. It can be computed as the ratio of mean hourly radar rainfall estimate and rain gauge measurement (Anagnostou and Krajewski, 1999; Yoo and Yoon, 2010; Hanchoowong et al., 2013; Shi et al., 2018). However, direct application of MFB does not account for uncertainty of the bias associated with each radar-gauge measurement. Alternatively, a KF has previously been used to estimate the spatially uniform MFB in real-time in several studies, including Ahnert et al. (1986), Smith and Krajewski (1991), Anagnostou et al. (1998), Seo et al. (1999), Chumchean et al. (2006), Kim and Yoo (2014), Shi et al. (2018). The KF

has the benefit of accounting for uncertainties in the observations by weighting the contribution of measurements by their respective error variances (Kalman, 1960). This is also the approach adopted here, but we extend it to include two types of observations with different error characteristics, i.e., hourly rain gauge data from TMD ($y_t$) and hourly downscaled citizen rain gauge data ($z_t$).

First, we define the logarithmic mean field radar rainfall bias $\beta_t$ at hour $t$ as (Smith and Krajewski, 1991; Anagostou et al., 1998):

$$\beta_t = log_{10}\left(\frac{\sum_{i=1}^{N_{G,t}} G_{i,t}}{\sum_{i=1}^{N_{G,t}} R_{i,t}}\right) \tag{2}$$

where $G_{i,t}$ and $R_{i,t}$ are as defined in Eq. 1. Following previous studies (e.g. Smith and Krajewski, 1991; Chumchean et al.,2006), we model the logarithmic mean field radar rainfall bias as a first-order autoregressive (AR1) process with stationary variance. Radar bias at time $t$ is expressed in terms of the bias at previous time $(\beta_{t-1})$ and the process noise $(W_t)$:

$$\beta_t = r_1\beta_{t-1} + W_t; \quad W_t \sim \mathcal{N}(0, \sigma_W^2) \tag{3}$$

$$\sigma_W^2 = (1 - r_1^2)\,\sigma_\beta^2 \tag{4}$$

where $r_1$ is lag-one correlation coefficient of the time-varying bias $\beta_t$, $W_t$ is a zero-mean random error with variance $\sigma_W^2$, and $\sigma_\beta^2$ is stationary variance of the process. We consider two types of observations, $y_t$ and $z_t$, of the unknown bias at time $t$, derived from the TMD and citizen rain gauges. Each observation is modelled as a random sample from a normal distribution conditioned on the underlying unknown bias with distinct measurement error variances $\sigma_{M_{y,t}}^2$ and $\sigma_{M_{z,t}}^2$:

$$y_t = \beta_t + M_{y,t}; \quad M_{y,t} \sim \mathcal{N}(0, \sigma_{M_{y,t}}^2) \tag{5}$$

$$z_t = \beta_t + M_{z,t}; \quad M_{z,t} \sim \mathcal{N}(0, \sigma_{M_{z,t}}^2) \tag{6}$$

where $\sigma_{M_{y,t}}^2$ and $\sigma_{M_{z,t}}^2$ are time-varying measurement error variances for the TMD and citizen rain gauges, respectively.

A factor graph representation of the radar bias and observation models is illustrated in Fig. 3 (a), with circles denoting random variables, and black squares denoting 'factors' or relations between variables in the model.

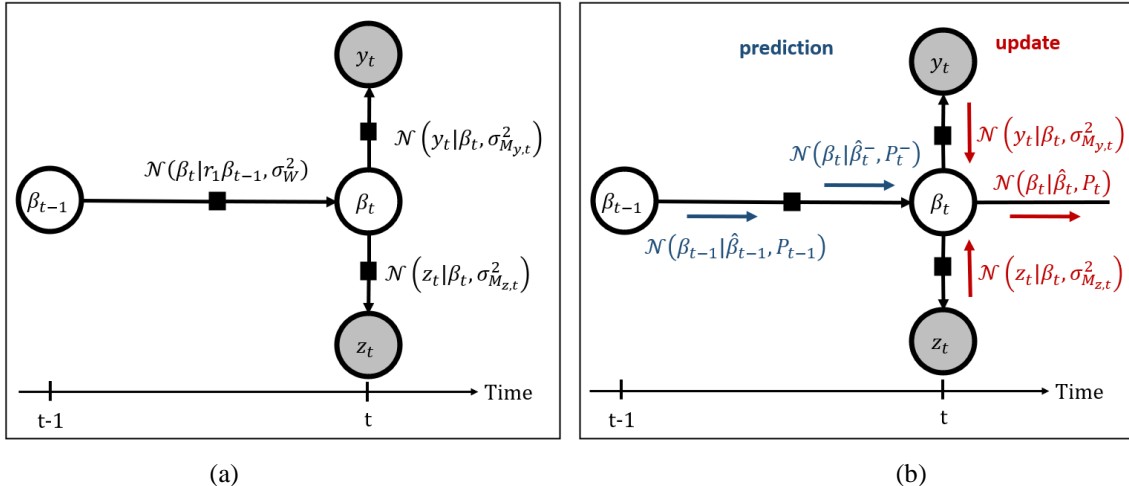

(a)                                                                (b)

Figure 3: On the left (a), a factor graph representation of the radar bias model: white circles depict random variables (bias at each time step), grey circles are rainfall bias observations ($y_t$ for TMD rainfall and $z_t$ for citizen rain gauge rainfall), and black squares are relations between variables (conditional normal distributions in this case). The right figure (b) depicts uncertainty propagation along the edges of the factor graph, from previous bias to current bias (Kalman prediction step in blue) and from the observations to current bias (two Kalman updates in red).

The model contains two parameters, i.e. $r_1$ and $\sigma_\beta^2$, that need to be specified, together with values for the measurement error variances, $\sigma_{M_{y,t}}^2$ and $\sigma_{M_{z,t}}^2$. Parameter estimation will be discussed in section 3.2.3. Section 3.2.2 first describes how observations are assimilated assuming values of the parameters and the measurement error variances have been specified.

### 3.2.2 Kalman filter with two observations: data assimilation

Having defined the model, we describe how observations are assimilated into the model, resulting in adjustments of the estimated bias. While the regular Kalman filter has two steps (prediction and update), assimilating two observations at each time step involves three steps, i.e. a prediction followed by two updates. Figure 3 (b) shows a visual depiction of the prediction and update steps as probabilistic information propagating along the edges of the factor graph.

1) Time update step (prediction)

The first step for each hour $t$ computes the prior mean $\hat{\beta}_t^-$ and variance $P_t^-$ of $\beta_t$ :

$$\hat{\beta}_t^- = r_1 \hat{\beta}_{t-1} \tag{7}$$
$$P_t^- = r_1^2 P_{t-1} + \sigma_W^2 \tag{8}$$

where $P_{t-1}$ is the posterior variance of $\beta_{t-1}$. For the first time step 0 ($t = 0$), we assume $\beta_0 = 0$ (climatological logarithmic mean field bias) and $P_0 = (1-r_1^2)\,\sigma_\beta^2$ (represents stationary process variance) (Smith and Krajewski, 1991; Chumchean et al., 2006).

2) The first measurement update step (1st correction)

The first update merges the bias prediction from step 1 with noisy observation $y_t$ (with measurement error variance $\sigma_{M_{y,t}}^2$), resulting in posterior mean $\hat{\beta}_{y,t}$ and variance $P_{y,t}$ of $\beta_t$ given by the following Kalman update equations:

$$K_{y,t} = P_t^- \left(P_t^- + \sigma_{M_{y,t}}^2\right)^{-1} \tag{9}$$

$$\hat{\beta}_{y,t} = \hat{\beta}_t^- + K_{y,t}(y_t - \hat{\beta}_t^-) \tag{10}$$

$$P_{y,t} = (1 - K_{y,t})P_t^- \tag{11}$$

where $K_{y,t}$ is the Kalman gain for assimilating observation $y_t$. If there is no observation available at time $t$, $K_{y,t}$ is zero (mathematically, a missing observation is equivalent to an observation with infinite variance $\sigma_{M_{y,t}}^2$ in Eq. 9), and the previous equations reduce to not performing any update, i.e. the posterior mean and variance are equal to the prior mean and variance.

3) The second measurement update step (2nd correction)

The second update is done using the posterior values from the 1st correction ($\hat{\beta}_{y,t}$ and $P_{y,t}$) as the prior values. The resulting Kalman gain and posterior mean and variance are given by:

$$K_{z,t} = P_{y,t}\left(P_{y,t} + \sigma_{M_{z,t}}^2\right)^{-1} \tag{12}$$

$$\hat{\beta}_{z,t} = \hat{\beta}_{y,t} + K_{z,t}(z_t - \hat{\beta}_{y,t}) \tag{13}$$

$$P_{z,t} = (1 - K_{z,t})P_{y,t} \tag{14}$$

If there is no observation available at time $t$, Kalman gain $K_{z,t}$ is zero and these equations result in no update being applied, i.e.
posterior mean and variance are the same as after the first update.

Since the logarithmic mean field radar rainfall bias $\beta_t$ is assumed to be normally distributed, the mean field bias ($B_t$) is lognormally distributed with posterior mean at time $t$ obtained from the posterior mean and variance of $\beta_t$ (Smith and Krajewski, 1991):

$$\hat{B}_t = 10^{(\hat{\beta}_{z,t}+0.5P_{z,t})} \tag{15}$$

The overall procedure of sequentially assimilating and downscaling the data is referred to as CKF (Kalman Filter combined with the citizen rain gauge data), and is summarized by the flowchart in Figure 4. Operationally it is implemented as follows:

1) Since the citizen rain gauge data were received at the last hour of day $i$, before receiving the downscaled hourly citizen rain gauge data of day $i$, the ordinary KF and observed hourly data of TMD were used to predict and correct the hourly bias adjustment factor of the day $i$.

2) If the citizen rain gauge data was available at the end of the day $i$, the citizen rain gauge data were downscaled to hourly time-scale, as explained in section 3.1.

3) The TMD and downscaled citizen rain gauge data were together used to conduct two steps of measurement update in the CKF process for all hourly time-steps of day $i$.

4) Bias adjustment factors were applied every hourly time step to obtain the final product of hourly radar rainfall estimation of day $i$. The bias factor for the last hour of the day $i$ was used afterward as the initial value for calculating the Ordinary KF of day $i$+1.

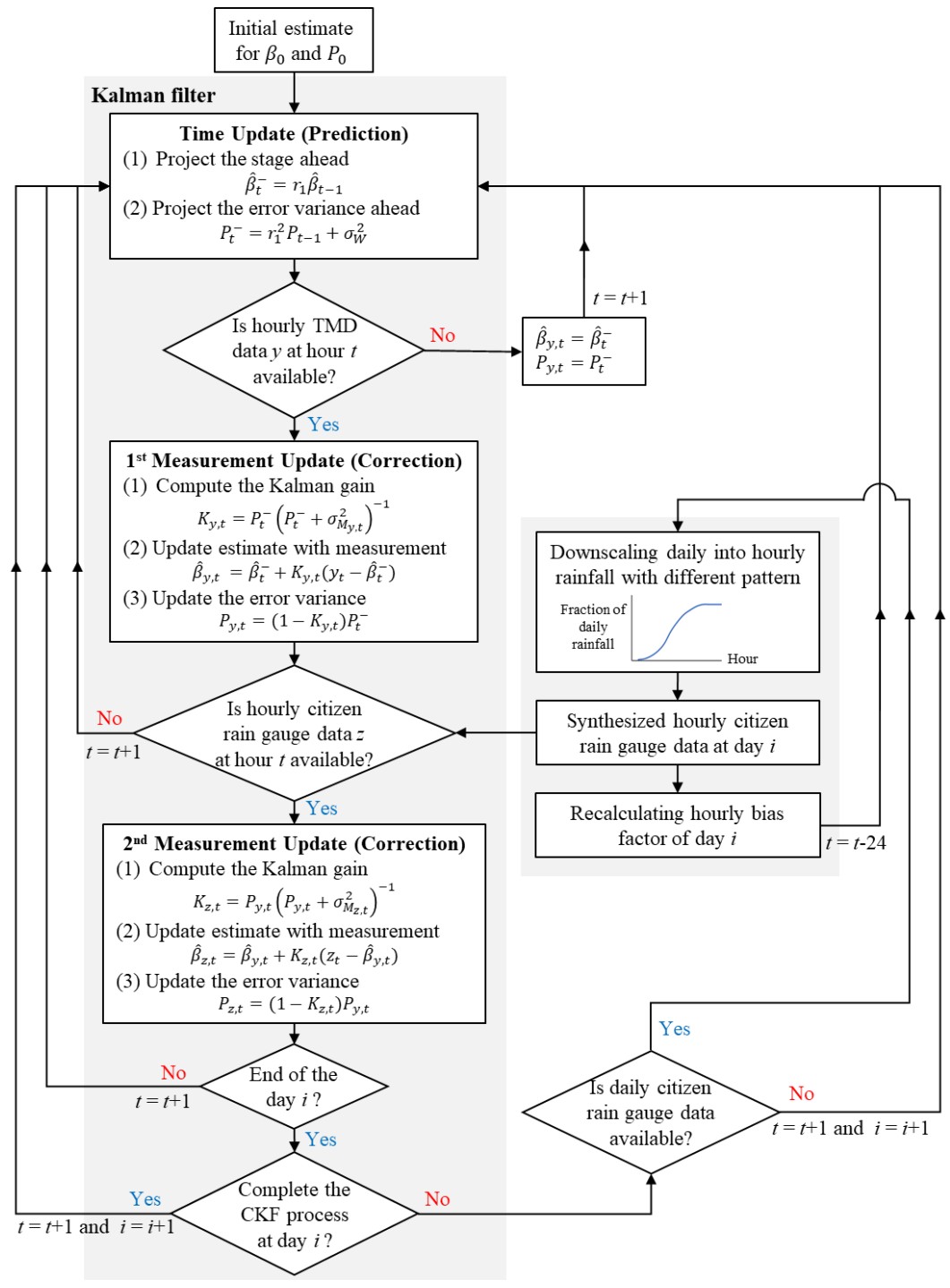

Figure 4: A diagram of the procedure of Kalman filter combined with the citizen rain gauge data (CKF)

 *3.2.3 Kalman filter with two observations: parameter estimation*

The model requires specification of the two parameters, $r_1$ and $\sigma_\beta^2$, and the measurement error variances, $\sigma_{M_{y,t}}^2$ and $\sigma_{M_{z,t}}^2$. We estimate the latter using formulas for the variance of the mean bias across individual TMD and citizen rain gauges:

$$\sigma_{M_{y,t}}^2 = \frac{\sigma_{\bar{y},t}^2}{n_{y,t}} \tag{16}$$

$$\sigma_{M_{z,t}}^2 = \frac{\sigma_{\bar{z}_t}^2}{n_{z,t}} \tag{17}$$

where $\sigma_{\bar{y},t}^2$ and $\sigma_{\bar{z},t}^2$ are variances that quantify spatial variability of radar bias at time $t$ at TMD and citizen rain gauge locations, respectively, and $n_{y,t}$ and $n_{z,t}$ are the corresponding number of observations at hour $t$ from the two networks.

Once measurement error variances are specified, values for parameters $r_1$ and $\sigma_\beta^2$ are obtained by maximizing the marginal likelihood function (Bock et al., 1981; Harvey, 1990; Proietti et al., 2013; Pulido et al., 2018). As mentioned earlier, we have two sources of observed log mean field bias at hour $t$, from TMD ($y_t$) and citizen rain gauge ($z_t$). When only TMD data are available, the marginal likelihood is computed as:

$$p(D) = \prod_{t=0}^{T} \int p(y_t|\beta_t)p(\beta_t|y_{t-1},\dots,y_0)d\beta_t = \prod_{t=0}^{T} \mathcal{N}(y_t; \hat{\beta}_t^-, \sigma_{y_t}^2 + P_t^-) \tag{18}$$

where $D$ is the data vector that contains all observed values, $\beta_t$ is the true hidden state of log mean field bias at hour $t$, and $T$ is the number of hourly time steps. When both TMD and citizen rain gauge data are available, the marginal likelihood is obtained as:

$$p(D) = \prod_{t=0}^{T} \int p(y_t|\beta_t)p(z_t|\beta_t)p(\beta_t|y_{t-1},z_{t-1},\dots,y_0,z_0)d\beta_t$$

$$= \prod_{t=0}^{T} \mathcal{N}(y_t; z_t, \sigma_{M_{y,t}}^2 + \sigma_{M_{z,t}}^2)\mathcal{N}(\tau^{-1}(\frac{y_t}{\sigma_{M_{y,t}}^2} + \frac{z_t}{\sigma_{M_{z,t}}^2}); \hat{\beta}_t^-, \tau^{-1} + P_t^-) \tag{19}$$

where $\tau = \frac{1}{\sigma_{M_{y,t}}^2} + \frac{1}{\sigma_{M_{z,t}}^2}$ and prior mean $\hat{\beta}_t^-$ and variance $P_t^-$ are from the prediction step in section 3.2.2.

To obtain optimal values of the two parameters $r_1$ and $\sigma_\beta^2$ that maximize the (logarithm of the) marginal likelihood, the Nelder-Mead Simplex was used, which is an algorithm for searching a local optimum of a function (Lagarias et al., 1998; Luersen et al., 2004; Gao et al., 2012).

### 3.3 Verification of the proposed bias correction approaches

To investigate which bias adjustment technique (MFB, KF, and CKF) gives the most suitable radar rainfall estimates for the Tubma basin, the adjusted radar rainfall estimates were validated against measured rainfall data. There was only one automatic TMD rain gauge available in the basin, which was insufficient for validation purposes. Consequently, for testing the performance of hourly rainfall bias correction, data from 13 TMD stations located within a 90 km radius from the center of the Tubma basin were used, together with one TMD station in the basin. Furthermore, daily time-scale validation was conducted, using the daily rainfall data from 16 citizen rain gauges located in the Tubma basin. Leave-one-out cross-validation (LOOCV) algorithm was implemented to avoid bias in selecting the validation rain gauges. For each round of cross-validation, one rain gauge was left out for validation and the remaining rain gauges were used as the calibration rain gauges to calculate the bias adjustment factor using the three different techniques. This was repeated for all combinations and then the error of radar rainfall estimates after correcting with the estimated bias factor at each radar pixel corresponding to the held-out gauge was computed for all trials. In this study, Root Mean Square Error (RMSE) and Mean Bias Error (MBE) were applied as statistical measures to evaluate the effectiveness of the different bias correction methods at each validation rain gauge. The number of possible combinations is equal to the total number of validated gauges ($N_G$). Data for the period August-October 2019 were used in the evaluation. Four scenarios combining the three bias adjustment techniques were evaluated, summarized in Table 2.

Table 2: Simulation cases for evaluating the effectiveness of bias correction techniques.

| Evaluation Case | Tested approaches | Number of rain gauges used for different purposes | | | | Temporal and spatial scale of rainfall for validation |
|---|---|---|---|---|---|---|
| | | rain gauges datasets | gauges for calculating MFB and KF | gauges for combining with KF | validation gauges ($N_G$) | |
| KF-TMD-H | MFB and KF | 14 TMD | 13 TMD | - | 14 TMD | Hourly, Tubma plus 90 km radius |
| KF-TMD-D | MFB and KF | 14 TMD and 16 citizen rain gauges | 14 TMD | - | 16 citizen rain gauges | Daily, Tubma basin |
| CKF-D | MFB, KF and CKF | 14 TMD and 16 citizen rain gauges | 14 TMD | 15 citizen rain gauges | 16 citizen rain gauges | Daily, Tubma basin |
| CKF-H* | MFB, KF and CKF | 14 TMD and 16 citizen rain gauges | 13 TMD | 16 citizen rain gauges | 14 TMD | Hourly, Tubma plus 90 km radius |

*CKF-H includes four scenarios for four different hourly downscaling patterns for the citizen rain gauges, according to Table 1.

KF-TMD-H: One TMD rain gauge from the total of 14 gauges was left out for validation and the remaining 13 gauges were separated for calculating the bias adjustment factors using MFB and KF. This process was iterated 10 times until all 14 TMD rain gauges were left out for cross-validation. Aggregated hourly rainfall between the adjusted radar and gauge rainfall data were compared to obtain the RMSE and MBE.

KF-TMD-D: To identify whether MFB or KF is more accurate for daily rainfall simulation in the Tubma basin if there are only 14 TMD rain gauges available, 14 TMD and 16 citizen rain gauges were used for the analysis. All TMD gauges were used for assessing MFB and KF, and estimated bias factors were applied at the daily time-scale. Assessment of RMSE and MBE of daily rainfall at all 16 citizen rain stations was used for validation.

CKF-D: To evaluate the added value of using citizen rain gauges in the basin for bias correction, 15 citizen rain gauges (leave one citizen rain gauge out for validation) were used in addition to the TMD gauges following the CKF procedure explained in 3.2.2. Estimation of daily RMSE and MBE was carried out at the held-out citizen rain gauge.

CKF-H*: To test whether the CKF with the most suitable storm pattern could benefit radar rainfall estimates in the area further away from the Tubma basin, 14 TMD gauges were used to generate four cases of hourly rainfall distribution patterns as described in Table 1 for downscaling the selected 16 daily citizen rain gauge data to hourly time scale. The synthesized hourly citizen rain gauge data were later used as a second observation for the 2nd correction of the CKF. Thirteen TMD gauges (leave one TMD out) were used to produce estimates with MFB and KF, and all 16 citizen rain gauges were merged for CKF computation.

All bias adjustment techniques evaluated the effectiveness at the held-out gauge for all possible combinations of the LOOCV procedure.

## 4. Results and discussion

### 4.1 Simulation of bias adjustment factor

#### 4.1.1 Parameter estimation for the KF and CKF

Five scenarios were investigated for radar bias correction using the KF, based on TMD and citizen rain gauge observations, including four scenarios comparing different hourly downscaling approaches for the citizen rain gauge data (Table 1). Parameter estimates of the KF are shown in Table 3. These results indicate that the parameter $r_1$, the lag-one correlation coefficient of the logarithmic mean field bias, ranges from 0.15 to 0.53, depending on the hourly downscaling approach. While $\sigma_\beta^2$ representing the stationary variance of the logarithmic mean field bias remains relatively invariant (ranging from 0.24-0.28) over the same period of simulation.

Table 3: The parameters of the KF estimated from different datasets of observation gauge rainfall. KF-TMD is using only TMD hourly rain gauge observations, CKF is using TMD and citizen rain gauge observations, where $R_P$, $R_{MP}$, $G_{MP}$ and $G_{Tubma}$ represent different strategies for hourly downscaling of the citizen rain gauge observations (Table 1)

| Type of observation | The KF's parameters | |
| gauge rainfall | $r_1$ | $\sigma_\beta^2$ |
| --- | --- | --- |
| KF-TMD | 0.29 | 0.24 |
| CKF-$R_P$ | 0.53 | 0.28 |
| CKF-$R_{MP}$ | 0.33 | 0.24 |
| CKF-$G_{MP}$ | 0.15 | 0.24 |
| CKF-$G_{Tubma}$ | 0.38 | 0.25 |

### 4.1.2 Bias adjustment factor comparison

To test the performance of the bias adjustment techniques among KF-TMD, CKF-$R_P$, CKF-$R_{MP}$, CKF-$G_{MP}$, and CKF-$G_{Tubma}$, all approaches were used to assess the mean field bias for each hour using the data period August – October 2019. The results were compared to the MFB calculated using the 14 TMD rain gauges (MFB-TMD) in the Tubma basin and 90 km
radius surroundings. Results summarized in Fig. 5 show that:

- The daily observed bias is somewhat higher and shows larger variability for the citizen gauges compared to TMD gauges. The hourly observed bias based on downscaled citizen gauge data are in the same range as hourly bias based on TMD gauges, with somewhat higher median values and spread (25-75 percentile range) for the $R_P$ and $G_{Tubma}$ downscaling scenarios.

- Hourly observation error variance is smallest for the CKF-$R_P$ downscaling approach and somewhat larger for the
other CKF approaches compared to observation error variance for the TMD gauges.

- Estimated hourly bias values based on KF-TMD show a slightly higher mean and smaller variability range compared to observations. The bias produced by the KF-TMD is close to the MFB-TMD if the observation error variance is small. In case that no measured data is available for the bias update, the computed bias factor ($B_t$) progressively converged to 1.3, to meet the climatological logarithmic mean field bias.

- Estimated bias values based on the CKF approaches are able to reproduce bias variability as observed by TMD gauges, with median values deviating by 0.2 to 0.4 and value range slightly larger for CKF-$R_P$ and smaller for CKF-$R_{MP}$.

- CKF gives different bias values according to the storm distribution pattern (see Appendix B) and the availability of the daily citizen rain gauge data used in combination with the KF. In case that no citizen rain gauge data is available for updating, the bias generated by the CKF for every combination is close to the ordinary KF with small differences depending on their respective $r_1$ and $\sigma_\beta^2$ parameters.

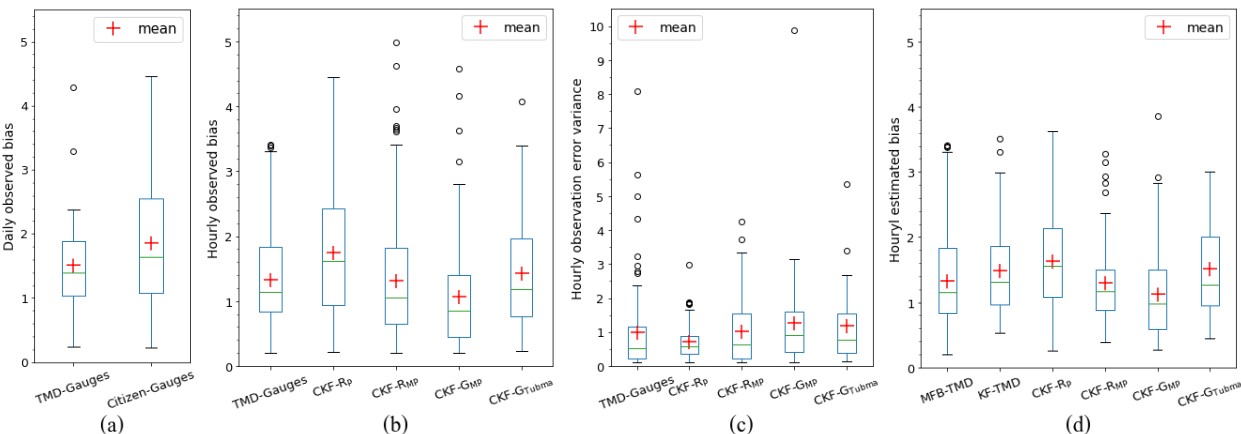

Figure 5: Comparison of (a) daily observed mean field bias based on TMD rain gauges in the region and the citizen rain gauges in the Tubma basin, (b) hourly observed mean field bias based on TMD rain gauge observations and downscaled citizen rain gauge observations, (c) hourly observation error variances and (d) hourly estimated mean field bias obtained based on MFB and the five different KF approaches. Bias calculations cover 16 citizen gauges in the Tubma basin and 14 TMD gauges within 90 km radius from the Tubma basin. Hourly scale calculations for the citizen gauges (CKF) are based on four different sub-daily interpolation scenarios ($R_P$, $R_{MP}$, $G_{MP}$ and $G_{Tubma}$, Table 3).

## 4.2 Effectiveness evaluation of bias correction approaches

*4.2.1 Hourly rainfall validation for the larger region (90 km radius) surrounding Tubma basin using MFB and KF approaches (Case 1)*

Figure 6 (a) shows cross-validation results based on RMSE and MBE between TMD rain gauges and adjusted radar rainfall using MFB and KF for hourly bias adjustment. Bias adjustment reduces RMSE and especially MBE, with KF-TMD performing somewhat better than MFB-TMD especially in terms of RMSE. This confirms the ability of the KF approach that considers the error variance of observed hourly data as the weight for correcting the predicted mean bias instead of using only the calculated mean field bias (Smith and Krajewski, 1991; Chumchean et al., 2006).

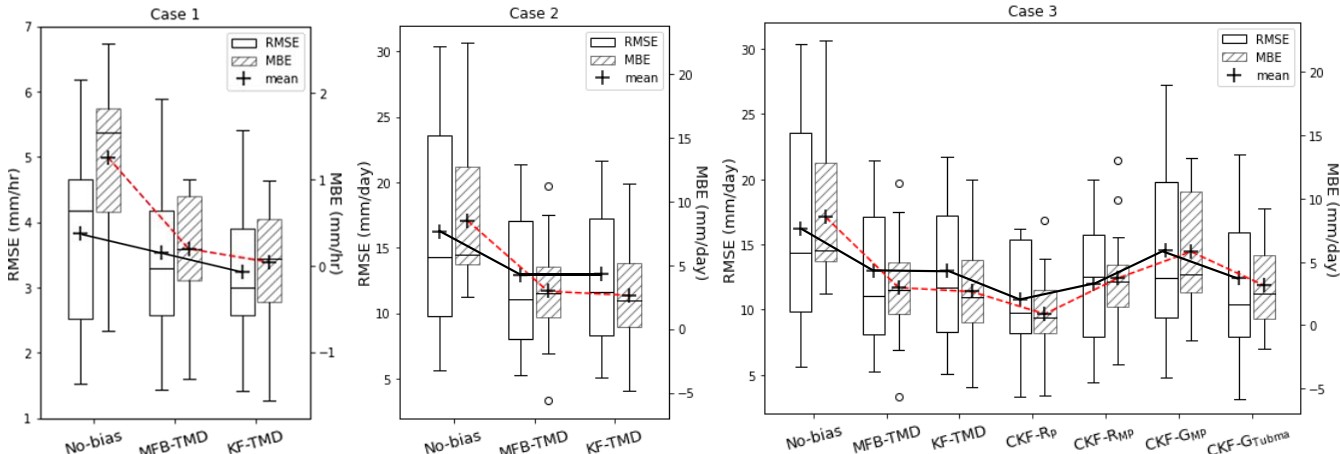

Figure 6: Variation in RMSE and MBE across the cross-validation scenarios for the various evaluation cases: (a) case 1, hourly bias updating based on MFB and KF using TMD gauges (b) case 2, daily bias updating based on MFB and KF using only TMD gauges and (c) case 3, daily bias updating using MFB, KF (TMD gauges) and CKF (TMD and citizen gauges). Validation covers 16 gauges in the Tubma basin for daily scale and 14 gauges within 90 km radius from the Tubma basin for hourly scale.

*4.2.2 Daily rainfall validation in the Tubma using MFB and KF approaches, citizen gauges for validation (Case 2)*

Results associated with validating the bias correction performance within the Tubma basin are presented in Fig. 6 (b). This shows bias correction performance within the Tubma basin, for MFB and KF-based daily bias adjustment. The two approaches show similar performance at the daily scale and improve RMSE by 20-30% and MBE by 50-60% (for median and upper 75percentile, respectively). The added value of a KF-based approach is limited for this case, since 14 TMD rain gauges in the region were used to compute observation variance which cannot represent the mean field bias behaviour in the Tubma basin.

*4.2.3 Daily rainfall validation in the Tubma using CKF approaches (Case 3)*

Figure 6 (c) shows cross-validation results at daily scale for the Tubma basin, comparing bias correction approaches using TMD only and TMD combined with citizen gauges. Following the CKF steps, citizen rain gauge data are downscaled to hourly time scale using four different approaches, resulting in variation in hourly observed bias and error variances as shown in Fig. 5 (b) and (c), respectively. Cross-validation results after accumulation to daily scale show that CKF-$R_P$ outperforms the other approaches (CKF-$R_{MP}$, CKF-$G_{Tubma}$, MFB-TMD, KF-TMD, and CKF-$G_{MP}$) in terms of both RMSE and MBE. The performance of CKF techniques for radar rainfall simulation in the Tubma basin relates to the reliability of the downscaled hourly observations. This is reflected in the variation of the estimated observation error variances for CKF-$R_P$ as shown in Fig. 5 (b) and (c). The better performance of CKF-$R_P$ is explained by the smallest range in observation error variance, indicative

of better consistency observation bias. Comparison with No-bias, CKF-R$_P$ can improve RMSE by 32-25 % and MBE by 90-80 % for median and upper 75 percentile, respectively. While CKF-G$_{MP}$ exhibits the worst performance compared with the other CKF approaches with the improvement of RMSE by 13-16 % and MBE by 57-56 %, respectively. This apparently decrease in efficiency of the CKF can confirm by the highest median value of the estimated observation error variances of CKF-G$_{MP}$ (see Fig. 5 (c)) with 33% higher than that of CKF-R$_P$.

*4.2.4 Hourly rainfall validation using MFB, KF, and CKF approaches (Case 4)*

Results for this section are presented in Figs. 7 and 8. The performance for radar rainfall estimates using CKF-RP with different distances from the Tubma basin is reported in Appendix C. Cross-validation results at hourly time-scale show a strong improvement achieved by bias adjustment using citizen gauges, in particular close to the Tubma basin where the citizen gauges are located. Figure 7 (b) and (c) show validation results based on TMD gauges for gauges close to (0-40 km radius) and further away (40-90 km radius) from the center of Tubma basin (see Fig. 7 (a)), both ranges cover the similar number of TMB gauges). Figure 7 (b) and (c) show that CKF-R$_P$ bias adjustment significantly improves radar rainfall estimates at hourly time scale, compared to bias adjustment approaches based on TMD gauges only in the 0-40 km range closest to Tubma basin. While there is a modest improvement in mean RMSE (see the black line connecting the mean values of the box plots from MFB-TMD to CKF-R$_P$), the upper 75 percentile RMSE is reduced from about 6 mm/h to 3.5 mm/h. Mean MBE is changed from 0.1 to -0.15 mm/h (see the red-dotted line connecting the mean values from MFB-TMD to CKF-R$_P$). For the 40-90 km range, CKF-R$_P$ performs similarly to MFB-TMD and KF-TMD.

Figure 8 illustrates that hourly rainfall distribution patterns of TMD rain gauges in the 0-40 km range, influenced mainly by the southwest monsoon, appear to be more similar to the mean citizen rain gauge data than the range beyond 40 km. Consequently, the application of CKF-R$_P$ based on combining citizen rain gauge network to TMD rain gauge network with similarity of rainfall characteristic is a key for improving radar rainfall estimates.

Results in Fig. 7 also show that the MBE values in the 0-40 km range are explicitly lower than that in the 40-90 km range. Apparently, at shorter range, positive and negative errors represented in MBE cancel out more frequently than they do for the gauges at larger distance. In other words, gauges more or less randomly over or underestimate rainfall values as we can see similar rainfall distribution patterns among all gauges with high variation of rainfall amount during the storm period in Figure 8 (b). Conversely, in the 40-90 km range, bias correction at gauge locations consistently leads to over- or underestimation of rainfall. This can be explained by gauge at larger distance being affected by different rainfall generation patterns, associated with their location closer to the coast or mountains (see Fig. 7 (a)). The influence of the southwest monsoon strongly affects all gauges located in the coastal region on the windward side of a mountain, while rain gauge locations on the leeward side have less rainfall amount. Figure 8 (c) shows that TMD gauges located on the leeward side (e.g., 4590009 and 4590011) obviously appear a steady gradient of the mass curve reflecting light rainfall accumulation, whereas the gauges on the windward side (e.g., 4590002 and 4590003) show the mass curves with a sharper gradient. Further details on the sensitivity bias adjustment techniques accuracy to rainfall characteristics are provided in Appendix D.

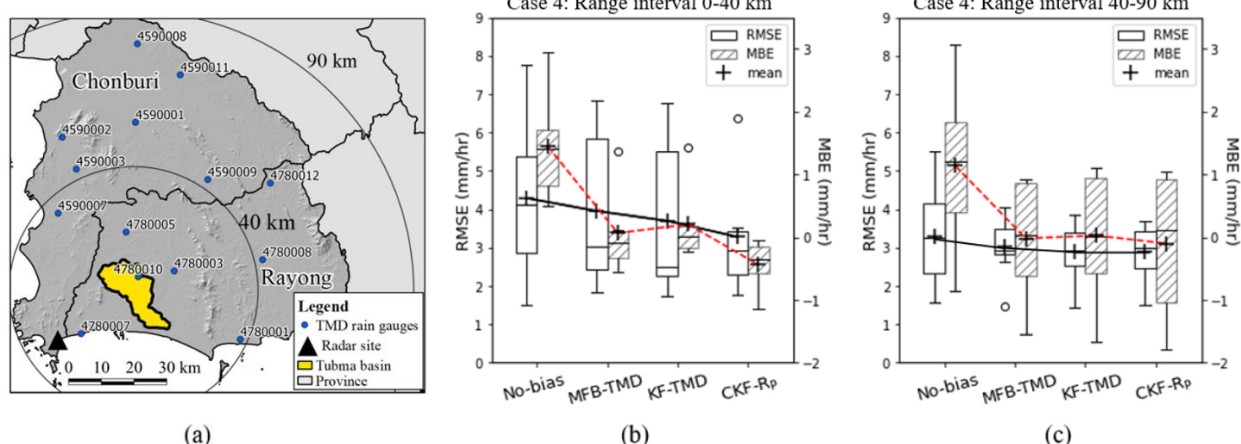

Figure 7: Comparison of the RMSE and MBE for different range interval from the centroid of the Tubma (a) Rain gauge locations at each range interval (b) the comparisons for the range 0-40 km (c) the comparisons for the range 40-90 km. For CKF, only results for the CKF-$R_P$ approach are shown, based on its better performance at daily time-scale (shown in Figure 6c).

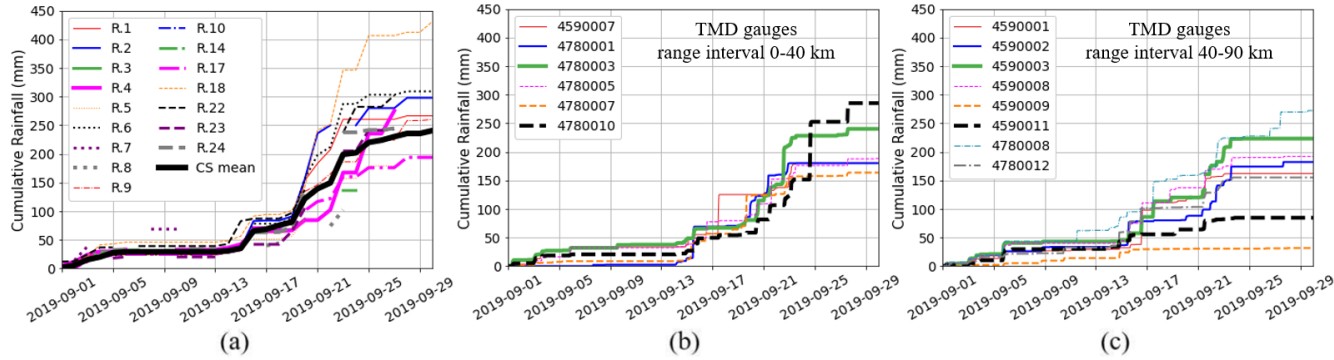

Figure 8: Comparison of mass curves of hourly rainfall among various rain-gauge locations (a) the citizen rain gauges located in the Tubma (b) TMD rain gauges within 0-40 km radius from the Tubma basin (c) TMD rain gauges within 40-90 km radius from the Tubma basin.

## 5. Conclusion

In this study we introduced a modified KF approach in radar bias correction in the Tubma basin, eastern Thailand. The two-step KF integrates daily data from a dense citizen rain gauge network with hourly data from a much sparser network of conventional, yet higher quality tipping bucket rain gauges. Daily citizen rain gauge observations were downscaled to hourly time scale using four different approaches. The question we aimed to answer is to what extent the downscaled citizen rainfall

observations improve the accuracy of hourly radar rainfall estimates. Results showed that citizen rain gauges significantly improve the performance of radar rainfall bias adjustment, up to a range of about 40 km from the centre of the Tubma basin (197 km$^2$) where the citizen rain gauge network is located. While a modest improvement in mean RMSE was obtained, the

upper 75 percentile RMSE was reduced from 6 mm/h to 3.5 mm/h. The mean bias error was changed from 0.1 to -0.15 mm/h across the validation period (August–October 2019). In the Tubma basin, beyond the 40 km range, no significant improvement by inclusion of the citizen gauges was found. The rainfall distribution pattern is key for downscaling the daily measured citizen rain gauge observations into hourly temporal resolution. We found that in the Tubma basin downscaling based on the rainfall patterns derived from hourly radar rainfall at overlying radar pixels corresponding to the citizen gauge location was the most

suitable technique, resulting in the smallest variation of observation error variances of the mean field bias. In the case of a sparse rain gauge network, the mean field bias and the Kalman filter approach both show improvement, and the degree of improvement was similar between the two approaches. In other words, in a sparse gauge network, the added value of error information represented in the Kalman filter is limited. Note that citizen rain gauge data are available only at the end of the day and consequently, the modified two-step Kalman filter as used in this study has restrictions in real-time applications.

However, the here proposed method has clear potential when creating high quality historical radar-rainfall time series for climatological studies and in post-event analysis. Moreover, near real-time assessment could be achievable if the citizen rain gauge data are collected at sub-daily time scale.

**Appendix**

**Appendix A: Citizen rain observation and validation**

An example of installing a cone-shape transparent plastic citizen rain gauge which is standardly used in South Africa at location R.22, Map Tong school, Rayong province is illustrated in Fig. A1. While validation of the cone-shaped citizen gauges was done at location R.3 by comparing the measured daily citizen gauge data with an automatic TMD gauge at same location as presented the result in Fig. A2. Root Mean Square Error (RMSE) and Bias were applied as statistical measures for the validation as shown in Eq. A1 and Eq. A2, respectively.

$$RMSE = \sqrt{\frac{1}{T}\sum_{t=1}^{T}(G_{TMD,t} - G_{CR,t})^2} \tag{A1}$$

$$Bias = \frac{\sum_{t=1}^{T} G_{TMD,t}}{\sum_{t=1}^{T} G_{CR,t}} \tag{A2}$$


where $G_{CR,t}$ is daily citizen gauge rainfall at location R.3 for day $t$, $G_{TMD,t}$ is daily TMD gauge rainfall at the same location of R.3 for day $t$, *and T* is the total period used in the calculation.

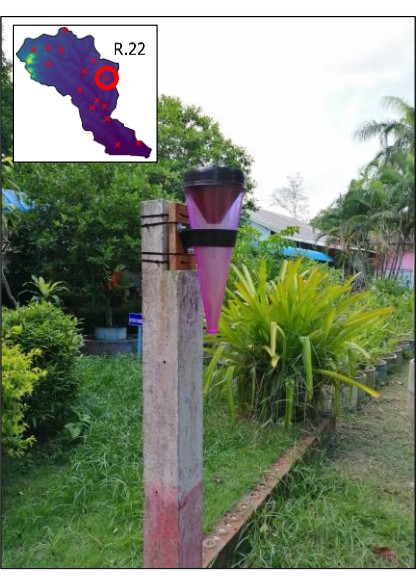


Figure A1: An example of installing a cone-shape transparent plastic citizen rain gauge at location R.22, Map Tong school.

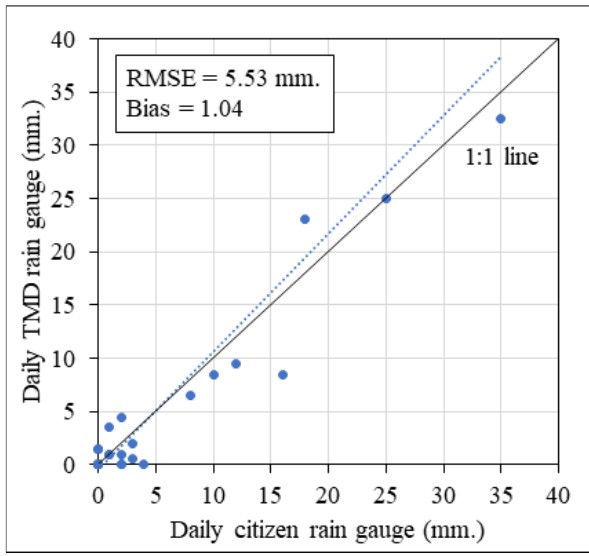


Figure A2: Daily rainfall depth comparison between the TMD and citizen rain gauge at location R.3 during August – October 2019.

**Appendix B: Hourly rainfall distribution patterns**

Four hourly rainfall distribution patterns were obtained as outlined in Table 1. Figure B1 illustrates the cumulative fraction of daily rainfall at hourly scale during the simulation period August-October 2019. It can be seen that most rainfall was concentrated in the afternoon hours, with very little rainfall falling at night. $R_P$ and $R_{MP}$ showed relatively more rainfall concentrated afternoon rainfall, while $R_P$ and $G_{MP}$ showed larger variability in the downscaled hourly data with substantial
outliers in the box plots, associated with variability in the locations underlying the rainfall distributions (multiple radar pixels within the Tubma basin for $R_P$ versus multi TMD gauges surrounding the Tubma basin for $G_{MP}$). $G_{MP}$ showed the flattest distribution with the longest rainy period of around 11 hours compared to the others having a period of heavy rainfall around 4-5 hours a day. This is explained by the larger spatial variability in the gauges covered by $G_{MP}$.

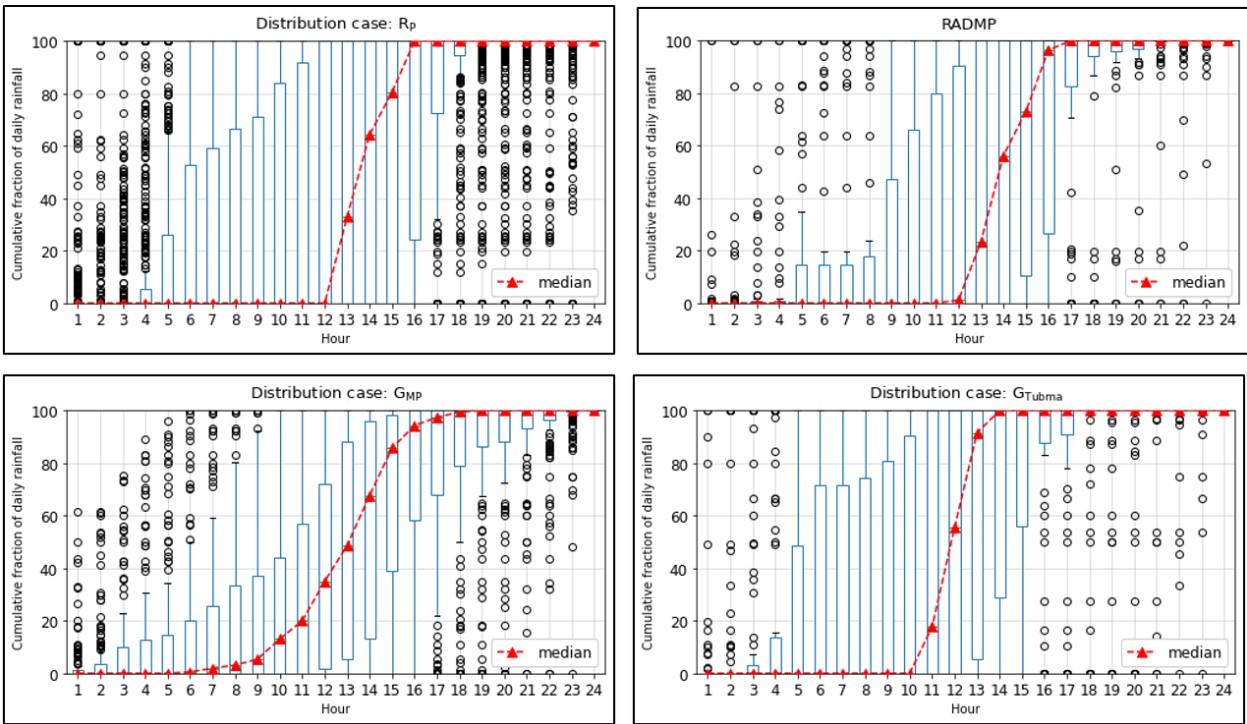


Figure B1: Variation of fraction of 24-hour rainfall for each rainfall distribution scenario.

**Appendix C: The performance in radar rainfall estimates of using CKF-RP with different distances from the Tubma basin.**

We chose the 40 km separation boundary to achieve an equal number of gauges in the "near" and "far" groups. Following the referee's comment, we investigated RMSE between gauge rainfall and radar rainfall without bias adjustment ($RMSE_{No-Bias}$) and with the CKF-R$_P$ ($RMSE_{CKF-RP}$) for individual stations (located at distances 5 – 80 km from the Tubma basin). We computed the percentage improvement in radar rainfall estimates using CKF-R$_P$ compared to No-Bias at each rain gauge, indicating the relative errors changing with distance from the Tubma basin (Fig. C1). Figure C1 shows that the improvement percentage of using CKF-R$_P$ tends to reduce with increasing distance from the Tubma basin, where the citizen gauges are located. The percentage reduction gradually decreases beyond a distance of about 40 km.

Note that the gauge 4780010 situated at the nearest range is expected to provide the best improvement, however, the Sattahip radar temporarily stopped measuring for 3 hours (during 16:00 h - 19:00 h of 24 September 2019) which is associated with a heavy storm's center was only at the gauge 4780010. This leads to significant degradation of the radar rainfall performance. Furthermore, the lower percentage improved for station 4780005 is associated with a localized heavy rainfall event that was recorded only at this location, negatively affecting its performance as a representative station for bias correction.

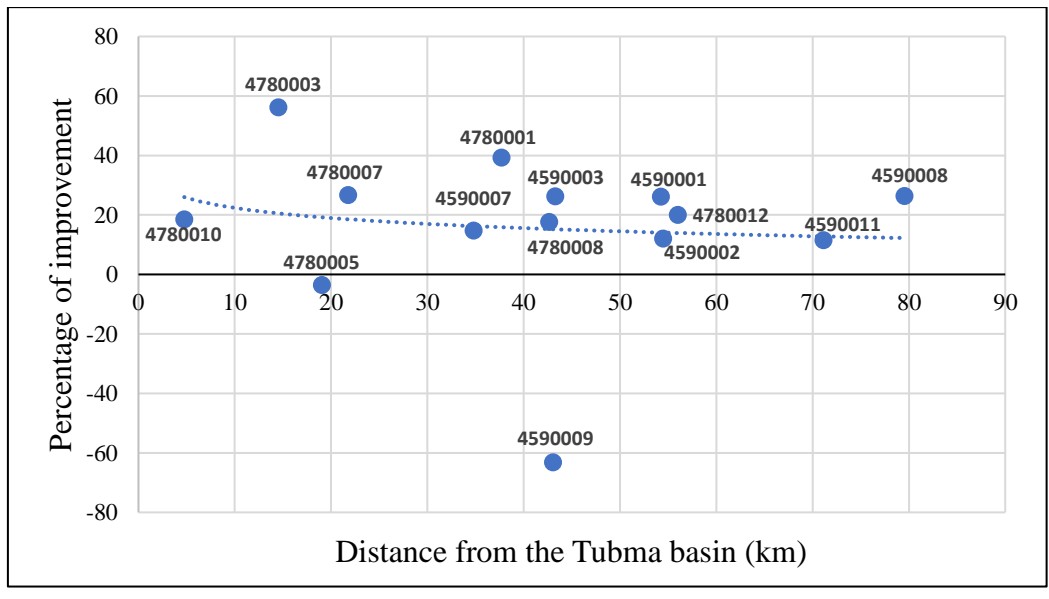

Figure C1: Alteration of the percentage improvement in radar rainfall estimates of using CKF-R$_P$ compared to No-Bias at each rain gauge with different distance from the Tubma basin.

**Appendix D: Investigation of rainfall characteristics affecting the accuracy of bias adjustment techniques**

Figure 7 (b) in the manuscript shows that the upper 75 percentile RMSE of the shorter range is remarkably high while using only TMD gauges for the bias adjustment. These errors occurred in 3 hours at different 3-gauge locations when heavy rainfall data were only measured at the validated gauge location while there was relatively uniform light rainfall at all available surrounding TMD gauges used for the bias adjustment calculation. Consequently, the calculated bias factors from the available gauges cannot represent the heavy rainfall at the tested location leading to the significant RMSE. Analysis of hourly rainfall

hyetographs obtained from TMD rain gauge network compared with the validated rain gauge occurring in three different days are illustrated in Fig. D1. It shows considerable RMSE from three hours for three days comprising 15 September 2019, 12:00; 21 September 2019, 15:00; and 22 September 2019, 14:00 associated with the validated gauge 4780001, 4780005, and 4780003, respectively. However, these RMSE values decrease considerably if the CKF-R$_P$ was implemented only in the shorter range.


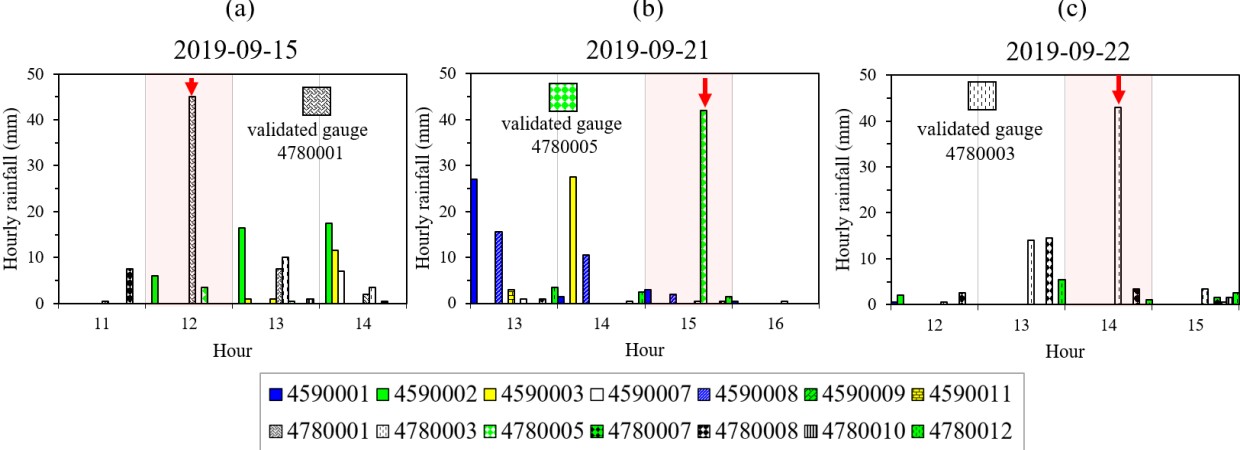

Figure D1: Hourly rainfall hyetographs obtained from TMD rain gauge network available for each hour compared with the validated rain gauge occurring in 3 different days (a) storm event during 15 September 2019 based on using 4780001 as the validated gauge, (b) storm event during 21 September 2019 based on using 4780005 as the validated gauge, and (c) storm

event during 22 September 2019 based on using 4780003 as the validated gauge.

        Indeed, Fig. 8 (a) and (b) show that one of the citizen gauges (R.18) collected cumulative monthly rainfall higher than the range of the other citizen and TMD rain gauges. This is associated with a storm event that occurred in September 2019, with the storm centre over R.18, while the surroundings received appreciably less rainfall. Figure D2 shows the radar

reflectivity field at 13.00 h on 22 September 2019, during the peak of the storm, confirming the heavy rainfall affecting gauge location R.18. This shows the citizen gauge network is able to capture local storm features thanks to the high density of the

network. The multiple reporting gaps visible in Fig. 8 (a) are caused by time errors in the observations submitted by local residents were removed from the analysis (as explained in section 2.3.2).


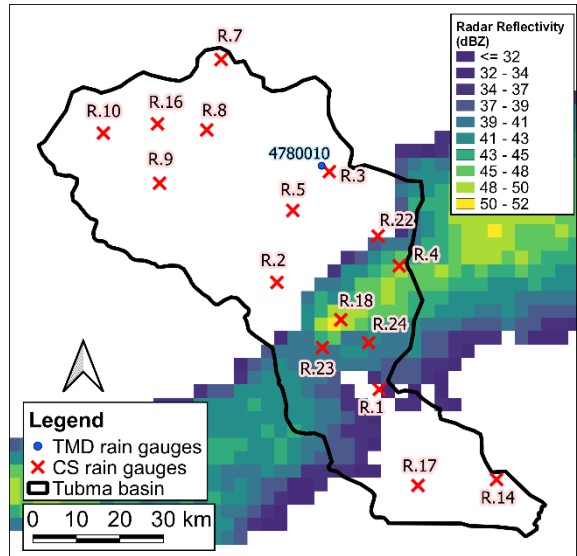

Figure D2: Overlapping between citizen rain gauges network and spatial radar reflectivity data on 22 September 2019 at 13.00 hour.


**Code and data availability**

All raw data can be provided by the corresponding authors upon request.

**Author contributions**

The study was designed by PPM, TAB and MCTV; MM, PPM and TAB were responsible of data collection and validation; MM, PPM and MCTV developed and conducted the radar bias adjustment procedure; GS, MM and PPM developed and implemented the 2-step Kalman filter. The paper was written by PPM and edited by MCTV, TAB and GS.

**Competing interests**

Two authors are members of the editorial board of Hydrology and Earth System Sciences. Thom Bogaard is executive editor and editor of HESS. Marie-Claire ten Veldhuis is editor of HESS. The peer-review process was guided by an independent editor, and the authors have also no other competing interests to declare.

**Acknowledgements.** This research is due to the research exchange between Kasetsart University (KU), Thailand and Delft
University of Technology (TU Delft), the Netherlands. The first, second and third authors gratefully acknowledge the Faculty

of Engineering, Kasetsart University for financially supporting this research, 3-weeks guest visiting in TU Delft of the first author, 3-months guest visiting in TU Delft of the second author, and 2-months guest visiting in KU of the third author. The authors wish to thank the Department of Royal Rainmaking and Agricultural Aviation (DRRAA) and Thai Meteorological Department (TMD) for providing radar reflectivity and rain gauge data used in this study. We also appreciate Mobile Water Management (MWM), the Netherlands for providing mobile application for measuring daily citizen rain gauge rainfall in the study area.

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
