# Peer review of "Citizen rain gauges improve hourly radar rainfall bias correction using a two-step Kalman filter"

_Hydrology and Earth System Sciences, 2021_

## Referee Comment (RC2)

**Review**

Title: Citizen rain gauge improves hourly radar rainfall bias correction using a two-step Kalman filter
Author(s): Punpim Puttaraksa Mapiam et al.
MS No.: hess-2021-262
MS Type: Research article

**Major comments**

- In Chapter 2.3.1 you describe that all TMD rain gauges „with more than 80% of the dataset below the threshold was exluded" (P5L125). Since the treshold of 0.5 is equal to the tipping-bucket resolution, you exluded all station with an hourly p0 (probability of value zero in the dataset) above 0.8. On P16L363 you mention that „the others having a period of heavy rainfall around 4-5 hours a day" which leads to a p0 of around 0.8.
  This applied filter excluded approximately 55% of the stations which is a lot due to the fact that the data is provided by the Thai Meteorological Department.
  Can you please provide a figure showing all TMD gauges on the one side and the used gauges on the other side?
- Was there a specific reason why the validation was done with two different data sets? This means that a comparison between the daily and the hourly data set is not meaningful.
- Was there a specific reason why the boundary between "near" stations and the "far" stations was chosen at 40km in Case 4? I think that the boundary could have been anywhere outside the Tubma basin and there would have been a significant change in the results.

**Minor comments**

- Title: „Citizen rain gauge improves …" → Citizen rain gauges improve …"
- Title, P1L11, P1L15, P1L20, P2L55,…: Kalman filter, Kalman Filter, … please be consistent in using capital letters or not
- P2L32: „the A and b parameters" → „the parameters A and b"
- P2L43, P2L44: „… the reference A paramters (…) to the A parameters for sub-daily resolutions." → please reformulate
- P2L61 and P2L62: „They found …" → please reformulate
- P3L66: „…typically provided at daily scale" → Please define citizen rain observation more precisely so that it is clear that this refers to soda bottles, for example.
- P3L75: 101°17'51" → : 101°17'51"E
- P3L75: „of approximately 197km²" → „of 197km²" or „of approximately 200km²"
- P3L83, P3L84: "240 km x 240 km (…) 0.6x0.6km" → Please stay with one style
- P3L84: "… spatial resolution and 6-min temporal resolution" → „… spatial and 6-min temporal resolution"
- P3L90: „3 datasets" → „three datasets"
- P3L90-92: What happend between 2014 and 2019?
- P3L121: Is it "Thai Meteorological Department" or "Thailand Meteorological Department"?
- P4L102 and L103: "1-hour" → "hourly"
- P4L103: "A parameters" → "parameters A"
- P4L104: "b exponent" → "exponent b"

- P4L106: "b parameter" → "parameter b"
- P4L108: „mean absolut error" → „MAE" (defined on P4L104)
- P4L110: Since $N_{G,t}$ seems not be equal to N over the entire period T, the Equation is incorrect. Which equation was used in your calculations?
  → $MAE = \frac{1}{T}\sum_{t=1}^{T} \frac{1}{N_{G,t}} \sum_{i=1}^{N_{G,t}} |G_{i,t} - R_{i,t}|$
- P4L122: „have tipping-bucket sizes of 0.5mm" → something like "have a resolution of 0.5mm"
- P4L130: „"in the 197km² Tubma basin" → „in the Tubma basin"
- P4L133: 1 gauge/15km²: How did you calculate that? One TMD station + 16 citizen rain gauges = 17 stations. 197km² / 17 stations → 1 gauge/~12km²
- P6Table1: „Code description" → „Code Description"
- P7L158: Is MFB the abbreviation for „Mean field bias" or „Mean fiel bias adjustment"? If the latter, I would suggest MFBA, since MFB is a common abbreviation for mean field bias. If the former, use this abbreviation (P7L162, P7L168).
- P7L163f: „...Smith and Krajewski (1991), Anagnostou et al. (1998), and Seo et al. (1999), Chumchean et al. (2006), Kim and Yoo, (2014), Shi et al. (2018)." → „...Smith and Krajewski (1991), Anagnostou et al. (1998), Seo et al. (1999), Chumchean et al. (2006), Kim and Yoo, (2014), and Shi et al. (2018)."
- P7L177: "The radar bias at time t …" → The radar bias at time $t$ …")
- P10L254: „... day i …" → „... day $i$ …"
- P11Figure3: „Is hourly TMD data at hour $t$ available?" → „Is hourly TMD data $y$ at hour $t$ available?"
- P11Figure3: „Is hourly citizen rain gauge data at hour $t$ available?" → „Is hourly citizen rain gauge data $z$ at hour $t$ available?"
- P12L287: „... hour t …" → „... hour $t$ …"
- P12L290: „... time t …" → „... time $t$ …"
- P12L295: „Kalman Filter" → „KF" From here on, no more explicit mention for missing abbreviations. Please check independently in the following.
- P13L303: „1 TMD" → „one TMD"
- P13L306: „randomly": Was the LOOCV done for all 16 citizen rain gauges or did you randomly sample 16 times? If the former, it is not really randomly.
- P13L306: „3 different techniques, and 1 rain gauge" → „three different techniques, and one rain gauge"
- P13L310: „... gauge i …" → „... gauge $i$ …"
- P14L324: see P13L306
- P14L325: „1 TMD" → „one TMD"
- P14L327f: „fourteen TMD" → „14 TMD"
- P14L338: „(leave 1 TMD out)" → „(leave one TMD out)"
- P14L345 and L347: "Kalman Filter" → "KF"
- P15L347: "$r_1$ parameter" → "parameter $r_1$"
- P15L350: "over the same time-series period" → "over the same period"
- P16L372: "figure 5" → "Fig.5"
- P17L391: "observationss" → "observations"
- P17L393: "are based on 4" → "are based on four"
- Figure 6 and 7: Since RMSE and MBE have different limits (0 to infinity vs. -infinity to infinity), it does not make sense to put both assessment measures on one graph.

- P18L414: "respectively)" → "respectively)."
- P18L432: "Figure 7 (b) and Fig. 7 (c)" → "Figure 7 (b) and (c)"
- P18L435: see P18L432
- Figure 9: Please increase the fontsize and add a grid.
- P22L490: "(August-October, 2019)" → "(August-October 2019)"
- P22L495 and L497: "Kalman filter" → "KF" or „Kalman Filter"

---

## Author Comment (AC1)

**Author Response to the referee comments on "Citizen rain gauge improves hourly radar rainfall bias correction using a two-step Kalman filter"**

by Punpim Puttaraksa Mapiam [1*], Monton Methaprayun [1], Thom Adrianus Bogaard [2], Gerrit Schoups[2], Marie-Claire Ten Veldhuis[2]

[1]Department of Water Resources Engineering, Kasetsart University, PO Box 1032, Bangkok, 10900, Thailand
[2]Department of Water Management, Delft University of Technology, PO Box 5048, 2600 GA Delft, The Netherlands

**Responses to referee #1**

RC: Referee comment
AR: Author response

**General comments**

My research interest is focused on citizen science weather observations (in particular rainfall monitoring), so whilst I am confident in reviewing that element of the paper, I am not well placed to comment on the technical aspects of radar rainfall bias correction.

The paper presents a novel approach to undertaking radar rainfall bias correction in an area with sparse official ground based rain gauges. The focus of the paper is on the methodology for bias correction, however there perhaps should be equal prominence for the selection of a method for the disaggregation of daily data to hourly, which seems as significant. Many researchers are sceptical about the accuracy of citizen science weather observations, and it would be beneficial to allay such concerns by providing more detail on how to undertake citizen science that generates data good enough to draw the conclusion made herein. Given the intrinsic difficulties in establishing the "true" rainfall amount due to spatial and temporal variation, the conclusions could have been more robustly supported by applying the methodology to an area where validation using official gauges could have been more effectively applied.

AR: Thanks for your comment, we agree that acquiring good quality citizen observations is a challenge and we put considerable effort into citizen engagement during the measurement campaign and in data quality control afterwards. Since the focus of this paper is on bias correction, we chose not to include details on the citizen science campaign in the paper. We agree that conducting a similar campaign in an area well supported by official gauges, could have led to more robust results, however we believe that the added value of citizen science lies precisely in the type of region where we conducted our study. Regions supported by dense, high-quality rain gauge networks (like EU, US) represent different conditions for implementation of citizen science and generally different rainfall climatologies.

**Specific comments**

RC (1): A detailed methodology relating to the Kalman Filter is provided, but it overshadows some of the more basic and fundamental details on which the paper is based. There is no justification provided on the selection of the Tubma basin as the research area. More information would be appreciated in "Section 2.1 Study Area" describing the nature of the basin and the climatic characteristics. Lines 456 – 459 highlight some of the limitations of the study area, which I feel deserve more prominence. The location seems like a difficult place to study and have confidence in the results due to the limited opportunity for external validation via TMD gauging.

AR: More information describing the reason why the Tubma basin was selected as a research area will be added in section 2.1 of the revised manuscript as described below.

*"In Figure 1 (will be inserted in the revised manuscript) we show the climatological variation across the study area and its surroundings, based on 30-year (1987-2017) annual mean rainfall from the network of 311 daily rain gauges owned by the TMD and situated within 200 km range from the Tubma basin. Spatial rainfall patterns were generated by inverse distance squared (IDS) between the gauge locations. The map shows that while there is small gradient in mean annual rainfall (1,100 to 1,700 mm mean annual rainfall) across the area of Rayong and Chonburi provinces (within 90 km from the study area), changes are more pronounced when the distance exceeds the beyond the 90 km boundary, especially to the east of the study area. This is because these areas are affected differently by the southwest monsoon. Consequently, evaluating the effectiveness of bias correction techniques were carried out within 90 km range from the study area with similar climatology."*

[Figure]

*Figure 1: Climatological spatial rainfall distribution within 200 km range from the centroid of the Tubma basin calculated from 30-year average annual rainfall data of 311 daily rain gauge network by using IDS method.*

RC (2): Given the climatic variation across the study area noted in Section 4.3.4, justification on the use of a 100Km radius for evaluation would be welcome.
AR: The 100 km radius refers to the coverage area of the Sattahip radar. The correct radar range is actually 90 km, as indicated in section 4.3.4, we will make sure to correct all references to 100 km by 90 km in the revised manuscript.

RC (3): There is no explanation of how citizen science observations were made, how participants were recruited and trained, or discussion of citizen science limitations either in general or encountered in this research.

AR: For more precise understanding about the citizen rain observation, line between 130 to 138 in the old manuscript will be rewritten as a new section "2.3.2 Citizen Rain Observation" as described below.

*2.3.2 Citizen Rain Observation*

*Out of the total TMD rain gauge network, only one rain gauge is located in the Tubma basin. To increase the density of the rain gauge network in the basin, low-cost citizen rain gauges were implemented in this study to better capture spatial heterogeneity of rainfall in the basin. Sixteen citizen rain gauges were installed (Fig. 2: Note that Fig . 2 in the revised manuscript will be adjusted from the Fig. 1 of the old manuscript) with local residents taking daily measurements. The additional 16 citizen rain gauges with one station located at the same place of the existing TMD gauge increased the density of rain gauges in the Tubma basin to 1 gauge/12 km². The citizen observations were made by installing a cone-shape transparent plastic rain gauge which is standardly used in South Africa (see Fig. S1) with a diameter of 5 inches and capacity of 100 mm in open space area around a school, Monasteries, bridge or other building. Mobile application developed by Mobile Water Management (MWM) (Mobile Water Management, 2020), the Netherlands, was used to record rainfall data for each rain gauge on a daily basis. The application has a an easily accessible and user-friendly interface where participants simply fill in the observed rainfall amount, take a photo of the rain gauge and upload this to the application. The photo and the rainfall data, together with the measuring location and time, are automatically stored in the database. Photos are used for visual validation of the recorded rainfall depth to eliminate errors.*

*In this study, participants were recruited amongst government officers, teachers, and local residents living close to the stations and were trained to take measurements at around 7 a.m. daily according to the TMD standards. Quality of the collected data was assured by the high photo resolution for double-checking the observations and strict requirements on measurement times to be consistent with the same standard of TMD for daily rainfall recording. Note that the maximum rainfall for the citizen gauges is 100mm/day.*

*Validation of the cone-shaped citizen gauges was conducted based on a citizen gauge co-located with an automatic TMD gauge located in the Tubma basin, during August – October 2019. The citizen gauge installed at the same location R3 (Fig. S2) as a TMD gauge showed good similarity with a random RMSE of 5.5 mm.*

*Quality control consisted of screening all citizen rain gauge data for errors and inconsistencies using double mass curves. If citizen rain gauges reported >100mm/day rainfall (maximum capacity of the citizen rain gauge) this data was excluded from the analysis. If days with no-rainfall data were found from all citizen rain gauges, the bias correction of that day was*

*discarded from the dataset. By considering the data selection criteria, rainfall data recorded during August–October 2019 with rainy days, more than 80% of the whole period for the bias adjustment process was then used for further analysis."*

[Figure]

*Figure S1: (in supplementary information to the revised manuscript) An example of* installing a cone-shape transparent plastic citizen rain gauge *at location R.22, Map Tong school.*

[Figure]

*Figure S2: (in supplementary information to the revised manuscript) Daily rainfall depth comparison between the TMD and citizen rain gauge at location R.3 during August – October 2019.*

RC (4): There do appear to be interesting results from the citizen science gauges that warrant further discussion. E.g, Fig.9 (a) indicates a range in the monthly cumulative rainfall from citizen science gauges of ~170 – 400mm and multiple reporting gaps, whereas (b) the TMD gauge range is ~150 – 260mm. A map with the gauges identified and some detail on elevation or climatic region could be included.

AR: Indeed, Fig.9 (a) and (b) show that one of the citizen gauges (R.18) collected cumulative monthly rainfall higher than the range of the other citizen and TMD rain gauges. This is associated with a storm event that occurred in September 2019, with the storm center over R.18, while the surroundings received appreciably less rainfall. Figure S2 (this figure will be added in the supplementary of the revised manuscript) shows the radar reflectivity field at 13.00h on 22 September 2019, during the peak of the storm, confirming the heavy rainfall affecting gauge location R.18. This shows the citizen gauge network is able to capture local storm features thanks to the high density of the network. The multiple reporting gaps visible in Fig. 9 (a) are caused by time errors in the observations submitted by local residents were removed from the analysis (as will be explained in new section 2.3.2). The map shown in Figure S3 will be provided in the revised manuscript.

[Figure]

*Figure S2: Overlapping between citizen rain gauges network and spatial radar reflectivity data on 22 September 2019 at 13.00 hour*

RC (5) If some detail on the application of existing methods could be moved to "Supplementary Information", it would allow more space for consideration of the citizen science element.

AR: Since the focus of our study is on radar bias adjustment, we have not elaborated in detail on the citizen observations network separately. We believe it would distract from the focus of the study, which is on the 2-step Kalman-filter bias adjustment methodology. This is the most novel element of the work.

**Technical comments**

RC (5): General point – Although common practice I found the use of acronyms made the paper hard to follow at times e.g. Section 4.2

AR: Because we have a lot of approaches for comparison and would like to make the paper short, we then use the acronyms to tell the story. However, we will try to use as less acronyms as possible.

RC (6): Figure 1 – Identification of citizen science gauges

AR: Figure. 1 was modified to be Figure 2 in the revised manuscript to show all used gauges including the 14 rain gauges in Chonburi and Rayong provinces as shown by the following. The identification of citizen rain gauges was also added in the figure.

[Figure]

*Figure 2 (in the revised manuscript): Location of study domain, showing Thai Meteorological Department (TMD) automatic rain gauges, citizen rain gauges, Sattahip radar, and Tubma basin.*

RC (7): Line 164 – replace full stop with comma
AR: We will check the format and replace them correctly in the revised manuscript.

RC (8): Line 254 – Point 1 is not clear about the timing of data, could this be rephrased?
AR: The point 1 of P10L254 will be rephrased according to the comment by follows.

"1) Since the citizen rain gauge data were received at the last hour of day i, before receiving the downscaled hourly citizen rain gauge data of day i, the ordinary KF and observed hourly data of TMD were used to predict and correct the hourly bias adjustment factor of the day "

RC (9): Line 270 – replace "In case" with "Where"
AR: This correction will be implemented in the revised manuscript.

RC (10): Figure 4 – a grid may make reading easier, or may be too cluttered?
AR: Figure 4 was modified as shown below and will be placed in the revised manuscript.

[Figure]

RC (11): Figure 9 – The colours are indistinct for the different gauges, the x ticks could be 'day' and the figures in general are very small making them hard to read on the page

AR: Thank for the suggestion, the figure will be adjusted as the following.

[Figure]

*Figure 9 (in the revised manuscript): Comparison of mass curve of hourly rainfall among various rain-gauge locations (a) the citizen rain gauges located in the Tubma basin (b)TMD rain gauges within 0-40 km radius from the Tubma basin (c) TMD rain gauges within 40-90 km radius from the Tubma basin.*

Reference

Mobile Water Management: https://mobilewatermanagement.nl/, last access: 25 October 2020.

---

## Author Comment (AC2)

**Author Response to the referee comments on "Citizen rain gauge improves hourly radar rainfall bias correction using a two-step Kalman filter"**

by Punpim Puttaraksa Mapiam [1*], Monton Methaprayun [1], Thom Adrianus Bogaard [2], Gerrit Schoups[2], Marie-Claire Ten Veldhuis[2]

[1]Department of Water Resources Engineering, Kasetsart University, PO Box 1032, Bangkok, 10900, Thailand
[2]Department of Water Management, Delft University of Technology, PO Box 5048, 2600 GA Delft, The Netherlands

**Responses to referee #2**

RC: Referee comment
AR: Author response

**Major comments**

RC (1) In Chapter 2.3.1 you describe that all TMD rain gauges "with more than 80% of the dataset below the threshold was excluded" (P5L125). Since the threshold of 0.5 is equal to the tipping bucket resolution, you excluded all station with an hourly p0 (probability of value zero in the dataset) above 0.8. On P16L363 you mention that "the others having a period of heavy rainfall around 4-5 hours a day" which leads to a p0 of around 0.8. This applied filter excluded approximately 55% of the stations which is a lot due to the fact that the data is provided by the Thai Meteorological Department. Can you please provide a figure showing all TMD gauges on the one side and the used gauges on the other side?

AR: thanks for pointing this out, we realize the phrasing may have created confusion.
The rain gauge selection was based on a threshold p0 of 0.8 at the daily scale. Only 134 out of 297 stations passed this test. Actually, many of the rejected stations recorded zero values throughout most of the study period. Of the remaining stations, many reported heavy rainfall during 4-5 hours of the day (which indeed leads to a p0 of 0.8 at the hourly scale), as stated in P16L363.

We propose to rephrase the sentence (P5L125) as follows, to avoid confusion:
*"Rain gauges with more than 80% of the recordings below the 0.5 mm threshold at daily scale were excluded from the analysis. It turns out that many of these faulty gauges recorded zero rainfall throughout most of the study period."*

We will also modify figure 1, to distinguish the selected rain gauges:

[Figure]

*Figure 1 (modified for revised manuscript): Location of study domain, showing Thai Meteorological Department (TMD) automatic rain gauges, citizen rain gauges, Sattahip radar, and Tubma basin.*

RC (2) Was there a specific reason why the validation was done with two different data sets? This means that a comparison between the daily and the hourly data set is not meaningful.

AR: The question we aimed to answer in this manuscript is to what extent citizen rainfall observations improve the accuracy of hourly radar rainfall estimates (by using a 2-step Kalman filter approach). Since the citizen rainfall observations are made at daily scale, we need a downscaling procedure to match the radar data scale. The downscaling step introduces additional uncertainty, that's why we chose to validate the bias correction results not only at hourly but also at daily time-scale. The validation procedures are explained in detail in Section 3.4 and summarized in Table 2.

RC (3) Was there a specific reason why the boundary between "near" stations and the "far" stations was chosen at 40km in Case 4? I think that the boundary could have been anywhere outside the Tubma basin and there would have been a significant change in the results.

AR: We chose the 40 km separation boundary to achieve an equal number of gauges in the "near" and "far" groups. Following the referee's comment, we investigated RMSE between gauge rainfall and radar rainfall without bias adjustment ($RMSE_{No-Bias}$) and with the CKF-$R_P$ ($RMSE_{CKF-RP}$) for individual stations (located at distances 5 – 80 km from the Tubma basin). We computed the percentage improvement in radar rainfall estimates using CKF-$R_P$ compared to No-Bias at each rain gauge, indicating the relative errors changing with distance from the Tubma basin (Figure 2).

Figure 2 shows that the improvement percentage of using CKF-R$_P$ tends to reduce with increasing distance from the Tubma basin, where the citizen gauges are located. The percentage reduction gradually decreases beyond a distance of about 40 km.

Note that the gauge 4780010 situated at the nearest range is expected to provide the best improvement, however, the Sattahip radar temporarily stopped measuring for 3 hours (during 16:00 h - 19:00 h of 24 September 2019) which is associated with a heavy storm's center was only at the gauge 4780010. This leads to significant degradation of the radar rainfall performance. Furthermore, the lower percentage improved for station 4780005 is associated with a localized heavy rainfall event that was recorded only at this location, negatively affecting its performance as a representative station for bias correction.

[Figure]

*Figure 2: Alteration of the percentage improvement in radar rainfall estimates of using CKF-R$_P$ compared to No-Bias at each rain gauge with different distance from the Tubma basin.*

We propose to add this information and the associated figure in Supplementary Information.

**Minor comments**

RC (1) Title: "Citizen rain gauge improves …" → "Citizen rain gauges improve …"
*AR: Thanks for correcting this, we will revise the manuscript accordingly.*

RC (2) Title, P1L11, P1L15, P1L20, P2L55, …: Kalman filter, Kalman Filter, … please be consistent in using capital letters or not.
*AR: This mistake will be corrected in the revised manuscript by using only the Kalman filter.*

RC (3) P2L32: "the A and b parameters" → "the parameters A and b"
RC (4) P2L43, P2L44: "… the reference A parameters (…) to the A parameters for sub-daily resolutions." → please reformulate
RC (5) P2L61 and P2L62: "They found …" → please reformulate
*AR: Comments number (3) to (5) will be reformulated in the revised the manuscript.*

RC (6) P3L66: "…typically provided at daily scale" → Please define citizen rain observation more precisely so that it is clear that this refers to soda bottles, for example.
*AR: For more precise understanding about the citizen rain observation, line between 130 to 138 in the old manuscript will be rewritten as a new section "2.3.2 Citizen Rain Observation" as described below.*

*"2.3.2 Citizen Rain Observation*

*Out of the total TMD rain gauge network, only one rain gauge is located in the Tubma basin. To increase the density of the rain gauge network in the basin, low-cost citizen rain gauges were implemented in this study to better capture spatial heterogeneity of rainfall in the basin. Sixteen citizen rain gauges were installed (Fig. 1) with local residents taking daily measurements. The additional 16 citizen rain gauges with one station located at the same place of the existing TMD gauge increased the density of rain gauges in the Tubma basin to 1 gauge/12 km². The citizen observations were made by installing a cone-shape transparent plastic rain gauge which is standardly used in South Africa (see Fig. S1) with a diameter of 5 inches and capacity of 100 mm in open space area around a school, Monasteries, bridge or other building. Mobile application developed by Mobile Water Management (MWM) (Mobile Water Management, 2020), the Netherlands, was used to record rainfall data for each rain gauge on a daily basis. The application has a an easily accessible and user-friendly interface where participants simply fill in the observed rainfall amount, take a photo of the rain gauge and upload this to the application. The photo and the rainfall data, together with the measuring location and time, are automatically stored in the database. Photos are used for visual validation of the recorded rainfall depth to eliminate errors.*

*In this study, participants were recruited amongst government officers, teachers, and local residents living close to the stations and were trained to take measurements at around 7 a.m. daily according to the TMD standards. Quality of the collected data was assured by the high photo resolution for double-checking the observations and strict requirements on measurement times to be consistent with the same standard of TMD for daily rainfall recording. Note that the maximum rainfall for the citizen gauges is 100mm/day.*

*Validation of the cone-shaped citizen gauges was conducted based on a citizen gauge co-located with an automatic TMD gauge located in the Tubma basin, during August – October 2019. The citizen gauge installed at the same location R3 (Fig. S2) as a TMD gauge showed good similarity with a random RMSE of 5.5 mm.*

*Quality control consisted of screening all citizen rain gauge data for errors and inconsistencies using double mass curves. If citizen rain gauges reported >100mm/day rainfall (maximum capacity of the citizen rain gauge) this data was excluded from the analysis. If days with no-rainfall data were found from all citizen rain gauges, the bias correction of that day was discarded from the dataset. By considering the data selection criteria, rainfall data recorded during August–October 2019 with rainy days, more than 80% of the whole period for the bias adjustment process was then used for further analysis."*

[Figure]

*Figure S1: (in supplementary information to the revised manuscript) An example of* installing a cone-shape transparent plastic citizen rain gauge *at location R.22, Map Tong school.*

[Figure]

*Figure S2: (in supplementary information to the revised manuscript) Daily rainfall depth comparison between the TMD and citizen rain gauge at location R.3 during August – October 2019.*

RC (7) P3L75: 101°17'51" → : 101°17'51"E
RC (8) P3L75: "of approximately 197km²" → "of 197km²" or "of approximately 200km²"
*AR: Comments number (7) to (8) will be changed in the revised the manuscript.*

RC (9) P3L83, P3L84: "240 km x 240 km (…) 0.6x0.6km" → Please stay with one style
*AR: We will edit 240 km x 240 km to 240x240 km.*

RC (10) P3L84: "… spatial resolution and 6-min temporal resolution"  → "spatial and 6-min temporal resolution"
RC (11) P3L90: "3 datasets"→ "three datasets"
*AR: Comments number (10) to (11) will be rewritten in the revised the manuscript.*

RC (12) P3L90-92: What happened between 2014 and 2019?
*AR: Since, the citizen rain gauges were installed in the Tubma basin in rainy season of 2019. This is quite a short period for the Z-R calibration and verification. We then collected more data in 2013 and 2014 for the analysis of Z-R calibration and validation, while keep the data in 2019 for the bias correction development and evaluation.*

RC (13) P3L121: Is it "Thai Meteorological Department" or "Thailand Meteorological Department"?

*AR: It is Thai Meteorological Department officially. We will correct the caption of Figure 1 in the manuscript accordingly.*

RC (14) P4L102 and L103: "1-hour" → "hourly"
RC (15) P4L103: "A parameters" → "parameters A"
RC (16) P4L104: "b exponent" → "exponent b"
RC (17) P4L106: "b parameter" → "parameter b"
RC (18) P4L108: "mean absolute error" → "MAE" (defined on P4L104)

*AR: Comments number (14) to (18) will be corrected in the revised the manuscript.*

RC (19) P4L110: Since NG,t seems not be equal to N over the entire period T, the Equation is incorrect. Which equation was used in your calculations?

*AR: Thanks for pointing this out. $N_{G,t}$ was changed to $N_G$ to represent the number of rain gauges and N was replaced with multiplying between T and $N_G$ to represent total number of data pairs used in the calculation. The equation was revised including related description of each variable as described below.*

$$MAE = \frac{1}{TN_G}\sum_{t=1}^{T}\sum_{i=1}^{N_G}|G_{i,t} - R_{i,t}|$$

*where $G_{i,t}$ is the gauge rainfall (mm/h) at gauge i for hour t, $R_{i,t}$ is the radar rainfall accumulation (mm/h) at the pixel corresponding to the $i^{th}$ rain gauge for hour t, $N_G$ is the number of rain gauges, and T is the number of time period used in the calculation.*

RC (20) P4L122: "have tipping-bucket sizes of 0.5mm" → something like "have a resolution of 0.5mm"

*AR: We will replace "have tipping-bucket sizes of 0.5mm" with "have a resolution of 0.5mm"*

RC (21) P4L130: "in the 197km² Tubma basin" → "in the Tubma basin"

*AR: This will be corrected in the revised the manuscript.*

RC (22) P4L133: 1 gauge/15km²: How did you calculate that? One TMD station + 16 citizen rain gauges = 17 stations. 197km² / 17 stations → 1 gauge/~12km²

*AR: There was a mistake because of using a wrong number of citizen rain gauges in the calculation. The correct total number of rain gauges is 16 stations since 1 station of citizen rain gauge was located at the same place of the existing TMD. The density was change to be 197 km² / 16 stations → 1 gauge/~12km² in the revised manuscript.*

RC (23) P6Table1: "Code description" → "Code Description"

*AR: This will be corrected in the revised the manuscript.*

RC (24) P7L158: Is MFB the abbreviation for "Mean field bias" or "Mean field bias adjustment"? If the latter, I would suggest MFBA, since MFB is a common abbreviation for mean field bias. If the former, use this abbreviation (P7L162, P7L168).

*AR: MFB is the abbreviation for Mean field bias, so P7L162 and P7L168 we will carefully be corrected.*

RC (25) P7L163f: "…Smith and Krajewski (1991), Anagnostou et al. (1998), and Seo et al. (1999), Chumchean et al. (2006), Kim and Yoo, (2014), Shi et al. (2018)." → "…Smith and Krajewski (1991), Anagnostou et al. (1998), Seo et al. (1999), Chumchean et al. (2006), Kim and Yoo, (2014), and Shi et al. (2018)."

RC (26) P7L177: "The radar bias at time t …" → "The radar bias at time $t$ …")

RC (27) P10L254: "… day i …" → "… day $i$ …"

RC (28) P11Figure3: "Is hourly TMD data at hour t available?" → "Is hourly TMD data $y$ at hour $t$ available?"

RC (29) P11Figure3: "Is hourly citizen rain gauge data at hour t available?" → "Is hourly citizen rain gauge data $z$ at hour $t$ available?"

RC (30) P12L287: "… hour t …" → "… hour $t$ …"

RC (31) P12L290: "… time t …" → "… time $t$ …"

*AR: All the mistakes mentioned in the comment numbers (25)-(31) will carefully be corrected in the revised manuscript.*

RC (32) P12L295: "Kalman Filter" → "KF" From here on, no more explicit mention for missing abbreviations. Please check independently in the following.

*AR: We thank for pointing this out, we will carefully check the abbreviations.*

RC (33) P13L303: "1 TMD" → "one TMD"

*AR: This suggestion will be implemented in the revised manuscript.*

RC (34) P13L306: "randomly": Was the LOOCV done for all 16 citizen rain gauges or did you randomly sample 16 times? If the former, it is not really randomly.

*AR: The LOOCV was done for all the 16 citizen rain gauges. Therefore, P13L306 will rewritten as follows.*

*"Leave-one-out cross-validation (LOOCV) algorithm was implemented to avoid bias occurring from selecting the validation rain gauges. For each round of cross-validation, one rain gauge was left out for validation and the remaining rain gauges was used as the calibration rain gauges to calculate the bias adjustment factor using the 3 different techniques."*

RC (35) P13L306: "3 different techniques, and 1 rain gauge" → "three different techniques, and one rain gauge"

RC (36) P13L310: "… gauge i …" → "… gauge $i$ …"

RC (37) P14L324: see P13L306
RC (38) P14L325: "1 TMD" → "one TMD"
RC (39) P14L327f: "fourteen TMD" → "14 TMD"
RC (40) P14L338: "(leave 1 TMD out)" → "(leave one TMD out)"
RC (41) P14L345 and L347: "Kalman Filter" → "KF"
RC (42) P15L347: "r1 parameter" → "parameter r1"
RC (43) P15L350: "over the same time-series period" → "over the same period"
RC (44) P16L372: "figure 5" → "Fig.5"
RC (45) P17L391: "observationss" → "observations"
RC (46) P17L393: "are based on 4" → "are based on four"
*AR: All the mistakes mentioned in the comment numbers (35)-(46) will carefully be corrected in the revised manuscript.*

RC (47) Figure 6 and 7: Since RMSE and MBE have different limits (0 to infinity vs. -infinity to infinity), it does not make sense to put both assessment measures on one graph.
*AR: Regarding these figures, we intended to compare RMSE and MBE across different techniques corresponding to each scenario of the study. We then designed the graph to be readable from two independent y-axes. The primary y-axis on the left indicates the RMSE and the secondary y-axis on the right indicates the MBE.*

RC (48) P18L414: "respectively)" → "respectively)."
RC (49) P18L432: "Figure 7 (b) and Fig. 7 (c)" → "Figure 7 (b) and (c)"
RC (50) P18L435: see P18L432
*AR: All the mistakes mentioned in the comment numbers (48)-(50) will carefully be corrected in the revised manuscript.*

RC (51) Figure 9: Please increase the font size and add a grid.
*AR: Figure 9 will be adjusted in the revised manuscript as follows.*

[Figure]

*Figure 9 (in the manuscript): Comparison of mass curve of hourly rainfall among various rain-gauge locations (a) the citizen rain gauges located in the Tubma basin (b)TMD rain gauges within 0-40 km radius from the Tubma basin (c) TMD rain gauges within 40-90 km radius from the Tubma basin.*

RC (52) P22L490: "(August-October, 2019)" → "(August-October 2019)"

*AR: This suggestion will be implemented in the revised manuscript.*

RC (53) P22L495 and L497: "Kalman filter" → "KF" or "Kalman Filter"

*AR: Kalman filter will only be used in the revised manuscript.*

---

## Author Comment (AC3)

**Author Response to the referee comments on "Citizen rain gauge improves hourly radar rainfall bias correction using a two-step Kalman filter"**

*by Punpim Puttaraksa Mapiam [1*], Monton Methaprayun [1], Thom Adrianus Bogaard [2], Gerrit Schoups[2], Marie-Claire Ten Veldhuis[2]*

[1]*Department of Water Resources Engineering, Kasetsart University, PO Box 1032, Bangkok, 10900, Thailand*
[2]*Department of Water Management, Delft University of Technology, PO Box 5048, 2600 GA Delft, The Netherlands*

**Responses to referee #3**
RC: Referee comment
AR: Author response

**General comments**

The sparsity of the conventional rain gauge network is a limiting factor for radar rainfall bias correction. Citizen rain gauges offer an opportunity to provide additional information at higher spatial resolution. This paper performs radar rainfall bias adjustment using two sources of rainfall information: observations measured by TMD (Thailand Meteorological Department) and daily citizen rainfall observations. The radar rainfall bias correction factor was sequentially updated based on the TMD data and downscaled hourly citizen data via a two-step Kalman filter. The results showed that citizen rain gauges improved the performance of radar rainfall bias adjustment especially for small ranges from the center of these gauges.

My reading through this manuscript suggests that the following three issues should be dealt with or stressed more clearly in the paper:

RC (1):  The citizen rain gauges are very important in the context of this paper. However, the relevant introduction is too simple. What kind of equipment are they? How uncertain are the measurements compared to the official TMD data? Please add more information on them.

AR: For more in-depth description of the citizen rain observation, lines 130 to 138 in the old manuscript will be rewritten as a new section "2.3.2 Citizen Rain Observation" as described below.
*"2.3.2 Citizen Rain Observation*
*Out of the total TMD rain gauge network, only one rain gauge is located in the Tubma basin. To increase the density of the rain gauge network in the basin, low-cost citizen rain gauges were implemented in this study to better capture spatial heterogeneity of rainfall in the basin. Sixteen citizen rain gauges were installed (Fig. 1) with local residents taking daily measurements. The additional 16 citizen rain gauges with one station located at the same place of the existing TMD gauge increased the density of rain gauges in the Tubma basin to 1 gauge/12 km². The citizen observations were made by installing a cone-shape transparent plastic rain gauge which is standardly used in South Africa (see Fig. S1) with a diameter of 5 inches and capacity of 100*

*mm in open space area around a school, Monasteries, bridge or other building. Mobile application developed by Mobile Water Management (MWM) (Mobile Water Management, 2020), the Netherlands, was used to record rainfall data for each rain gauge on a daily basis. The application has a an easily accessible and user-friendly interface where participants simply fill in the observed rainfall amount, take a photo of the rain gauge and upload this to the application. The photo and the rainfall data, together with the measuring location and time, are automatically stored in the database. Photos are used for visual validation of the recorded rainfall depth to eliminate errors.*

*In this study, participants were recruited amongst government officers, teachers, and local residents living close to the stations and were trained to take measurements at around 7 a.m. daily according to the TMD standards. Quality of the collected data was assured by the high photo resolution for double-checking the observations and strict requirements on measurement times to be consistent with the same standard of TMD for daily rainfall recording. Note that the maximum rainfall for the citizen gauges is 100mm/day.*

*Validation of the cone-shaped citizen gauges was conducted based on a citizen gauge co-located with an automatic TMD gauge located in the Tubma basin, during August – October 2019. The citizen gauge installed at the same location R3 (Fig. S2) as a TMD gauge showed good similarity with a random RMSE of 5.5 mm.*

*Quality control consisted of screening all citizen rain gauge data for errors and inconsistencies using double mass curves. If citizen rain gauges reported >100mm/day rainfall (maximum capacity of the citizen rain gauge) this data was excluded from the analysis. If days with no-rainfall data were found from all citizen rain gauges, the bias correction of that day was discarded from the dataset. By considering the data selection criteria, rainfall data recorded during August–October 2019 with rainy days, more than 80% of the whole period for the bias adjustment process was then used for further analysis."*

[Figure]

*Figure S1: (in supplementary information to the revised manuscript) An example of* installing a cone-shape transparent plastic citizen rain gauge *at location R.22, Map Tong school.*

[Figure]

*Figure S2: (in supplementary information to the revised manuscript) Daily rainfall depth comparison between the TMD and citizen rain gauge at location R.3 during August – October 2019.*

RC (2): The strategy used to downscale the daily measured citizen rainfall observations into hourly temporal resolution is essential in this context. However, there were no references on this topic. Four relatively simple downscaling strategies were tested, and the results showed that the one that utilized gauge-respective radar rainfall patterns performed the best. Yet I do think there is room for improvement.

AR: The focus of the paper is on developing a new algorithm called two-step Kalman filter using two sources of observed rain gauge data. The strategy for combing TMD and citizen rain gauge datasets in the second measurement updating of the Kalman filter was a core of the development. Simulating different hourly citizen dataset with 4 downscaled strategies was carried out to investigate the sensitivity of the modified KF approach. As a result, in this study, we decided to use a simple fraction method for disaggregating daily rainfall to hourly values. A similar strategy has been applied in several previous applications (Paulat et al., 2008; Wüest et al., 2010; Vormoor and Skaugen, 2013; Sideris et al., 2014; Barton et al., 2019). Additional literature review on downscaling techniques will be provided in the introduction section of the revised manuscript as described below.

*"There has been a variety of temporal rainfall downscaling methods developed since the 1970s. The simplest approach is to distribute daily rainfall data to sub-daily resolutions by assuming uniform distributions. Stochastically generating sub-period data or spatially transferring finer resolution rainfall from a nearby rain gauge station to the study area based on spatial correlations are alternative approaches (Koutsoyiannis, 2003; Debele et al., 2007). However, these methods are not usually designed for real-time data disaggregation over large areas. Instead, a common approach for such scenarios is to downscale daily rainfall based on a simple fraction technique by considering the distribution patterns of high-resolution gridded rainfall products from radar or satellite sensors (Paulat et al., 2008; Wüest et al., 2010; Vormoor and Skaugen, 2013; Sideris et al., 2014; Barton et al., 2019)."*

RC (3): I am concerned about the strategy used for applying the proposed approach. There is a spatial separation of the gauges used for performing the 1st step of the Kalman filter (KF) and the gauges used for performing the 2nd step of the KF. The 1st step used 14 TMD gauges that are within 100 km radius of the center of Tubma basin, from which only 1 is inside the basin, whereas the 2nd step used 16 citizen rain gauges that are within the basin. The 1st step was applied under the assumption that the radar rainfall bias correction factor is relatively stable in space. However, there were signs of spatial instability, e.g., the downscaling strategy G_{MP (where the hourly rainfall patterns of the 14 TMD gauges were averaged and used for downscaling) had the worst performance, as shown in Fig. 6(c). More obvious signs were shown in Figs. 8 and 9. Hence, I strongly recommend the authors improve the strategy for applying the approach.

AR: Thank you very much for your comments. To possibly reduce both temporal and spatial uncertainty in radar rainfall estimates, this would require a spatiotemporal varying bias model in the analysis, which may be more challenging given the limited number of gauges. Instead, in our model, any spatial variations are captured as random noise and quantified by a standard deviation (or variance), which is then used in the Kalman filter to properly weight and incorporate each observation. Kalman Filter has the benefit of accounting for uncertainties in the observations by weighting the contribution of measurements by their respective variances. The 1st step represents a real situation that there are only 14 TMD gauges available. The observation error variance reflecting spatial instability of the observed data at time t was thereafter calculated as the weight for correcting the predicted mean bias instead of using only the calculated mean field bias. Measurement updating is a key step to correct the spatially uniform mean field bias. To improve the accuracy of the KF, enhancing the density of rainfall data with lower observation error variance can play an important role for the 2nd step. The downscaling strategy $G_{MP}$ showed the highest median value of the estimated observation error variances of CKF-$G_{MP}$ (see Fig. 5(c) in the manuscript) leading to the worst performance of radar rainfall estimates.

**Specific comments/ technical corrections**

**1. Introduction**

RC (1.1) The description related to Citizen rain gauges/rainfall observations is too simple. Whereas, as indicated by the title "Citizen rain gauge improves hourly radar rainfall bias correction using a two-step Kalman filter", the readership might expect more information on the citizen rain gauges/rainfall observations from the introduction. Besides, it seems that the downscaling strategy is very important in this context. It might be necessary to give a brief review of the downscaling methods in scientific literature.

**AR:** We will insert a brief review of the citizen rain observations and the downscaling methods in the introduction between lines 64 to 66 as explained below.

*"In basins where a dense rainfall network is not available, Citizen Science (CS) offers a promising opportunity for enhancing the density of rainfall observations (Davids et al., 2019). With the popularization of smartphones and the availability of (relatively) simple and cheap equipment, abundant mobile applications and projects have been initiated in Water Resources Management to measure hydrometeorological variables like rainfall, water level height or water quality, as well as to ground-truth remotely sensed information on e.g. land use (Srivastra et al., 2018; Davids et al., 2019; See, 2019; Seibert et al., 2019). In the current study, we focus on rainfall measured by local citizens using a network of cheap rain gauges and a specially designed mobile application. Since citizen rainfall observations are typically provided at daily scale, a temporal downscaling*

*technique is needed for sub-daily applications. There has been a variety of temporal rainfall downscaling methods developed since the 1970s. The simplest approach is to distribute daily rainfall data to sub-daily resolutions by assuming uniform distributions. Stochastically generating sub-period data or spatially transferring finer resolution rainfall from a nearby rain gauge station to the study area based on spatial correlations are alternative approaches (Koutsoyiannis, 2003; Debele et al., 2007). However, these methods are not usually designed for real-time data disaggregation over large areas. Instead, a common approach for such scenarios is to downscale daily rainfall based on a simple fraction technique by considering the distribution patterns of high-resolution gridded rainfall products from radar or satellite sensors (Paulat et al., 2008; Wüest et al., 2010; Vormoor and Skaugen, 2013; Sideris et al., 2014; Barton et al., 2019)."*

**2. Study Area and Data**

RC (2.1): (Pg 5, L.125-126) I'm not clear with the sentence "A rain gauge with more than 80% of the dataset below the threshold was excluded from the analysis." Please explain.

AR: The meaning of the sentence "A rain gauge with more than 80% of the dataset below the threshold was excluded from the analysis." is that any rain gauge with more than 80% of the dataset being daily rainfall values of less than 0.5 mm was excluded from the analysis. To clarify, we replaced "80% of the dataset" by "80% of recorded rainfall amounts".

RC (2.2): (Pg 5, L.128-129) It is necessary to show the 14 rain gauges in Fig. 1 as well because the 14 rain gauges were intensively utilized in the following sections.

AR: Figure. 1 was modified to show all used gauges including the 14 rain gauges in Chonburi and Rayong provinces as shown by the following.

[Figure]

*Figure 1 (in the revised manuscript): Location of study domain, showing Thai Meteorological Department (TMD) automatic rain gauges, citizen rain gauges, Sattahip radar, and Tubma basin.*

RC (2.3): (Pg.5, L. 131-136) Concerning the low-cost citizen rain gauges, are they tipping buckets? What kind of equipment are they? Were the measurements compared with the TMD rain gauges (As shown in Fig. 1, there are 2 TMD gauges very close to them)? If so, how uncertain were the data? Or, in the other case, were the quality checks performed in an intra-group manner. It is hard to tell from the current description. Please make it clear.

AR: More information on the citizen rain gauge observation will be added in a new section "2.3.2 Citizen Rain Observation" in the revised manuscript as described in the first general comment. It is noted that in the revised manuscript Fig.1 will show total rain gauges and used rain gauge networks that passed the quality control. Consequently, there is only TMD rain gauge located in the Tubma basin and can be used for the measurement comparison between the citizen and TMD rain gauges.

**3. Methods**

RC (3.1): (Pg. 7, Table 1) Concerning the downscaling method "G_{Tubma}", instead of using only one TMD station within the basin (I noticed there is another one very close to the basin as shown in the small figure in Fig. 1), why not use this one as well. Besides, how variable are the hourly rainfall patterns across the basin. A comparison of the patterns from these two stations might provide useful information.

AR: Note that we have only one TMD rain gauge that passed quality control. This limitation then forced us to use only one TMD station for constructing the $G_{Tubma}$.

RC (3.2): (Pg. 14, L. 324): Please keep the term "KF-TMD" consistent with that shown in Table 2.

AR: We will revise the manuscript accordingly.

**4. Results and discussion**

RC (4.1): (Pg. 15, L. 360): "..., while R_P and G_{MP} showed larger variability over the day, ...". There might be a mistake. The R_P-based result has a sharp mean cumulative fraction curve and large variance in the downscaled hourly data if one observes the box plots, whereas the other has a flat mean cumulative fraction curve and small variance in the downscaled hourly data. In other words, these two are different in both respects. The problem comes down to how the variability is defined. Please explain.

AR: The variability in that context defines the variability in the downscaled hourly data, which both $R_P$ and $G_{MP}$ exhibited a lot of outliers from the boxplot in each hour. P15L360 will be rephrased for clearer understanding as explained below.

*"…, while $R_P$ and $G_{MP}$ showed larger variability in the downscaled hourly data with substantial outliers in the box plots, … "*

RC (4.2): (Pg. 18, Sect. 4.2.2) I am concerned about the validation scheme referred to as "KF - TMDD", especially about the separation of the gauges for performing KF/MFB and gauges for validation purposes. The bias correction was made for a large area with a radius of 100 km, whereas the validation was performed on a much smaller basin. I am afraid the separation makes the validation results less representative. Perhaps LOOCV for the 14 TMD gauge (as used in the "KF -TMD-H" strategy) is more persuading. If the purpose of using "KF -TMD-D" is to validate the bias correction performance within the basin, the authors should specify it explicitly (perhaps, also in Sect. 3.4). Anyhow, the separation mentioned above could be a bit problematic.

AR: For the KF -TMD-D case, we tried to imitate the actual situation if we have only 14 TMD rain gauges available for radar rainfall estimation in the Tubma basin, which bias adjustment method (MFB or KF) can produce more accurate radar rainfall in the basin. In Section 4.2.2, we, therefore, aimed to validate the bias correction performance within the basin by testing at all

the citizen rain gauge locations. Any problems arising from the gauge's separation could be solved by the CKF-D. Sections 3.4 and 4.2.2 will be modified in the revised manuscript as described below.

*The Sentence "To identify which approach between MFB and KF is more accurate for daily rainfall simulation in the Tubma basin if there are only 14 TMD rain gauges available, …" will be modified between Lines 327 to 328 of section 3.4. Furthermore, "Results associated with validating the bias correction performance within the Tubma basin were presented in Fig. 6 (b). This shows …" will be added to line 411 at the beginning of the paragraph.*

Edit to text: on line 415, remove "be"

RC (4.3): (Pg. 19, L. 437-438) "While there is a modest improvement in mean RMSE, the upper 75%-ile RMSE is reduced from about 6 mm/h to 3.5 mm/h. Mean MBE is changed from 0.1 to -0.15 mm/h." I found it hard to follow here. Please use the terms presented in Fig. 7(b) to refer to the results.

AR: These sentences will be adjusted as follows.

*"While there is a modest improvement in mean RMSE (see the black line connecting the mean values of the box plots from MFB-TMD to CKF-R$_P$), the upper 75%-ile RMSE is reduced from about 6 mm/h to 3.5 mm/h. Mean MBE is changed from 0.1 to -0.15 mm/h (see the red-dotted line connecting the mean values from MFB-TMD to CKF-R$_P$).*

RC (4.4): (Pg. 19, L. 439) Please correct "MFB-TMB" and "KF-TMB" to "MFB-TMD" and "KF-TMD", respectively.

AR: The mistake will carefully be corrected in the revised manuscript.

RC (4.5): (Pg 19-20, Sect. 4.3.4) I think the description of the results/figures could be organized in a more readable way. For example, there is a direct shift from the description of Fig. 7 to that of Fig. 8 (Pg.19, L. 440-444) without telling the readers that the content is related to Fig. 8, and there is a separation in the description of Fig. 7 (Pg. 19, L. 433- and Pg. 20, L. 451-).

*AR:* For clearer understanding, we will start section 4.3.4 with: "Results for this section are presented in Figs. 7, 8, and 9.". Additionally, *the following sentences, "Analysis of hourly rainfall hyetographs obtained from TMD rain gauge network compared with the validated rain gauge occurring in 3 different days were illustrated in Figure 8. It appears … ." will be inserted to line 444 in the revised manuscript.*

By the way, because Fig. 7 was the main results for section 4.3.4, while Fig.8 and 9 are supporting figures. Therefore, it is not easy to end all discussions related to Fig.7, followed by

Fig.8 and 9, respectively. Therefore, please allow separation in the description of Fig. 7 as mentioned above.

*Edits to text:*

*-line 444: replace "Figure 8 appears that the considerable RMSE occurs from…" by "Figure 8 shows considerable RMSE from…"*

*-line 446: replace "However, these RMSE can considerably reduce if…" by "However, these RMSE values decrease considerably if…"*

RC (4.6): (Pg. 19 at the end) I am not clear with the sentence " Figure 9 illustrates that hourly rainfall distribution patterns of TMD rain gauges in the 40-90 km range, influenced mainly by the southwest monsoon, appear to be more similar to the mean citizen rain gauge data than the range beyond 40 km." Please explain.

AR: In the mentioned sentence, we realize to change the words "40-90 km" to be "0-40 km" in the revised manuscript.

*Reference*

*Barton, Y., Sideris, I. V., Germann, U., and Martius, O.: A method for real-time temporal disaggregation of blended radar–rain gauge precipitation fields, 27, e1843, https://doi.org/10.1002/met.1843, 2020.*

*Davids, J. C., Devkota, N., Pandey, A., Prajapati, R., Ertis, B. A., Rutten, M. M., Lyon, S. W., Bogaard, T. A., and van de Giesen, N.: Soda Bottle Science-Citizen Science Monsoon Precipitation Monitoring in Nepal, Frontiers in Earth Science, 7, 10.3389/feart.2019.00046, 2019.*

*Debele, B., Srinivasan, R., and Parlange, J. J. A. i. W. R.: Accuracy evaluation of weather data generation and disaggregation methods at finer timescales, 30, 1286-1300, 2007.*

*Koutsoyiannis, D.: Rainfall disaggregation methods: Theory and applications, in, 1–23, 10.13140/RG.2.1.2840.8564, 2003.*

*Mobile Water Management: https://mobilewatermanagement.nl/, last access: 25 October 2020.*

Paulat, M., Frei, C., Hagen, M., and Wernli, H.: A gridded dataset of hourly precipitation in Germany: Its construction, climatology and application, Meteorologische Zeitschrift, 17, 719-732, 10.1127/0941-2948/2008/0332, 2008.

See, L.: A Review of Citizen Science and Crowdsourcing in Applications of Pluvial Flooding, 7, 10.3389/feart.2019.00044, 2019.

Seibert, J., Strobl, B., Etter, S., Hummer, P., and van Meerveld, H. J.: Virtual Staff Gauges for Crowd-Based Stream Level Observations, 7, 10.3389/feart.2019.00070, 2019.

Sideris, I. V., Gabella, M., Erdin, R., and Germann, U.: Real-time radar–rain-gauge merging using spatio-temporal co-kriging with external drift in the alpine terrain of Switzerland, 140, 1097-1111, https://doi.org/10.1002/qj.2188, 2014.

Srivastava, S., Vaddadi, S., and Sadistap, S.: Smartphone-based System for water quality analysis, Applied Water Science, 8, 10.1007/s13201-018-0780-0, 2018.

Vormoor, K. and Skaugen, T.: Temporal Disaggregation of Daily Temperature and Precipitation Grid Data for Norway, Journal of Hydrometeorology, 14, 989-999, 10.1175/JHM-D-12-0139.1, 2013.

Wüest, M., Frei, C., Altenhoff, A., Hagen, M., Litschi, M., and Schär, C.: A gridded hourly precipitation dataset for Switzerland using rain-gauge analysis and radar-based disaggregation, International Journal of Climatology, 30, 1764-1775, 10.1002/joc.2025, 2009.

---

## Referee Report (RR1)

**Citizen rain gauges improve hourly radar rainfall bias correction using a two-step Kalman filter - Review**

Anonymous Referee

November 5, 2021

**1 General comments**

Thank you for this manuscript.

Using citizen rain gauge data is an interesting approach and the authors analyzed the gathered data carefully and with a lot of different techniques. There seems to be a benefit in using these Citizen rain gauge data, so from my point of view, this study should be published.

I had some difficulties to understand where all the errors are coming from and how these errors are calculated. There are often some unreferenced assumptions on the error characteristics. Sometimes I would have wished to have a bit more insight into the error distributions of the different measurements.

There is also nothing mentioned on radar calibration errors. Most weather radars are not well calibrated and this obviously influences the estimation of $R$ via the $Z - R$ relationship. Is this measurement bias automatically canceled out by calculating an individual $Z - R$ relationship?

Regarding the KF and its mathematical formulation: I guess that this has been done properly, at least from my understanding. However, Kalman filters are especially great for multi-dimensional data sets with observation vectors being larger than 1. So I'm therefore not sure if averaging everything together and calculating one bias value is the best approach. But at least it is an approach that shows some nice results.

There are still quite some language issues and after a couple of page I got tired to correct them all. So there should be some polishing done on that.

Overall I think that the manuscript can be published after some corrections, which are detailed below.

**2 Detailed comments**

- Line 12: *of Sattahip radar station and gauge rainfall* → of Sattahip radar station, gauge rainfall...

- Line 27: *accuracy of flash flood estimates and warning.* → accuracy of flash flood estimates and warnings.

- Line 28: *However, weather radar provides indirect measurement of backscattered electromagnetic waves called radar reflectivity data (Z).* **rightarrow** However, a weather radar provides an indirect measurement of backscattered electromagnetic waves called radar reflectivity  (Z).

- Line 30: *ground-truthing by rain gauge data is required to calibrate the Z-R relationship ($Z = AR^b$)* → I'm not happy with this sentence. It's not a priori clear that just the Z-R relation needs to be calibrated. Biases between the radar rainfall intensity can also stem from radar calibration errors, i.e., from a bias in Z. Most radar operators do not alter the Z-R relationship at all, but just adjust the radar rainfall vs gauge rainfall bias (Sideris, 2014).

- Line 60: *variances affecting the mean field bias estimate*.

- Line 76: *However, these methods are not usually designed for real-time* → However, these methods are usually not designed for real-time

- Figure 1: I can't see the location of the radar in this map. It's indicated in the legend but not on the map (or hardly visible).

- Line 107: *The Tubma basin is covered within the range of Sattahip radar station.* → The Tubma basin is located within the coverage of the? Sattahip radar station.

- Line 107: *a beam width of 1.0°* → a half power beam width of 1.0°

- Line 109: *The radar reflectivity product is in a Cartesian* → The radar reflectivity product is provided in a Cartesian

- Line 110: *The Sattahip radar provides the CAPPI reflectivity data derived from the 2.5-km constant altitude plan position indicator (CAPPI).* → CAPPI derived from CAPPI: that's not a meaningful sentence.

- Line 114: *Additionally, the noise and hail effects were eliminated by setting reflectivity values below 15 dBZ to zero, and reflectivity values greater than 53 dBZ to 53 dBZ.* → I'm not sure if this is allowed. Could you give some reasoning on this method or a reference? If I use a Marhsall-Palmer relation ($Z = 200\,R^{1.6}$ then 100 mm of rain already give 55 dBZ. 100 mm are not unrealistic in the tropics, even higher values are possible. But probably Marshall-Palmer is not suitable in the tropics? And clipping below 15 dBZ might be ok, but reflectivity values below this value or not necessarily due to noise, especially not receiver noise. It might need a specification of what you mean by 'noise'.

- Line 137: *radar rainfall accumulation (mm/h)* → Accumulation is not in mm/h

- Line 141: *was validated against a second, independent dataset. Results found that a locally calibrated Z-R relationship that was used in this* → I don't understand this sentence. What is the $Z - R$ relation of the climatological dataset? What is the $Z - R$ relation of the locally calibrated dataset?

- Line 146: *These 15-min rain gauges*: → These rain gauge data have a temporal resolution of 15 minutes

- Line 148: *double mass curves method*: → What is this?

- Line 155: *based on spatial decorrelation analysis in the process.* → based on spatial decorrelation analysis for this? process.

- Line 180: RMSE does not tell you much about the bias between the TMD and the citizen gauge. Bias needs to be given as well.

- Line 188: *First, daily citizen...* → From line 126 onward you were using 'Firstly, Secondly...'.

- Line 206: I'm not sure if 'noise' is the right word here. KF accounts for measurement errors or uncertainties, but not for noise.

- Line 208: *different uncertainty characteristics, i.e., hourly...*

- Line 210: *Since the MFB (G/R ratio) is assumed to follow a log-normal distribution.* → Has this sentence a relationship to the previous sentence? And what is an ordinary KF scheme

- Line 300: *downscaled hourly citizen rain gauge data were used to back-calculate the hourly citizen rain gauges data* → I don't understand this.

- Figure 5: This is Kalman filter bias corrected rain accumulation data from gauges?

- Line 420: *%-ile* → percentile

- Line 508: *obviously appear steady light rainfall accumulation* → ?

- Figure 9: *Rainfall Depth* → probably not the right annotation.

**3   Kalman filter**

I guess I have understood your KF approach, but there are some questions left:

- $y_t$ and $z_t$ are your log-transformed bias observations from the TMD and the citizen gauge network, respectively, corresponding to $O_t$ in Eq. 10?

- Figure 4: In the second KF step you take the variance estimates $P_t$ from the previous KF step as a priori variances?

- You calculate the Kalman gain $K_t$ in the second step with these $P_t$ variances. From my point of view, this Kalman gain should be named differently, since $K_t$ is already used for the first step.

- Likewise, the equation (2) in the 2nd KF step in Figure 4 should be rewritten, since the value on the left side of the equation ($\hat{\beta}_t$) is an update of the bias value on the right side.

- Same is true for Eq. 3 in Figure 4, 2nd step: $P_t$ is an update of $P_t$ on the right side of the equation.

- I'm somehow irritated by the usage of these capital letters $K$ and $P$, which, at least for me, would represent matrices. In KF theory, $P$ then represents the error covariance matrix, but since you are using variances only it is somehow strange to go from $\sigma^2_{M_{z,t}}$ to $P_t$. Probably just an unimportant detail.

- Line 229 onwards: Error estimation of $\sigma^2_{M_{z,t}}$ and $\sigma^2_{M_{y,t}}$: I did not fully understand this. $\sigma^2_{O_t}$ is the (spatial?) variance of all stations? And why is Equation 19 necessary if you calculate $\sigma^2_{M_{z,t}}$ and $\sigma^2_{M_{y,t}}$ by the spatial variance of the individual stations?

---

## Author Response (AR4)

**Author Response to the referee comments on "Citizen rain gauge improves hourly radar rainfall bias correction using a two-step Kalman filter"**

*by Punpim Puttaraksa Mapiam [1*], Monton Methaprayun [1], Thom Adrianus Bogaard [2], Gerrit Schoups[2], Marie-Claire Ten Veldhuis[2]*

[1]*Department of Water Resources Engineering, Kasetsart University, PO Box 1032, Bangkok, 10900, Thailand*
[2]*Department of Water Management, Delft University of Technology, PO Box 5048, 2600 GA Delft, The Netherlands*

**Responses to referee #1**
RC: Referee comment
AR: Author response

**General comments**

My research interest is focused on citizen science weather observations (in particular rainfall monitoring), so whilst I am confident in reviewing that element of the paper, I am not well placed to comment on the technical aspects of radar rainfall bias correction.

The paper presents a novel approach to undertaking radar rainfall bias correction in an area with sparse official ground based rain gauges. The focus of the paper is on the methodology for bias correction, however there perhaps should be equal prominence for the selection of a method for the disaggregation of daily data to hourly, which seems as significant. Many researchers are sceptical about the accuracy of citizen science weather observations, and it would be beneficial to allay such concerns by providing more detail on how to undertake citizen science that generates data good enough to draw the conclusion made herein. Given the intrinsic difficulties in establishing the "true" rainfall amount due to spatial and temporal variation, the conclusions could have been more robustly supported by applying the methodology to an area where validation using official gauges could have been more effectively applied.

AR: Thanks for your comment, we agree that acquiring good quality citizen observations is a challenge and we put considerable effort into citizen engagement during the measurement campaign and in data quality control afterwards. Since the focus of this paper is on bias correction, we chose not to include details on the citizen science campaign in the paper. We agree that conducting a similar campaign in an area well supported by official gauges, could have led to more robust results, however we believe that the added value of citizen science lies precisely in the type of region where we conducted our study. Regions supported by dense, high-quality rain gauge networks (like EU, US) represent different conditions for implementation of citizen science and generally different rainfall climatologies.

**Specific comments**

RC (1): A detailed methodology relating to the Kalman Filter is provided, but it overshadows some of the more basic and fundamental details on which the paper is based. There is no justification provided on the selection of the Tubma basin as the research area. More information would be appreciated in "Section 2.1 Study Area" describing the nature of the basin and the climatic characteristics. Lines 456 – 459 highlight some of the limitations of the study area, which I feel deserve more prominence. The location seems like a difficult place to study and have confidence in the results due to the limited opportunity for external validation via TMD gauging.

AR: More information describing the reason why the Tubma basin was selected as a research area was added in section 2.1 between lines 91 and 98 of the revised manuscript as described below.

*"In Figure 1, we show the climatological variation across the study area and its surroundings, based on 30-year (1987-2017) annual mean rainfall from the network of 311 daily rain gauges owned by the TMD and situated within 200 km range from the Tubma basin. Spatial rainfall patterns were generated by inverse distance squared (IDS) between the gauge locations. The map shows that while there is small gradient in mean annual rainfall (1,100 to 1,700 mm mean annual rainfall) across the area of Rayong and Chonburi provinces (within 90 km from the study area), changes are more pronounced when the distance exceeds the beyond the 90 km boundary, especially to the east of the study area. This is because these areas are affected differently by the southwest monsoon. Consequently, evaluating the effectiveness of bias correction techniques were carried out within 90 km range from the study area with similar climatology."*

[Figure]

*Figure 1: Climatological spatial rainfall distribution within 200 km ranges from the centroid of the Tubma basin calculated from 30-year average annual rainfall data of 311 daily rain gauge network by using IDS method.*

RC (2): Given the climatic variation across the study area noted in Section 4.3.4, justification on the use of a 100Km radius for evaluation would be welcome.
AR: The 100 km radius refers to the coverage area of the Sattahip radar. The correct radar range is actually 90 km, as indicated in section 4.3.4, we carefully made sure to correct all references to 100 km by 90 km in the revised manuscript.

RC (3): There is no explanation of how citizen science observations were made, how participants were recruited and trained, or discussion of citizen science limitations either in general or encountered in this research.

AR: For more precise understanding about the citizen rain observation, lines between 130 to 138 in the old manuscript were rewritten as a new section "2.3.2 Citizen Rain Observation" as described below.

*"2.3.2 Citizen Rain Observation*

*Out of the total TMD rain gauge network, only one rain gauge is located in the Tubma basin. To increase the density of the rain gauge network in the basin, low-cost citizen rain gauges were implemented in this study to better capture spatial heterogeneity of rainfall in the basin. Sixteen citizen rain gauges were installed (Fig. 2) with local residents taking daily measurements. The additional 16 citizen rain gauges with one station located at the same place of the existing TMD gauge increased the density of rain gauges in the Tubma basin to 1 gauge/12 km². The citizen observations were made by installing a cone-shape transparent plastic rain gauge which is standardly used in South Africa (see Fig. S1) with a diameter of 5 inches and capacity of 100 mm in open space area* around a school, Monasteries, bridge or other building. Mobile application *developed by Mobile Water Management (MWM) (Mobile Water Management, 2020), the Netherlands, was used to record rainfall data for each rain gauge on a daily basis. The application has a an easily accessible and user-friendly interface where participants simply fill in the observed rainfall amount, take a photo of the rain gauge and upload this to the application. The photo and the rainfall data, together with the measuring location and time, are automatically stored in the database. Photos are used for visual validation of the recorded rainfall depth to eliminate errors.*

*In this study, participants were recruited amongst government officers, teachers, and local residents living close to the stations and were trained to take measurements at around 7 a.m. daily according to the TMD standards. Quality of the collected data was assured by the high photo resolution for double-checking the observations and strict requirements on measurement times to be consistent with the same standard of TMD for daily rainfall recording. Note that the maximum rainfall for the citizen gauges is 100mm/day. Validation of the cone-shaped citizen gauges was conducted based on a citizen gauge co-located with an automatic TMD gauge located in the Tubma basin, during August – October 2019. The citizen gauge installed at the same location R3 (Fig. S2) as a TMD gauge showed good similarity with a random RMSE of 5.5 mm.*

*Quality control consisted of screening all citizen rain gauge data for errors and inconsistencies using double mass curves. If citizen rain gauges reported >100mm/day rainfall (maximum capacity of the citizen rain gauge) this data was excluded from the analysis. If days with no-rainfall data were found from all citizen rain gauges, the bias correction of that day was discarded from the dataset. By considering the data selection criteria, rainfall data recorded during August–October 2019 with rainy days, more than 80% of the whole period for the bias adjustment process was then used for further analysis."*

[Figure]

*Figure S1: (in supplementary information to the revised manuscript) An example of* installing a cone-shape transparent plastic citizen rain gauge *at location R.22, Map Tong school.*

[Figure]

*Figure S2: (in supplementary information to the revised manuscript) Daily rainfall depth comparison between the TMD and citizen rain gauge at location R.3 during August – October 2019.*

RC (4): There do appear to be interesting results from the citizen science gauges that warrant further discussion. E.g, Fig.9 (a) indicates a range in the monthly cumulative rainfall from citizen science gauges of ~170 – 400mm and multiple reporting gaps, whereas (b) the TMD gauge range is ~150 – 260mm. A map with the gauges identified and some detail on elevation or climatic region could be included.

AR: *Note that Fig. 9 in the old manuscript was changed to Fig. 10 in the updated paper. Indeed, Fig.10 (a) and (b) show that one of the citizen gauges (R.18) collected cumulative monthly rainfall higher than the range of the other citizen and TMD rain gauges. This is associated with a storm event that occurred in September 2019, with the storm center over R.18, while the surroundings received appreciably less rainfall. Figure S4 (this figure was added in the supplementary of the revised manuscript) shows the radar reflectivity field at 13.00 h on 22 September 2019, during the peak of the storm, confirming the heavy rainfall affecting gauge location R.18. This shows the citizen gauge network is able to capture local storm features thanks to the high density of the network.* The multiple reporting gaps visible in Fig. 10 (a) are caused by time errors in the observations submitted by local residents were removed from the analysis (as explained in new section 2.3.2 of the revised manuscript). This information and the associated figure were provided in Supplementary Information of the revised *manuscript.*

[Figure]

*Figure S4: Overlapping between citizen rain gauges network and spatial radar reflectivity data on 22 September 2019 at 13.00 hour*

RC (5) If some detail on the application of existing methods could be moved to "Supplementary Information", it would allow more space for consideration of the citizen science element.

*AR: Since the focus of our study is on radar bias adjustment, we have not elaborated in detail on the citizen observations network separately. We believe it would distract from the focus of the study, which is on the 2-step Kalman-filter bias adjustment methodology. This is the most novel element of the work.*

**Technical comments**

RC (5): General point – Although common practice I found the use of acronyms made the paper hard to follow at times e.g. Section 4.2

AR: Because we have many approaches for comparison and would like to make the paper short, we then use the acronyms to drive the story.

RC (6): Figure 1 – Identification of citizen science gauges

AR: Figure. 1 was modified to Figure 2 in the revised manuscript to show all used gauges including the 14 rain gauges in Chonburi and Rayong provinces as shown by the following. The identification of citizen rain gauges was also added in the figure.

[Figure]

*Figure 2 (in the revised manuscript): Location of study domain, showing Thai Meteorological Department (TMD) automatic rain gauges, citizen rain gauges, Sattahip radar, and Tubma basin.*

RC (7): Line 164 – replace full stop with comma
AR: We checked the format and replaced them correctly in the revised manuscript.

RC (8): Line 254 – Point 1 is not clear about the timing of data, could this be rephrased?
AR: The point 1 of P10L254 was rephrased according to the comment between lines 295 and 297 in the revised manuscript as appeared below.

"1) Since the citizen rain gauge data were received at the last hour of day $i$, before receiving the downscaled hourly citizen rain gauge data of day $i$, the ordinary KF and observed hourly data of TMD were used to predict and correct the hourly bias adjustment factor of the day $i$"

RC (9): Line 270 – replace "In case" with "Where"
AR: This correction was implemented in the revised manuscript.

RC (10): Figure 4 – a grid may make reading easier, or may be too cluttered?
AR: Figure 4 was modified as shown below and was placed in the revised manuscript.

[Figure]

RC (11): Figure 9 – The colours are indistinct for the different gauges, the x ticks could be 'day' and the figures in general are very small making them hard to read on the page

AR: Thank for the suggestion, the figure was adjusted as the following.

[Figure]

*Figure 9 (in the revised manuscript): Comparison of mass curve of hourly rainfall among various rain-gauge locations (a) the citizen rain gauges located in the Tubma basin (b)TMD rain gauges within 0-40 km radius from the Tubma basin (c) TMD rain gauges within 40-90 km radius from the Tubma basin.*

Reference

Mobile Water Management: https://mobilewatermanagement.nl/, last access: 25 October 2020.

**Author Response to the referee comments on "Citizen rain gauge improves hourly radar rainfall bias correction using a two-step Kalman filter"**

*by Punpim Puttaraksa Mapiam [1*], Monton Methaprayun [1], Thom Adrianus Bogaard [2], Gerrit Schoups[2], Marie-Claire Ten Veldhuis[2]*

[1]*Department of Water Resources Engineering, Kasetsart University, PO Box 1032, Bangkok, 10900, Thailand*
[2]*Department of Water Management, Delft University of Technology, PO Box 5048, 2600 GA Delft, The Netherlands*

**Responses to referee #2**

RC: Referee comment
AR: Author response

**Major comments**

RC (1) In Chapter 2.3.1 you describe that all TMD rain gauges "with more than 80% of the dataset below the threshold was excluded" (P5L125). Since the threshold of 0.5 is equal to the tipping bucket resolution, you excluded all station with an hourly $p0$ (probability of value zero in the dataset) above 0.8. On P16L363 you mention that "the others having a period of heavy rainfall around 4-5 hours a day" which leads to a $p0$ of around 0.8. This applied filter excluded approximately 55% of the stations which is a lot due to the fact that the data is provided by the Thai Meteorological Department. Can you please provide a figure showing all TMD gauges on the one side and the used gauges on the other side?

AR: thanks for pointing this out, we realize the phrasing may have created confusion.
The rain gauge selection was based on a threshold $p0$ of 0.8 at the daily scale. Only 134 out of 297 stations passed this test. Actually, many of the rejected stations recorded zero values throughout most of the study period. Of the remaining stations, many reported heavy rainfall during 4-5 hours of the day (which indeed leads to a $p0$ of 0.8 at the hourly scale), as stated in P16L363.

We propose to rephrase the sentence (P5L125) as follows, to avoid confusion:
*"Rain gauges with more than 80% of the recorded rainfall amounts below the 0.5 mm threshold at daily scale were excluded from the analysis. It turns out that many of these faulty gauges recorded zero rainfall throughout most of the study period."*

We also modified Figure 1 (changed to be Figure 2 in the revised manuscript), to distinguish the selected rain gauges:

[Figure]

*Figure 2 (modified for revised manuscript): Location of study domain, showing Thai Meteorological Department (TMD) automatic rain gauges, citizen rain gauges, Sattahip radar, and Tubma basin.*

RC (2) Was there a specific reason why the validation was done with two different data sets? This means that a comparison between the daily and the hourly data set is not meaningful.

AR: The question we aimed to answer in this manuscript is to what extent citizen rainfall observations improve the accuracy of hourly radar rainfall estimates (by using a 2-step Kalman filter approach). Since the citizen rainfall observations are made at daily scale, we need a downscaling procedure to match the radar data scale. The downscaling step introduces additional uncertainty, that's why we chose to validate the bias correction results not only at hourly but also at daily time-scale. The validation procedures are explained in detail in Section 3.4 and summarized in Table 2.

RC (3) Was there a specific reason why the boundary between "near" stations and the "far" stations was chosen at 40km in Case 4? I think that the boundary could have been anywhere outside the Tubma basin and there would have been a significant change in the results.

AR: To clarify this point, we added the following information and the Fig. S3 in section S2 of the supplementary Information.

*"We chose the 40 km separation boundary to achieve an equal number of gauges in the "near" and "far" groups. Following the referee's comment, we investigated RMSE between gauge rainfall and radar rainfall without bias adjustment ($RMSE_{No-Bias}$) and with the CKF-R$_P$ ($RMSE_{CKF-RP}$) for individual stations (located at distances 5 − 80 km from the Tubma basin). We computed the percentage improvement in radar rainfall estimates using CKF-R$_P$ compared to No-Bias at each rain gauge, indicating the relative errors changing with distance from the Tubma basin (Fig. S3). Figure S3 shows that the improvement percentage of using CKF-R$_P$ tends to reduce with increasing distance from the Tubma basin, where the citizen gauges are located. The percentage reduction gradually decreases beyond a distance of about 40 km.*

*Note that the gauge 4780010 situated at the nearest range is expected to provide the best improvement, however, the Sattahip radar temporarily stopped measuring for 3 hours (during 16:00 h - 19:00 h of 24 September 2019) which is associated with a heavy storm's center was only at the gauge 4780010. This leads to significant degradation of the radar rainfall performance. Furthermore, the lower percentage improved for station 4780005 is associated with a localized heavy rainfall event that was recorded only at this location, negatively affecting its performance as a representative station for bias correction."*

[Figure]

*Figure S3: Alteration of the percentage improvement in radar rainfall estimates of using CKF-R$_P$ compared to No-Bias at each rain gauge with different distance from the Tubma basin.*

**Minor comments**

RC (1) Title: "Citizen rain gauge improves ..." → "Citizen rain gauges improve ..."
*AR: Thanks for correcting this, we revised the manuscript accordingly.*

RC (2) Title, P1L11, P1L15, P1L20, P2L55, ...: Kalman filter, Kalman Filter, ... please be consistent in using capital letters or not.
*AR: This mistake was corrected in the revised manuscript by using only the Kalman filter.*
RC (3) P2L32: "the A and b parameters" → "the parameters A and b"
RC (4) P2L43, P2L44: "... the reference A parameters (...) to the A parameters for sub-daily resolutions." → please reformulate
RC (5) P2L61 and P2L62: "They found ..." → please reformulate
*AR: Comments number (3) to (5) were reformulated in the revised the manuscript.*

RC (6) P3L66: "...typically provided at daily scale" → Please define citizen rain observation more precisely so that it is clear that this refers to soda bottles, for example.
*AR: For more precise understanding about the citizen rain observation, line between 130 to 138 in the old manuscript were rewritten as a new section "2.3.2 Citizen Rain Observation" as described below.*

*"2.3.2 Citizen Rain Observation*
*Out of the total TMD rain gauge network, only one rain gauge is located in the Tubma basin. To increase the density of the rain gauge network in the basin, low-cost citizen rain gauges were implemented in this study to better capture spatial heterogeneity of rainfall in the basin. Sixteen citizen rain gauges were installed (Fig. 1) with local residents taking daily measurements. The additional 16 citizen rain gauges with one station located at the same place of the existing TMD gauge increased the density of rain gauges in the Tubma basin to 1 gauge/12 km². The citizen observations were made by installing a cone-shape transparent plastic rain gauge which is standardly used in South Africa (see Fig. S1) with a diameter of 5 inches and capacity of 100 mm in open space area around a school, Monasteries, bridge or other building. Mobile application developed by Mobile Water Management (MWM) (Mobile Water Management, 2020), the Netherlands, was used to record rainfall data for each rain gauge on a daily basis. The application has a an easily accessible and user-friendly interface where participants simply fill in the observed rainfall amount, take a photo of the rain gauge and upload this to the application. The photo and the rainfall data, together with the measuring location and time, are automatically stored in the database. Photos are used for visual validation of the recorded rainfall depth to eliminate errors.*

*In this study, participants were recruited amongst government officers, teachers, and local residents living close to the stations and were trained to take measurements at around 7 a.m. daily according to the TMD standards. Quality of the collected data was assured by the high photo*

*resolution for double-checking the observations and strict requirements on measurement times to be consistent with the same standard of TMD for daily rainfall recording. Note that the maximum rainfall for the citizen gauges is 100 mm/day.*

*Validation of the cone-shaped citizen gauges was conducted based on a citizen gauge co-located with an automatic TMD gauge located in the Tubma basin, during August – October 2019. The citizen gauge installed at the same location R3 (Fig. S2) as a TMD gauge showed good similarity with a random RMSE of 5.5 mm.*

*Quality control consisted of screening all citizen rain gauge data for errors and inconsistencies using double mass curves. If citizen rain gauges reported >100mm/day rainfall (maximum capacity of the citizen rain gauge) this data was excluded from the analysis. If days with no-rainfall data were found from all citizen rain gauges, the bias correction of that day was discarded from the dataset. By considering the data selection criteria, rainfall data recorded during August–October 2019 with rainy days, more than 80% of the whole period for the bias adjustment process was then used for further analysis."*

[Figure]

*Figure S1: (in supplementary information to the revised manuscript) An example of* installing a cone-shape transparent plastic citizen rain gauge *at location R.22, Map Tong school.*

[Figure]

*Figure S2: (in supplementary information to the revised manuscript) Daily rainfall depth comparison between the TMD and citizen rain gauge at location R.3 during August – October 2019.*

RC (7) P3L75: 101°17'51" → : 101°17'51"E
RC (8) P3L75: "of approximately 197km²" → "of 197km²" or "of approximately 200km²"
*AR: Comments number (7) to (8) were adjusted in the revised the manuscript.*

RC (9) P3L83, P3L84: "240 km x 240 km (…) 0.6x0.6km" → Please stay with one style
*AR: We edited 240 km x 240 km to 240x240 km.*

RC (10) P3L84: "… spatial resolution and 6-min temporal resolution"  → "spatial and 6-min temporal resolution"
RC (11) P3L90: "3 datasets"→ "three datasets"
*AR: Comments number (10) to (11) were rewritten in the revised the manuscript.*

RC (12) P3L90-92: What happened between 2014 and 2019?
*AR: Since, the citizen rain gauges were installed in the Tubma basin in rainy season of 2019. We then collected more data in 2013 and 2014 for the analysis of Z-R calibration and validation, while keep the data in 2019 for the bias correction development and evaluation.*

RC (13) P3L121: Is it "Thai Meteorological Department" or "Thailand Meteorological Department"?

*AR: It is Thai Meteorological Department officially. Line 10 and 158 in the revised manuscript was corrected accordingly.*

RC (14) P4L102 and L103: "1-hour" → "hourly"
RC (15) P4L103: "A parameters" → "parameters A"
RC (16) P4L104: "b exponent" → "exponent b"
RC (17) P4L106: "b parameter" → "parameter b"
RC (18) P4L108: "mean absolute error" → "MAE" (defined on P4L104)
*AR: Comments number (14) to (18) were corrected in the revised the manuscript.*

RC (19) P4L110: Since NG,t seems not be equal to N over the entire period T, the Equation is incorrect. Which equation was used in your calculations?

*AR: Thanks for pointing this out. $N_{G,t}$ was changed to $N_G$ to represent the number of rain gauges and N was replaced with multiplying between T and $N_G$ to represent total number of data pairs used in the calculation. The equation was revised including related description of each variable as described below in the revised manuscript.*

*"*

$$MAE = \frac{1}{TN_G} \sum_{t=1}^{T} \sum_{i=1}^{N_G} |G_{i,t} - R_{i,t}|$$

*where $G_{i,t}$ is the gauge rainfall (mm/h) at gauge i for hour t, $R_{i,t}$ is the radar rainfall accumulation (mm/h) at the pixel corresponding to the $i^{th}$ rain gauge for hour t, $N_G$ is the number of rain gauges, and T is the number of time period used in the calculation."*

RC (20) P4L122: "have tipping-bucket sizes of 0.5mm" → something like "have a resolution of 0.5mm"

*AR: We replaced "have tipping-bucket sizes of 0.5mm" with "have a resolution of 0.5mm"*

RC (21) P4L130: "in the 197km² Tubma basin" → "in the Tubma basin"
*AR: The manuscript was adjusted according to the comment.*

RC (22) P4L133: 1 gauge/15km²: How did you calculate that? One TMD station + 16 citizen rain gauges = 17 stations. 197km² / 17 stations → 1 gauge/~12km²

*AR: There was a mistake because of using a wrong number of citizen rain gauges in the calculation. The correct total number of rain gauges is 16 stations since 1 station of citizen rain gauge was located at the same place of the existing TMD. The density was changed to be 197 km² / 16 stations → 1 gauge/~12km² on line 166 in the revised manuscript.*

RC (23) P6Table1: "Code description" → "Code Description"
*AR: This was corrected in the revised the manuscript.*

RC (24) P7L158: Is MFB the abbreviation for "Mean field bias" or "Mean field bias adjustment"? If the latter, I would suggest MFBA, since MFB is a common abbreviation for mean field bias. If the former, use this abbreviation (P7L162, P7L168).
*AR: MFB is the abbreviation for Mean field bias, so P7L162 and P7L168 were carefully corrected.*

RC (25) P7L163f: "…Smith and Krajewski (1991), Anagnostou et al. (1998), and Seo et al. (1999), Chumchean et al. (2006), Kim and Yoo, (2014), Shi et al. (2018)." → "…Smith and Krajewski (1991), Anagnostou et al. (1998), Seo et al. (1999), Chumchean et al. (2006), Kim and Yoo, (2014), and Shi et al. (2018)."
RC (26) P7L177: "The radar bias at time t …" → "The radar bias at time $t$ …")
RC (27) P10L254: "… day i …" → "… day $i$ …"
RC (28) P11Figure3: "Is hourly TMD data at hour t available?" → "Is hourly TMD data $y$ at hour $t$ available?"
RC (29) P11Figure3: "Is hourly citizen rain gauge data at hour t available?" → "Is hourly citizen rain gauge data $z$ at hour $t$ available?"
RC (30) P12L287: "… hour t …" → "… hour $t$ …"
RC (31) P12L290: "… time t …" → "… time $t$ …"
*AR: All the mistakes mentioned in the comment numbers (25)-(31) were carefully corrected in the revised manuscript.*

RC (32) P12L295: "Kalman Filter" → "KF" From here on, no more explicit mention for missing abbreviations. Please check independently in the following.
*AR: We thank for pointing this out, the abbreviations were gently checked.*

RC (33) P13L303: "1 TMD" → "one TMD"
*AR: This suggestion was implemented in the revised manuscript.*

RC (34) P13L306: "randomly": Was the LOOCV done for all 16 citizen rain gauges or did you randomly sample 16 times? If the former, it is not really randomly.
*AR: The LOOCV was done for all the 16 citizen rain gauges. Therefore, P13L306 was rewritten between lines 347 and 349 in the revised manuscript as described below.*
*        "For each round of cross-validation, one rain gauge was left out for validation and the remaining rain gauges were used as the calibration rain gauges to calculate the bias adjustment factor using the three different techniques."*

RC (35) P13L306: "3 different techniques, and 1 rain gauge" → "three different techniques, and one rain gauge"

RC (36) P13L310: "… gauge i …" → "… gauge *i* …"

RC (37) P14L324: see P13L306

RC (38) P14L325: "1 TMD" → "one TMD"

RC (39) P14L327f: "fourteen TMD" → "14 TMD"

RC (40) P14L338: "(leave 1 TMD out)" → "(leave one TMD out)"

RC (41) P14L345 and L347: "Kalman Filter" → "KF"

RC (42) P15L347: "r1 parameter" → "parameter r1"

RC (43) P15L350: "over the same time-series period" → "over the same period"

RC (44) P16L372: "figure 5" → "Fig.5"

RC (45) P17L391: "observationss" → "observations"

RC (46) P17L393: "are based on 4" → "are based on four"

*AR: All the mistakes mentioned in the comment numbers (35)-(46) were corrected in the revised manuscript.*

RC (47) Figure 6 and 7: Since RMSE and MBE have different limits (0 to infinity vs. -infinity to infinity), it does not make sense to put both assessment measures on one graph.

*AR: Regarding these figures, we intended to compare RMSE and MBE across different techniques corresponding to each scenario of the study. We then designed the graph to be readable from two independent y-axes. The primary y-axis on the left indicates the RMSE and the secondary y-axis on the right indicates the MBE.*

RC (48) P18L414: "respectively)" → "respectively)."

RC (49) P18L432: "Figure 7 (b) and Fig. 7 (c)" → "Figure 7 (b) and (c)"

RC (50) P18L435: see P18L432

*AR: All the mistakes mentioned in the comment numbers (48)-(50) were corrected in the revised manuscript.*

RC (51) Figure 9: Please increase the font size and add a grid.

*AR: Figure 9 was adjusted in the revised manuscript as follows.*

[Figure]

*Figure 9 (in the manuscript): Comparison of mass curve of hourly rainfall among various rain-gauge locations (a) the citizen rain gauges located in the Tubma basin (b)TMD rain gauges within 0-40 km radius from the Tubma basin (c) TMD rain gauges within 40-90 km radius from the Tubma basin.*

RC (52) P22L490: "(August-October, 2019)" → "(August-October 2019)"

*AR: This suggestion was implemented in the revised manuscript.*

RC (53) P22L495 and L497: "Kalman filter" → "KF" or "Kalman Filter"

*AR: Let us use "Kalman filter" in the conclusion section.*

**Author Response to the referee comments on "Citizen rain gauge improves hourly radar rainfall bias correction using a two-step Kalman filter"**

*by Punpim Puttaraksa Mapiam [1*], Monton Methaprayun [1], Thom Adrianus Bogaard [2], Gerrit Schoups[2], Marie-Claire Ten Veldhuis[2]*

[1]*Department of Water Resources Engineering, Kasetsart University, PO Box 1032, Bangkok, 10900, Thailand*
[2]*Department of Water Management, Delft University of Technology, PO Box 5048, 2600 GA Delft, The Netherlands*

**Responses to referee #3**

RC: Referee comment
AR: Author response

**General comments**

The sparsity of the conventional rain gauge network is a limiting factor for radar rainfall bias correction. Citizen rain gauges offer an opportunity to provide additional information at higher spatial resolution. This paper performs radar rainfall bias adjustment using two sources of rainfall information: observations measured by TMD (Thailand Meteorological Department) and daily citizen rainfall observations. The radar rainfall bias correction factor was sequentially updated based on the TMD data and downscaled hourly citizen data via a two-step Kalman filter. The results showed that citizen rain gauges improved the performance of radar rainfall bias adjustment especially for small ranges from the center of these gauges.

My reading through this manuscript suggests that the following three issues should be dealt with or stressed more clearly in the paper:

RC (1): The citizen rain gauges are very important in the context of this paper. However, the relevant introduction is too simple. What kind of equipment are they? How uncertain are the measurements compared to the official TMD data? Please add more information on them.

AR: For more in-depth description of the citizen rain observation, lines 130 to 138 in the old manuscript were rewritten as a new section "2.3.2 Citizen Rain Observation" as described below.
*"2.3.2 Citizen Rain Observation*

*Out of the total TMD rain gauge network, only one rain gauge is located in the Tubma basin. To increase the density of the rain gauge network in the basin, low-cost citizen rain gauges were implemented in this study to better capture spatial heterogeneity of rainfall in the basin. Sixteen citizen rain gauges were installed (Fig. 1) with local residents taking daily measurements. The additional 16 citizen rain gauges with one station located at the same place of the existing TMD gauge increased the density of rain gauges in the Tubma basin to 1 gauge/12 km[2]. The citizen observations were made by installing a cone-shape transparent plastic rain gauge which is standardly used in South Africa (see Fig. S1) with a diameter of 5 inches and capacity of 100 mm in open space area around a school, Monasteries, bridge or other building. Mobile application*

*developed by Mobile Water Management (MWM) (Mobile Water Management, 2020), the Netherlands, was used to record rainfall data for each rain gauge on a daily basis. The application has a an easily accessible and user-friendly interface where participants simply fill in the observed rainfall amount, take a photo of the rain gauge and upload this to the application. The photo and the rainfall data, together with the measuring location and time, are automatically stored in the database. Photos are used for visual validation of the recorded rainfall depth to eliminate errors.*

*In this study, participants were recruited amongst government officers, teachers, and local residents living close to the stations and were trained to take measurements at around 7 a.m. daily according to the TMD standards. Quality of the collected data was assured by the high photo resolution for double-checking the observations and strict requirements on measurement times to be consistent with the same standard of TMD for daily rainfall recording. Note that the maximum rainfall for the citizen gauges is 100 mm/day.*

*Validation of the cone-shaped citizen gauges was conducted based on a citizen gauge co-located with an automatic TMD gauge located in the Tubma basin, during August – October 2019. The citizen gauge installed at the same location R3 (Fig. S2) as a TMD gauge showed good similarity with a random RMSE of 5.5 mm.*

*Quality control consisted of screening all citizen rain gauge data for errors and inconsistencies using double mass curves. If citizen rain gauges reported >100mm/day rainfall (maximum capacity of the citizen rain gauge) this data was excluded from the analysis. If days with no-rainfall data were found from all citizen rain gauges, the bias correction of that day was discarded from the dataset. By considering the data selection criteria, rainfall data recorded during August–October 2019 with rainy days, more than 80% of the whole period for the bias adjustment process was then used for further analysis."*

[Figure]

*Figure S1: (in supplementary information to the revised manuscript) An example of* installing a cone-shape transparent plastic citizen rain gauge *at location R.22, Map Tong school.*

[Figure]

*Figure S2: (in supplementary information to the revised manuscript) Daily rainfall depth comparison between the TMD and citizen rain gauge at location R.3 during August – October 2019.*

RC (2): The strategy used to downscale the daily measured citizen rainfall observations into hourly temporal resolution is essential in this context. However, there were no references on this topic. Four relatively simple downscaling strategies were tested, and the results showed that the one that utilized gauge-respective radar rainfall patterns performed the best. Yet I do think there is room for improvement.

AR: The focus of the paper is on developing a new algorithm called two-step Kalman filter using two sources of observed rain gauge data. The strategy for combing TMD and citizen rain gauge datasets in the second measurement updating of the Kalman filter was a core of the development. Simulating different hourly citizen dataset with 4 downscaled strategies was carried out to investigate the sensitivity of the modified KF approach. As a result, in this study, we decided to use a simple fraction method for disaggregating daily rainfall to hourly values. A similar strategy has been applied in several previous applications (Paulat et al., 2008; Wüest et al., 2010; Vormoor and Skaugen, 2013; Sideris et al., 2014; Barton et al., 2019). Additional literature review on downscaling techniques was provided in the introduction section between the lines 72 and 79 of the revised manuscript as described below.

*"There has been a variety of temporal rainfall downscaling methods developed since the 1970s. The simplest approach is to distribute daily rainfall data to sub-daily resolutions by assuming uniform distributions. Stochastically generating sub-period data or spatially transferring finer resolution rainfall from a nearby rain gauge station to the study area based on spatial correlations are alternative approaches (Koutsoyiannis, 2003; Debele et al., 2007). However, these methods are not usually designed for real-time data disaggregation over large areas. Instead, a common approach for such scenarios is to downscale daily rainfall based on a simple fraction technique by considering the distribution patterns of high-resolution gridded rainfall products from radar or satellite sensors (Paulat et al., 2008; Wüest et al., 2010; Vormoor and Skaugen, 2013; Sideris et al., 2014; Barton et al., 2019)."*

RC (3): I am concerned about the strategy used for applying the proposed approach. There is a spatial separation of the gauges used for performing the 1st step of the Kalman filter (KF) and the gauges used for performing the 2nd step of the KF. The 1st step used 14 TMD gauges that are within 100 km radius of the center of Tubma basin, from which only 1 is inside the basin, whereas the 2nd step used 16 citizen rain gauges that are within the basin. The 1st step was applied under the assumption that the radar rainfall bias correction factor is relatively stable in space. However, there were signs of spatial instability, e.g., the downscaling strategy G_{MP (where the hourly rainfall patterns of the 14 TMD gauges were averaged and used for downscaling) had the worst performance, as shown in Fig. 6(c). More obvious signs were shown in Figs. 8 and 9. Hence, I strongly recommend the authors improve the strategy for applying the approach.

AR: Thank you very much for your comments. To possibly reduce both temporal and spatial uncertainty in radar rainfall estimates, this would require a spatiotemporal varying bias model in the analysis, which may be more challenging given the limited number of gauges. Instead, in our model, any spatial variations are captured as random noise and quantified by a standard deviation (or variance), which is then used in the Kalman filter to properly weight and incorporate each observation. Kalman Filter has the benefit of accounting for uncertainties in the observations by weighting the contribution of measurements by their respective variances. The 1$^{st}$ step represents a real situation that there are only 14 TMD gauges available. The observation error variance reflecting spatial instability of the observed data at time t was thereafter calculated as the weight for correcting the predicted mean bias instead of using only the calculated mean field bias. Measurement updating is a key step to correct the spatially uniform mean field bias. To improve the accuracy of the KF, enhancing the density of rainfall data with lower observation error variance can play an important role for the 2$^{nd}$ step. The downscaling strategy G$_{MP}$ showed the highest median value of the estimated observation error variances of CKF-G$_{MP}$ (see Fig. 5(c) in the manuscript) leading to the worst performance of radar rainfall estimates.

**Specific comments/ technical corrections**

**1. Introduction**

RC (1.1) The description related to Citizen rain gauges/rainfall observations is too simple. Whereas, as indicated by the title "Citizen rain gauge improves hourly radar rainfall bias correction using a two-step Kalman filter", the readership might expect more information on the citizen rain gauges/rainfall observations from the introduction. Besides, it seems that the downscaling strategy is very important in this context. It might be necessary to give a brief review of the downscaling methods in scientific literature.

**AR:** We inserted a brief review of the citizen rain observations and the downscaling methods in the introduction section between the lines 65 and 79 in the revised manuscript as explained below.

*"In basins where a dense rainfall network is not available, Citizen Science (CS) offers a promising opportunity for enhancing the density of rainfall observations (Davids et al., 2019). With the popularization of smartphones and the availability of (relatively) simple and cheap equipment, abundant mobile applications and projects have been initiated in Water Resources Management to measure hydrometeorological variables like rainfall, water level height or water quality, as well as to ground-truth remotely sensed information on e.g. land use (Srivastra et al., 2018; Davids et al., 2019; See, 2019; Seibert et al., 2019). In the current study, we focus on rainfall measured by local citizens using a network of cheap rain gauges and a specially designed mobile application.*

*Since citizen rainfall observations are typically provided at daily scale, a temporal downscaling technique is needed for sub-daily applications. There has been a variety of temporal rainfall downscaling methods developed since the 1970s. The simplest approach is to distribute daily rainfall data to sub-daily resolutions by assuming uniform distributions. Stochastically generating sub-period data or spatially transferring finer resolution rainfall from a nearby rain gauge station to the study area based on spatial correlations are alternative approaches (Koutsoyiannis, 2003; Debele et al., 2007). However, these methods are not usually designed for real-time data disaggregation over large areas. Instead, a common approach for such scenarios is to downscale daily rainfall based on a simple fraction technique by considering the distribution patterns of high-resolution gridded rainfall products from radar or satellite sensors (Paulat et al., 2008; Wüest et al., 2010; Vormoor and Skaugen, 2013; Sideris et al., 2014; Barton et al., 2019)."*

**2. Study Area and Data**

RC (2.1): (Pg 5, L.125-126) I'm not clear with the sentence "A rain gauge with more than 80% of the dataset below the threshold was excluded from the analysis." Please explain.

AR: The meaning of the sentence "A rain gauge with more than 80% of the dataset below the threshold was excluded from the analysis." is that any rain gauge with more than 80% of the dataset being daily rainfall values of less than 0.5 mm was excluded from the analysis. To clarify, we replaced "80% of the dataset" by "80% of recorded rainfall amounts" on line 150 in the revised manuscript.

RC (2.2): (Pg 5, L.128-129) It is necessary to show the 14 rain gauges in Fig. 1 as well because the 14 rain gauges were intensively utilized in the following sections.

AR: Figure. 1 was changed to Figure 2 in the adjusted manuscript and was modified to show all used gauges including the 14 rain gauges in Chonburi and Rayong provinces as shown by the following.

[Figure]

*Figure 2 (in the revised manuscript): Location of study domain, showing Thai Meteorological Department (TMD) automatic rain gauges, citizen rain gauges, Sattahip radar, and Tubma basin.*

RC (2.3): (Pg.5, L. 131-136) Concerning the low-cost citizen rain gauges, are they tipping buckets? What kind of equipment are they? Were the measurements compared with the TMD rain gauges (As shown in Fig. 1, there are 2 TMD gauges very close to them)? If so, how uncertain were the data? Or, in the other case, were the quality checks performed in an intra-group manner. It is hard to tell from the current description. Please make it clear.

AR: More information on the citizen rain gauge observation was added in a new section "2.3.2 Citizen Rain Observation" in the revised manuscript as described in the first general comment. It is noted that in the revised manuscript Fig.2 shows total rain gauges and used rain gauge networks that passed the quality control. Consequently, there is only TMD rain gauge located in the Tubma basin and can be used for the measurement comparison between the citizen and TMD rain gauges.

**3. Methods**

RC (3.1): (Pg. 7, Table 1) Concerning the downscaling method "$G_{Tubma}$", instead of using only one TMD station within the basin (I noticed there is another one very close to the basin as shown in the small figure in Fig. 1), why not use this one as well. Besides, how variable are the hourly rainfall patterns across the basin. A comparison of the patterns from these two stations might provide useful information.

AR: Note that we have only one TMD rain gauge that passed quality control. This limitation then forced us to use only one TMD station for constructing the $G_{Tubma}$.

RC (3.2): (Pg. 14, L. 324): Please keep the term "KF-TMD" consistent with that shown in Table 2.

AR: We revised the manuscript accordingly.

**4. Results and discussion**

RC (4.1): (Pg. 15, L. 360): "..., while R_P and G_{MP} showed larger variability over the day, ...". There might be a mistake. The R_P-based result has a sharp mean cumulative fraction curve and large variance in the downscaled hourly data if one observes the box plots, whereas the other has a flat mean cumulative fraction curve and small variance in the downscaled hourly data. In other words, these two are different in both respects. The problem comes down to how the variability is defined. Please explain.

AR: The variability in that context defines the variability in the downscaled hourly data, which both $R_P$ and $G_{MP}$ exhibited a lot of outliers from the boxplot in each hour. P15L360 was rephrased for clearer understanding on the line 404 and 405 in the revised manuscript as explained below.

*"…, while $R_P$ and $G_{MP}$ showed larger variability in the downscaled hourly data with substantial outliers in the box plots, … "*

RC (4.2): (Pg. 18, Sect. 4.2.2) I am concerned about the validation scheme referred to as "KF - TMDD", especially about the separation of the gauges for performing KF/MFB and gauges for validation purposes. The bias correction was made for a large area with a radius of 100 km, whereas the validation was performed on a much smaller basin. I am afraid the separation makes the validation results less representative. Perhaps LOOCV for the 14 TMD gauge (as used in the "KF -TMD-H" strategy) is more persuading. If the purpose of using "KF -TMD-D" is to validate the bias correction performance within the basin, the authors should specify it explicitly (perhaps, also in Sect. 3.4). Anyhow, the separation mentioned above could be a bit problematic.

AR: For the KF -TMD-D case, we tried to imitate the actual situation if we have only 14 TMD rain gauges available for radar rainfall estimation in the Tubma basin, which bias adjustment method (MFB or KF) can produce more accurate radar rainfall in the basin. In Section 4.2.2, we, therefore, aimed to validate the bias correction performance within the basin by testing at all the citizen rain gauge locations. Any problems arising from the gauge's separation could be solved by the CKF-D. To clarify, sections 3.4 and 4.2.2 were modified in the revised manuscript as described below.

*The Sentence "To identify which approach between MFB and KF is more accurate for daily rainfall simulation in the Tubma basin if there are only 14 TMD rain gauges available, …" of section 3.4 in the revised manuscript were modified between Lines 370 to 371. Furthermore, "Results associated with validating the bias correction performance within the Tubma basin were presented in Fig. 6 (b). This shows …" was added to line 456 at the beginning of the paragraph of section 4.2.2 in the revised manuscript.*

Edit to text: on line 415, remove "be"

RC (4.3): (Pg. 19, L. 437-438) "While there is a modest improvement in mean RMSE, the upper 75%-ile RMSE is reduced from about 6 mm/h to 3.5 mm/h. Mean MBE is changed from 0.1 to -0.15 mm/h." I found it hard to follow here. Please use the terms presented in Fig. 7(b) to refer to the results.

AR: These sentences were adjusted between lines 482 and 485 in the revised manuscript as described follows.

*"While there is a modest improvement in mean RMSE (see the black line connecting the mean values of the box plots from MFB-TMD to CKF-R$_P$), the upper 75%-ile RMSE is reduced from about 6 mm/h to 3.5 mm/h. Mean MBE is changed from 0.1 to -0.15 mm/h (see the red-dotted line connecting the mean values from MFB-TMD to CKF-R$_P$).*

RC (4.4): (Pg. 19, L. 439) Please correct "MFB-TMB" and "KF-TMB" to "MFB-TMD" and "KF-TMD",respectively.

AR: The mistake was carefully corrected in the revised manuscript.

RC (4.5): (Pg 19-20, Sect. 4.3.4) I think the description of the results/figures could be organized in a more readable way. For example, there is a direct shift from the description of Fig. 7 to that of Fig. 8 (Pg.19, L. 440-444) without telling the readers that the content is related to Fig. 8, and there is a separation in the description of Fig. 7 (Pg. 19, L. 433- and Pg. 20, L. 451-).

*AR: Note that Figs. 7 to 9 in the old manuscript was reordered to be Figs. 8 to 10 in the revised manuscript.* For clearer understanding, we started section 4.3.4 of the revised manuscript with:

"Results for this section are presented in Figs. 8, 9, and 10." Additionally, *the following sentences, "Analysis of hourly rainfall hyetographs obtained from TMD rain gauge network compared with the validated rain gauge occurring in three different days are illustrated in Fig. 9. It appears … ." were inserted between lines 490 and 492 in the revised manuscript.* By the way, because Fig. 8 (in the revised manuscript) was the main results for section 4.3.4, while Fig.9 and 10 are supporting figures. Therefore, it is not easy to end all discussions related to Fig.8, followed by Fig.9 and 10, respectively. Therefore, please allow separation in the description of Fig. 8 as mentioned above.

*Edits to text:*

*-line 492 in the updated manuscript: replace "Figure 8 appears that the considerable RMSE occurs from…" by "It shows considerable RMSE from…"*

*-line 494 in the updated manuscript: replace "However, these RMSE can considerably reduce if…" by "However, these RMSE values decrease considerably if…"*

RC (4.6): (Pg. 19 at the end) I am not clear with the sentence " Figure 9 illustrates that hourly rainfall distribution patterns of TMD rain gauges in the 40-90 km range, influenced mainly by the southwest monsoon, appear to be more similar to the mean citizen rain gauge data than the range beyond 40 km." Please explain.

AR: In the mentioned sentence, we realize to change the words "40-90 km" to be "0-40 km" in the revised manuscript.

*Reference*

*Barton, Y., Sideris, I. V., Germann, U., and Martius, O.: A method for real-time temporal disaggregation of blended radar–rain gauge precipitation fields, 27, e1843, https://doi.org/10.1002/met.1843, 2020.*

*Davids, J. C., Devkota, N., Pandey, A., Prajapati, R., Ertis, B. A., Rutten, M. M., Lyon, S. W., Bogaard, T. A., and van de Giesen, N.: Soda Bottle Science-Citizen Science Monsoon Precipitation Monitoring in Nepal, Frontiers in Earth Science, 7, 10.3389/feart.2019.00046, 2019.*

*Debele, B., Srinivasan, R., and Parlange, J. J. A. i. W. R.: Accuracy evaluation of weather data generation and disaggregation methods at finer timescales, 30, 1286-1300, 2007.*

Koutsoyiannis, D.: Rainfall disaggregation methods: Theory and applications, in, 1–23, 10.13140/RG.2.1.2840.8564, 2003.

Mobile Water Management: https://mobilewatermanagement.nl/, last access: 25 October 2020.

Paulat, M., Frei, C., Hagen, M., and Wernli, H.: A gridded dataset of hourly precipitation in Germany: Its construction, climatology and application, Meteorologische Zeitschrift, 17, 719-732, 10.1127/0941-2948/2008/0332, 2008.

See, L.: A Review of Citizen Science and Crowdsourcing in Applications of Pluvial Flooding, 7, 10.3389/feart.2019.00044, 2019.

Seibert, J., Strobl, B., Etter, S., Hummer, P., and van Meerveld, H. J.: Virtual Staff Gauges for Crowd-Based Stream Level Observations, 7, 10.3389/feart.2019.00070, 2019.

Sideris, I. V., Gabella, M., Erdin, R., and Germann, U.: Real-time radar–rain-gauge merging using spatio-temporal co-kriging with external drift in the alpine terrain of Switzerland, 140, 1097-1111, https://doi.org/10.1002/qj.2188, 2014.

Srivastava, S., Vaddadi, S., and Sadistap, S.: Smartphone-based System for water quality analysis, Applied Water Science, 8, 10.1007/s13201-018-0780-0, 2018.

Vormoor, K. and Skaugen, T.: Temporal Disaggregation of Daily Temperature and Precipitation Grid Data for Norway, Journal of Hydrometeorology, 14, 989-999, 10.1175/JHM-D-12-0139.1, 2013.

Wüest, M., Frei, C., Altenhoff, A., Hagen, M., Litschi, M., and Schär, C.: A gridded hourly precipitation dataset for Switzerland using rain-gauge analysis and radar-based disaggregation, International Journal of Climatology, 30, 1764-1775, 10.1002/joc.2025, 2009.

**Author Response to the referee comments on "Citizen rain gauge improves hourly radar rainfall bias correction using a two-step Kalman filter"**

*by Punpim Puttaraksa Mapiam [1]*, Monton Methaprayun [1], Thom Adrianus Bogaard [2], Gerrit Schoups[2], Marie-Claire Ten Veldhuis[2]*

[1]Department of Water Resources Engineering, Kasetsart University, PO Box 1032, Bangkok, 10900, Thailand
[2]Department of Water Management, Delft University of Technology, PO Box 5048, 2600 GA Delft, The Netherlands

**Responses to referee #1**
RC: Referee comment
AR: Author response

**General comments**
RC: I found the topic of the manuscript highly interesting, as it exists a strong need to explore the potential of lower quality, but high density crowd-sourced observations. To this end, the authors suggest to use a Kalman filter, a well-established approach for data fusion in presence of uncertainty.

Following the invitation from the editor, I focused my attention on the methods of Section 3. In there, I could notice sound technical foundations and good consideration of previous work in the field. Despite these positive notes, I believe the clarity of the text in this Section can be improved, in particular by providing more details on the original contribution, while reducing some of the details from the basic algorithm that can be found in past references.

My understanding is that the methodology presented in Section 3.2 mostly follows the one already introduced in previous studies, particularly in Chomchean et al (2006), from which the authors seemed to have adopted the basic algorithm and notation. Therefore, it could be possible to move some of the mathematical details in the Appendix, and replace them with a higher-level summary of the algorithm together with a clear literature reference, so that the reader can more easily find the original formulations if needed.

In turn, this would allow to better highlight the original contribution of this study, which is the application of the algorithm to two sets of rain gauges of contrasting quality. I thus encourage the authors to consider some refactoring of Section 3. In particular, I found Section 3.2.2 a bit difficult to follow. I would suggest to first list more clearly the parameters needing estimation, and then explain the actual estimation methods alongside the underlying assumptions.
I have also listed below a number of minor comments and suggestions to the authors.

AR: Thanks for your valuable comments. To provide more clarity on the original contribution of the methodology in this study, we more explicitly mentioned the contributions of the study in Abstract, Introduction and Conclusions and reorganized Section 3.2 by stating the contributions directly at the beginning of the section, i.e. stating that "our approach extends a previously used Kalman filter radar bias model by including two different types of rainfall observations (data from

traditional and from citizen rain gauges) and by using a likelihood-based method for parameter estimation."

Furthermore, we refactored and clarified Section 3.2 as follows: instead of first describing the one-update/one-observation KF, followed by the two-update case, we focused each subsection on the two updates/observations (i.e. the contribution of our work). Namely:
-subsection 3.2.1 describes how radar bias is modelled using a Kalman filter model with two observations, contrasting it with previous KF bias models used in the literature.
-subsection 3.2.2 describes how observations are assimilated into the model, resulting in a KF filter with two updates instead of one. The two updates are now explicitly distinguished (also in response to reviewer 2), and this subsection has been streamlined so that it is easier to follow.
-subsection 3.2.3 describes how model parameters are estimated by maximizing marginal likelihood. Also this subsection was slightly rearranged to improve readability.

**Minor comments**

RC (1): Lines 210-211
The formulation "Since [...] . However [...]. " is unclear, please consider rephrasing.
AR: Since we refactored Section 3.2.1 as described above; these sentences were removed from the revised manuscript.

RC (2): Line 242
"comprising an updating (prediction) step" -> "comprising a time updating (prediction) step"
AR: Since we refactored Section 3.2.1 as described above; these sentences were removed from the revised manuscript.

RC (3): Equation 10
Why using $O\_t$ here to denote the observation at time t, while in Eq. 6 $y\_t$ was used?
AR: In the revised manuscript, $O_t$ was replaced with $y_t$ and $z_t$ to represent observed logarithmic mean field bias at hour t gained from the TMD and citizen rain gauge networks, respectively. Additionally, two steps of measurement updating using the $y_t$ and $z_t$ datasets were specified as presented in Eq (11) and Eq (14) of the modified manuscript.

RC (4): Line 272
Please check "and the a priori estimate error variance" -> "and the a posteriori estimate error variance"
AR: This mistake was corrected according to the suggestion.

RC (5): Equation 12

Without a measurement update, I would have expected to see this equation expressed simply as the posterior equal to the prior. Is there a particular reason for repeating the equations from the time update step?

AR: In the revised manuscript, we removed Eq.12 and added some sentences between lines 282 and 284 to explain how to calculate the CKF without a measurement update as shown below.

"If there is no observation data available at any time t, $K_{y,t}$ is zero (mathematically, a missing observation is equivalent to an observation with infinite variance $\sigma^2_{M_{y,t}}$ in Eq. 10), and the previous equations reduce to not performing any update , i.e. the posterior mean and variance are equal to the prior mean and variance."

RC (6): Equation 13

Why is the posterior estimate error variance P_t in absence of a measurement step not simply equal to Eq. 8?

AR: Thanks for pointing this out, we realize the equation in the previous manuscript is incorrect. If there is no observation available at any time t, the posterior mean and variance are equal to the prior mean and variance. In the revised manuscript, we removed Eq.13 and added some sentences between lines 282 and 284 to simply explain how to calculate the CKF without a measurement update as shown below.

"If there is no observation data available at any time t, $K_{y,t}$ is zero (mathematically, a missing observation is equivalent to an observation with infinite variance $\sigma^2_{M_{y,t}}$ in Eq. 10), and the previous equations reduce to not performing any update , i.e. the posterior mean and variance are equal to the prior mean and variance."

RC (7): Equation 14

The formula computes the mean of a log-normal distribution based on its parameters. I believe it would help the reader to remind this alongside the reference.

AR: For clearer explanation, we rewrote the sentences between line 299 and 301 in the revised manuscript as described below.

"Since the logarithmic mean field radar rainfall bias $\beta_t$ is assumed to be normally distributed, the mean field bias ($B_t$) is log-normal distributed with posterior mean at time $t$ obtained from the posterior mean and variance of $\beta_t$ (Smith and Krajewski, 1991):"

RC (8): Lines 295-297

The citizen rain gauge data are available only at the end of the day, meaning that their application cannot be in real time. Isn't this an important limitation since the Kalman filter is typically designed for real-time applications? Is this aspect worth mentioning and discussing, as for example in the conclusions?

AR: We appreciate this comment and agree with the referee to indicate the limitation of the modified KF approach in the conclusion of the updated manuscript between lines 551 and 555 as described below.

*"Note that citizen rain gauge data are available only at the end of the day and consequently, the modified two-step Kalman filter as used in this study has restrictions in real-time applications. However, the here proposed method has great potential when creating high quality historical radar-rainfall time series for climatological studies and in post-event analysis. Moreover, near real-time assessment could be achievable if the citizen rain gauge data are collected at sub-daily time scale."*

**Author Response to the referee comments on "Citizen rain gauge improves hourly radar rainfall bias correction using a two-step Kalman filter"**

*by Punpim Puttaraksa Mapiam [1*], Monton Methaprayun [1], Thom Adrianus Bogaard [2], Gerrit Schoups[2], Marie-Claire Ten Veldhuis[2]*

[1]*Department of Water Resources Engineering, Kasetsart University, PO Box 1032, Bangkok, 10900, Thailand*
[2]*Department of Water Management, Delft University of Technology, PO Box 5048, 2600 GA Delft, The Netherlands*

**Responses to referee #2**

RC: Referee comment
AR: Author response

**1. General comments**

RC (1): Thank you for this manuscript. Using citizen rain gauge data is an interesting approach and the authors analyzed the gathered data carefully and with a lot of different techniques. There seems to be a benefit in using these Citizen rain gauge data, so from my point of view, this study should be published.

I had some difficulties to understand where all the errors are coming from and how these errors are calculated. There are often some unreferenced assumptions on the error characteristics. Sometimes I would have wished to have a bit more insight into the error distributions of the different measurements.

There is also nothing mentioned on radar calibration errors. Most weather radars are not well calibrated and this obviously influences the estimation of $R$ via the $Z$ - $R$ relationship. Is this measurement bias automatically canceled out by calculating an individual $Z$ - $R$ relationship?

AR: For clearer explanation on radar rainfall estimation errors, we inserted more information in the revised manuscript between lines 31 and 35 as described below.

However, a weather radar provides an indirect measurement of backscattered electromagnetic waves called radar reflectivity data (Z) *"and quantitative estimation of radar rainfall data (R) is acknowledged as a complex process. Various sources of errors affect radar rainfall estimates, mainly errors in reflectivity measurements and reflectivity-rainfall conversion (Jordan et al., 2000). Correction of these two sources of error is a crucial procedure to increase the accuracy of radar rainfall estimates. Ground-truthing by rain gauge data is required to calibrate the Z-R relationship ($Z=AR^b$). The calibrated parameter A in the Z-R relationship will include any errors in the radar system caused by the electrical calibration of the radar (Seed et al., 2002)."*

RC (2): Regarding the KF and its mathematical formulation: I guess that this has been done properly, at least from my understanding. However, Kalman filters are especially great for multi-dimensional data sets with observation vectors being larger than 1. So I'm therefore not sure if

averaging everything together and calculating one bias value is the best approach. But at least it is an approach that shows some nice results.

AR: A few previous studies apply the Kalman filter for radar mean bias adjustment (Smith and Krajewski, 1991; Chumchean et al., 2006; Shi et al., 2018). Using the mean-field bias observed from only one rain gauge network and combining it with different techniques for assessing the observation noise variance at hour t ($\sigma^2_{M_t}$) is one of the inspiring tasks they tried to solve. In this study, we added one more observation dataset, while accounting for different error characteristics between the two datasets, to improve the measurement update procedures.

RC (3): There are still quite some language issues and after a couple of page I got tired to correct them all. So there should be some polishing done on that.

AR: English writing has been thoroughly checked in the revised manuscript.

**2 Detailed comments**

RC (1): Line 12: *of Sattahip radar station and gauge rainfall* → of Sattahip radar station, gauge rainfall...

AR: This mistake was corrected in the revised manuscript.

RC (2): Line 27: *accuracy of flash flood estimates and warning.* → accuracy of flash flood estimates and warnings.

AR: This mistake was corrected in the revised manuscript.

RC (3): Line 28: *However, weather radar provides indirect measurement of backscattered electromagnetic waves called radar reflectivity data (Z).* → However, a weather radar provides an indirect measurement of backscattered electromagnetic waves called radar reflectivity data (Z).

AR: This mistake was corrected in the revised manuscript.

RC (4): Line 30: *ground-truthing by rain gauge data is required to calibrate the Z-R relationship (Z = ARb)* → I'm not happy with this sentence. It's not a priori clear that just the Z-R relation needs to be calibrated. Biases between the radar rainfall intensity can also stem from radar calibration

errors, i.e., from a bias in Z. Most radar operators do not alter the Z-R relationship at all, but just adjust the radar rainfall vs gauge rainfall bias (Sideris, 2014).

AR: Thank you for pointing this out, we added more information and rephrased the sentence for clearer explanation in the revised manuscript between lines 31 and 35 as explained below.

However, a weather radar provides an indirect measurement of backscattered electromagnetic waves called radar reflectivity data (Z) *"and quantitative estimation of radar rainfall data (R) is acknowledged as a complex process. Various sources of errors affect radar rainfall estimates, mainly errors in reflectivity measurements and reflectivity-rainfall conversion (Jordan et al., 2000). Correction of these two sources of error is a crucial procedure to increase the accuracy of radar rainfall estimates. Ground-truthing by rain gauge data is required to calibrate the Z-R relationship (Z=AR$^b$). The calibrated parameter A in the Z-R relationship will include any errors in the radar system caused by the electrical calibration of the radar (Seed et al., 2002)."*

RC (5): Line 60: *variances affecting the mean field bias estimate*.

AR: This mistake was corrected in the revised manuscript.

RC (6): Line 76: *However, these methods are not usually designed for real-time* → However, these methods are usually not designed for real-time

AR: This mistake was corrected in the revised manuscript.

RC (7): Figure 1: I can't see the location of the radar in this map. It's indicated in the legend but not on the map (or hardly visible).

AR: Figure 1 was updated according to the comments in the revised manuscript as shown below.

[Figure]

Figure 1: Climatological spatial rainfall distribution in and around the Tubma basin calculated from 30-year average annual rainfall data of 311 daily rain gauge network by using IDS method.

RC (8): Line 107: *The Tubma basin is covered within the range of Sattahip radar station.* → The Tubma basin is located within the coverage of the? Sattahip radar station.

AR: This sentence was adjusted in the revised manuscript.

RC (9): Line 107: *a beam width of 1.0°* → a half power beam width of 1.0°

AR: This mistake was corrected in the revised manuscript.

RC (10): Line 109: *The radar reflectivity product is in a Cartesian* → The radar reflectivity product is provided in a Cartesian

AR: This sentence was adjusted in the revised manuscript.

RC (11): Line 110: *The Sattahip radar provides the CAPPI reflectivity data derived from the 2.5-km constant altitude plan position indicator (CAPPI).* → CAPPI derived from CAPPI: that's not a meaningful sentence.

AR: This sentence was corrected as appeared below.

*"The Sattahip radar provides the reflectivity data derived from the 2.5-km constant altitude plan position indicator (CAPPI)"*

RC (12-1): Line 114: *Additionally, the noise and hail effects were eliminated by setting reflectivity values below 15 dBZ to zero, and reflectivity values greater than 53 dBZ to 53 dBZ.* → I'm not sure if this is allowed. Could you give some reasoning on this method or a reference?

AR: Hail is a type of precipitation in the form of ice with a diameter of at least 5 mm (Rinehart, 1991). Reflectivity from dry hail has a lower reflectivity than wet hail of the same size. The apparent reflectivity of hail particles further increases when the hail melts and becomes coated with a film of water leading to an over-estimation of rainfall rates. Fulton et al. (1998) suggested that measured reflectivity that are greater than 53 dBZ are limited to 53 dBZ in radar rainfall estimation to mitigate false interpretation caused by hail. This reference was inserted in the revised manuscript at line 123.

RC (12-2): If I use a Marshall-Palmer relation ($Z = 200\ R^{1.6}$ then 100 mm of rain already give 55 dBZ. 100 mm are not unrealistic in the tropics, even higher values are possible. But probably Marshall-Palmer is not suitable in the tropics?

AR: The Z-R parameters significantly change by raindrop size distribution, hence there is no universal relationship connecting these parameters (Doviak and Zrnic, 1992). Marshall and Palmer was calibrated based on stratiform rainfall characteristics in Ottawa, Canada, the A and b parameters are then unrealistic to directly apply in different areas. In this study, the optimal relationship was $Z=251R^{1.5}$, as described in section 2.2.2. Applying the calibrated relationship to the maximum 53 dBZ produces 86 mm/h of rainfall, while Marshall and Palmer (1948) give 75 mm/h. In the other area with heavier rainfall characteristics, the Z-R parameters typically differ for converting the same reflectivity to higher rainfall rate. For example, the relation $Z=300R^{1.4}$ (Fulton et al., 1998) appropriate for storm characteristic in United States can be used to convert 53 dBZ of reflectivity to 103 mm/h of rainfall.

RC (12-3): And clipping below 15 dBZ might be ok, but reflectivity values below this value or not necessarily due to noise, especially not receiver noise. It might need a specification of what you mean by 'noise'.

AR: Noise is any undesired electrical disturbance or spurious signal which enters the radar receiver. Noise power is composed of signals originating at various sources such as emission from space (cosmic noise), radiation from electrical sources near the radar antenna, and internally generated noise. Variations in these noise sources occur, often at random intervals and are challenging to control. To avoid accumulation of the power of noise, measured radar reflectivity below 15 dBZ was set to zero (Fulton et al., 1998 and Doviak and Zrnic, 1984).

To clarify noise and hail issues, we rephrased the sentences between lines 121 and 125 in the revised manuscript as describe below.

*"Additionally, the noise power caused by various sources such as emission from space (cosmic noise), radiation from electrical sources near the radar antenna, and internally generated noise were eliminated by setting reflectivity values below 15 dBZ to zero (Doviak and Zrnic, 1984). While Fulton et al. (1998) suggested that measured reflectivity that are greater than 53 dBZ are limited to 53 dBZ in radar rainfall estimation to mitigate false interpretation caused by hail."*

RC (13): Line 137: *radar rainfall accumulation (mm/h)* → Accumulation is not in mm/h

AR: In the revised manuscript, the description of $R_{i,t}$ was changed from "radar rainfall accumulation (mm/h)" to "the hourly radar rainfall accumulation (mm).

RC (14): Line 141: *was validated against a second, independent dataset. Results found that a locally calibrated ZR relationship that was used in this* → I don't understand this sentence. What is the *Z - R* relation of the climatological dataset? What is the *Z - R* relation of the locally calibrated dataset?

AR: To clarify this point, this sentence was rewritten in the revised manuscript between lines 149 and 151 as appeared below.

*"Results found that the optimal climatological Z-R relationship for the Sattahip radar used in this study is $Z=251R^{1.5}$. This relation is appropriate for both the calibration and validation datasets with the MAE of 1.36 mm and 1.47 mm, respectively."*

RC (15): Line 146: *These 15-min rain gauges*: → These rain gauge data have a temporal resolution of 15 minutes

AR: This sentence was adjusted in the revised manuscript.

RC (16): Line 148: *double mass curves method*: → What is this?

AR: Double mass curves is a traditional method to be used for checking the consistency of gauge rainfall data. The cumulative data of a single station are plotted against the mean accumulated rainfall of adjacent gauges in the area. If the slope of the double mass curve tends to be straight with a single slope, it can ensure the reliability of rainfall data at that considered rain station.

RC (17): Line 155: *based on spatial decorrelation analysis in the process.* → based on spatial decorrelation analysis for this? process.

AR: This sentence was adjusted in the revised manuscript.

RC (18): Line 180: RMSE does not tell you much about the bias between the TMD and the citizen gauge. Bias needs to be given as well.

AR: According to the comment, the bias between the TMD and the citizen gauge was calculated as 1.04 and specified in the revised manuscript at line 190 and in the supplementary between lines 7 and 15.

RC (19): Line 188: *First, daily citizen...* → From line 126 onward you were using 'Firstly, Secondly...'.

AR: This line First was changed to Firstly as the suggestion.

RC (20): Line 206: I'm not sure if 'noise' is the right word here. KF accounts for measurement errors or uncertainties, but not for noise.

AR: The word 'noise' was replaced with uncertainties in the updated manuscript.

RC (21): Line 208: *different uncertainty characteristics, i.e., hourly...*

AR: This sentence was adjusted in the revised manuscript.

RC (22): Line 210: *Since the MFB (G/R ratio) is assumed to follow a log-normal distribution.* → Has this sentence a relationship to the previous sentence? And what is an ordinary KF scheme.

AR: Since we refactored Section 3.2.1 to describes how radar bias is modelled using a Kalman filter model with two observations, contrasting it with previous KF bias models used in the literature. This sentence was removed from the revised manuscript.

RC (23): Line 300: *downscaled hourly citizen rain gauge data were used to back-calculate the hourly citizen rain gauges data* → I don't understand this.

AR: This mistake was corrected in the revised manuscript between lines 312 and 313 as follows.

*"The TMD and downscaled hourly citizen rain gauge data were together used to conduct two steps of a second measurement update in the CKF process for all hourly time-steps of day i."*

RC (24): Figure 5: This is Kalman filter bias corrected rain accumulation data from gauges?

AR: Figure 5 is not the Kalman filter bias, but it shows the different hourly rainfall distribution patterns at each day generated from four downscaling techniques. The rainfall distribution patterns were presented as the cumulative fraction of daily rainfall at an hourly scale. Daily citizen rain gauge data were afterwards multiplied by the corresponding hourly fraction factor to obtain the hourly downscaled citizen rain gauge data to be used as input for the KF bias correction.

RC (25): Line 420: *%-ile* → percentile

AR: All the words '*%-ile*' were replaced with percentile in the revised manuscript.

RC (26): Line 508: *obviously appear steady light rainfall accumulation* → ?

AR: This sentence was rephased in the revised manuscript as described below.

*"Figure 10 (c).... obviously appear a steady gradient of the mass curve reflecting light rainfall accumulation, .."*

RC (27): Figure 9: *Rainfall Depth* → probably not the right annotation.

AR: Figure 9: Rainfall Depth was changed to Hourly Rainfall (mm) in the revised manuscript.

**3 Kalman filter**

I guess I have understood your KF approach, but there are some questions left:

RC (28) $y_t$ and $z_t$ are your log-transformed bias observations from the TMD and the citizen gauge network, respectively, corresponding to $O_t$ in Eq. 10?

AR: Yes, you are correct. We therefore updated the equations relating to $y_t$ and $z_t$ as shown in Eq (11) and Eq (14) of the modified manuscript for the first and second measurement updating, respectively.

RC (29) Figure 4: In the second KF step you take the variance estimates *Pt* from the previous KF step as a priori variances?

AR: Yes, it is correct. Confusion caused by the symbols was fixed in the revised manuscript.

RC (30) You calculate the Kalman gain $K_t$ in the second step with these *Pt* variances. From my point of view, this Kalman gain should be named differently, since $K_t$ is already used for the first step.

AR: We agree with the referee to separate symbols differently to distinguish the two updates. Please have a look at section 3.2.2 in the revised manuscript between lines 260 and 303 for new changes as appeared below.

*"3.2.2 Kalman filter with two observations: data assimilation*

Having defined the model, we describe how observations are assimilated into the model, resulting in adjustments of the estimated bias. While the regular Kalman filter has two steps (prediction and update), assimilating two observations at each time step involves three steps, i.e. a prediction followed by two updates. Figure 3 (b) shows a visual depiction of the prediction and update steps as probabilistic information propagating along the edges of the factor graph. .

*1) Time update step (prediction)*

The first step for each hour *t* computes the prior mean $\hat{\beta}_t^-$ and variance $P_t^-$ of $\beta_t$ :

$$\hat{\beta}_t^- = r_1\hat{\beta}_{t-1} \tag{8}$$

$$P_t^- = r_1^2 P_{t-1} + \sigma_W^2 \tag{9}$$

where *Pt-1* is the posterior variance of $\beta_{t-1}$. For the first time step 0 (*t* = 0), we assume $\beta_0$ = 0 (climatological logarithmic mean field bias) and $P_0$ = (1-$r_1^2$) $\sigma_\beta^2$ (represents stationary process variance) (Smith and Krajewski, 1991; Chumchean et al., 2006).

*2) The first measurement update step (1$^{st}$ correction)*

The first update merges the bias prediction from step 1 with noisy observation $y_t$ (with measurement error variance $\sigma_{M_{y,t}}^2$), resulting in posterior mean $\hat{\beta}_{y,t}$ and variance $P_{y,t}$ of $\beta_t$ given by the following Kalman update equations:

$$K_{y,t} = P_t^- \left( P_t^- + \sigma_{M_{y,t}}^2 \right)^{-1} \tag{10}$$

$$\hat{\beta}_{y,t} = \hat{\beta}_t^- + K_{y,t}(y_t - \hat{\beta}_t^-) \tag{11}$$

$$P_{y,t} = (1 - K_{y,t})P_t^- \tag{12}$$

where $K_{y,t}$ is the Kalman gain for assimilating observation $y_t$. If there is no observation available at time $t$, $K_{y,t}$ is zero (mathematically, a missing observation is equivalent to an observation with infinite variance $\sigma_{M_{y,t}}^2$ in Eq. 10), and the previous equations reduce to not performing any update, i.e. the posterior mean and variance are equal to the prior mean and variance.

*3) The second measurement update step (2^nd correction)*

The second update is done using the posterior values from the 1st correction ($\hat{\beta}_{y,t}$ and $P_{y,t}$) as the prior values. The resulting Kalman gain and posterior mean and variance are given by:

$$K_{z,t} = P_{y,t}\left( P_{y,t} + \sigma_{M_{z,t}}^2 \right)^{-1} \tag{13}$$

$$\hat{\beta}_{z,t} = \hat{\beta}_{y,t} + K_{z,t}(z_t - \hat{\beta}_{y,t}) \tag{14}$$

$$P_{z,t} = (1 - K_{z,t})P_{y,t} \tag{15}$$

If there is no observation available at time $t$, Kalman gain $K_{z,t}$ is zero and these equations result in no update being applied, i.e. posterior mean and variance are the same as after the first update.

Since the logarithmic mean field radar rainfall bias $\beta_t$ is assumed to be normally distributed, the mean field bias ($B_t$) is lognormally distributed with posterior mean at time $t$ obtained from the posterior mean and variance of $\beta_t$ (Smith and Krajewski, 1991):

$$\hat{B}_t = 10^{(\hat{\beta}_{z,t} + 0.5P_{z,t})} \tag{16)''}$$

RC (31) Likewise, the equation (2) in the 2nd KF step in Figure 4 should be rewritten, since the value on the left side of the equation ($\hat{\beta}_t$) is an update of the bias value on the right side.

AR: This mistake in Figure 4 was corrected according to the suggestion as shown below. The mentioned equation was changed to $\hat{\beta}_{z,t} = \hat{\beta}_{y,t} + K_{z,t}(z_t - \hat{\beta}_{y,t})$

[Figure]

Figure 4: A diagram of the procedure of Kalman filter combined with the citizen rain gauge data (CKF)

RC (32) Same is true for Eq. 3 in Figure 4, 2nd step: $P_t$ is an update of $P_t$ on the right side of the equation.

AR: This mistake in Figure 4 was corrected according to the suggestion. The mentioned equation was changed to. $P_{z,t} = (1 - K_{z,t})P_{y,t}$

RC (33) I'm somehow irritated by the usage of these capital letters $K$ and $P$, which, at least for me, would represent matrices. In KF theory, $P$ then represents the error covariance matrix, but since you are using variances only it is somehow strange to go from $\sigma^2_{Mz,t}$ to $Pt$. Probably just an unimportant detail.

AR: Yes, in this paper the Kalman gain and variances P are scalar numbers instead of matrices. To make this clear to the reader, we avoid the usual bold/upright notation for these variables as would be common if they were actually matrices.

RC (34) Line 229 onwards: Error estimation of $\sigma^2_{Mz,t}$ and $\sigma^2_{My,t}$: I did not fully understand this. $\sigma^2_{Ot}$ is the (spatial?) variance of all stations? And why is Equation 19 necessary if you calculate $\sigma^2_{Mz,t}$ and $\sigma^2_{My,t}$ by the spatial variance of the individual stations?

AR: In the revised manuscript, we refactored and clarified Section 3.2 to provide more clarity on the original contribution of the methodology in this study. We consider two types of observations, $y_t$ and $z_t$, ($O_t$ was replaced with $y_t$ and $z_t$) of the unknown bias at time $t$, derived from the TMD and citizen rain gauges. Each observation is modelled as a random sample from a normal distribution conditioned on the underlying unknown bias with distinct measurement error variances $\sigma^2_{M_{y,t}}$ and $\sigma^2_{M_{z,t}}$.

$$y_t = \beta_t + M_{y,t}; \quad M_{y,t} \sim \mathcal{N}(0, \sigma^2_{M_{y,t}}) \tag{6}$$

$$z_t = \beta_t + M_{z,t}; \quad M_{z,t} \sim \mathcal{N}(0, \sigma^2_{M_{z,t}}) \tag{7}$$

where $\sigma^2_{M_{y,t}}$ and $\sigma^2_{M_{z,t}}$ are time-varying measurement error variances for the TMD and citizen rain gauges, respectively. We estimate $\sigma^2_{M_{y,t}}$ and $\sigma^2_{M_{z,t}}$ using formulas for the variance of the mean bias across individual TMD and citizen rain gauges:

$$\sigma^2_{M_{y,t}} = \frac{\sigma^2_{y,t}}{n_{y,t}} \tag{17}$$

$$\sigma^2_{M_{z,t}} = \frac{\sigma^2_{z_t}}{n_{z,t}} \tag{18}$$

where $\sigma^2_{y,t}$ and $\sigma^2_{z,t}$ are variances that quantify spatial variability of radar bias at time $t$ at TMD and citizen rain gauge locations, respectively, and $n_{y,t}$ and $n_{z,t}$ are the corresponding number of observations at hour $t$ from the two networks.

Reference

Doviak, R. J. and Zmic, D. S.: Doppler Radar and Weather Observations, 1st edition, Academic Press Inc., San Diego, California, U.S.A., 575 pp., 1984.

Doviak, R. J. and Zmic, D. S.: Doppler Radar and Weather Observations, 2nd edition, Academic Press Inc., Orlando, Florida, U.S.A., 545 pp., 1992.

Fulton, R. A., Breidenbach, J. P., Seo, D.-J., Miller, D. A., O'Bannon, T. J. W., and forecasting: The WSR-88D rainfall algorithm, 13, 377-395, 1998.

Rinehart, R. E.: Radar for meteorologists, University of North Dakota, Office of the President1991.

Seed A., Siriwardena L., Sun X., Jordan P., and Elliot J.: On the calibration of Australian weather radars, Cooperative Research Centre for Catchment Hydrology, Australia, 49 pp., 2002.

**Author Response to the referee comments on "Citizen rain gauge improves hourly radar rainfall bias correction using a two-step Kalman filter"**

*by Punpim Puttaraksa Mapiam [1*], Monton Methaprayun [1], Thom Adrianus Bogaard [2], Gerrit Schoups[2], Marie-Claire Ten Veldhuis[2]*

[1]*Department of Water Resources Engineering, Kasetsart University, PO Box 1032, Bangkok, 10900, Thailand*
[2]*Department of Water Management, Delft University of Technology, PO Box 5048, 2600 GA Delft, The Netherlands*

**Responses to Editor**
RC: Referee comment
AR: Author response

**Minor comments**

RC (1): Lines 94-95. Please convert the coordinates to decimal degrees.
AR: The coordinates were converted to decimal degrees according to the comments.

RC (2): Equation 1 can be deleted as it is already mentioned in the introduction.
AR: Equation 1 was deleted in the revised manuscript.

3. Same for equations 21 and 22. You mentioned RMSE earlier to the mention of the equation and both equations are well known.
AR: Equations 21 and 22 were removed in the revised manuscript.

4. Consider plotting the circle presented in Figure 2 also in Figure 1 (maybe to replace the current circles).
AR: Figure 1 was modified by adding the circle for 240 km in the revised manuscript as shown below.

[Figure]

5. Section 4.1.2 and Figure 5 - can be moved to the SI? What essential information is provided in this subsection?

AR: We agree with the editor to move Section 4.1.2 and Figure 5 to SI.

6. Figure 9. Consider moving to SI.

AR: Figure 9 and related content between lines 493 to 502 in the old manuscript were moved to SI as the comment.

**Author Response to the referee comments on "Citizen rain gauge improves hourly radar rainfall bias correction using a two-step Kalman filter"**

*by Punpim Puttaraksa Mapiam [1*], Monton Methaprayun [1], Thom Adrianus Bogaard [2], Gerrit Schoups[2], Marie-Claire Ten Veldhuis[2]*

[1]*Department of Water Resources Engineering, Kasetsart University, PO Box 1032, Bangkok, 10900, Thailand*
[2]*Department of Water Management, Delft University of Technology, PO Box 5048, 2600 GA Delft, The Netherlands*

**Responses to referee #5**
RC: Referee comment
AR: Author response

**General comments**
RC: Thank you for the revision of this manuscript. This is significant work and hence the article should be published. I have one small comment that could be addressed before the article is being published:

- line 123-124: Regarding the hail correction, i.e., the truncation of the reflectivity at 53 dBZ: you have inserted a reference (Fulton et al. (1998) to justify the limitation to 53 dBZ. However, in the cited article it is written that "The nationwide default setting of the hail cap is currently 104 mm/h (53 dBZ), but a number of radar sites in more tropical environments along the gulf coast use higher values such as 150 mm/h".

With Z-R relationship you are using (251* Z *R^(1.5)), 150mm/h per hour would correspond to almost 57~dBZ of radar reflectivity. I understand that it is difficult to find that right value to eliminate hail from the measurements without eliminating too much rain signal. However, it might be worthwhile to add a sentence or two on this subject to clarify that this is a somehow subjective choice and that maybe a higher value could have been chosen for a tropical environment.

AR: Thanks for your valuable comments. To clarify this issue, we added two sentences between lines 126 and 128 in the revised manuscript as explained below.

*"The hail cap can be seen as an adaptable threshold representing the maximum expected instantaneous rain rate which is unfortunately quite difficult to determine for a particular storm. Note that in tropical environments also slightly higher values have been reported as hail threshold."*